# Comprehensive analysis of *Saccharomyces cerevisiae* intron structures in vivo

Ramya Rangan[1], Rui Huang[2], Oarteze Hunter[3], Phillip Pham[2], Manuel Ares Jr.[3] & Rhiju Das [1,2,4] ✉

Pre-mRNA secondary structures are hypothesized to regulate RNA processing pathways, but such structures have been difficult to visualize in vivo. Here, we characterize *Saccharomyces cerevisiae* pre-mRNA structures through transcriptome-wide dimethyl sulfate probing, enriching for low-abundance pre-mRNA through splicing inhibition. We cross-validate structures found from phylogenetic and mutational studies and identify structures within the majority of measured introns (79 of 88). We find widespread formation of 'zipper stems' between the 5′ splice site and branch point, 'downstream stems' between the branch point and the 3′ splice site, and previously uncharacterized long stems that distinguish pre-mRNA from spliced mRNA. Multi-dimensional chemical mapping reveals intron structures that independently form in vitro without the presence of binding partners, and structure ensemble prediction suggests that such structures appear in introns across the *Saccharomyces* genus. We further develop a high-throughput functional assay to characterize variants of RNA structure (VARS-seq), applying it to 135 sets of stems across 7 introns, identifying structured elements that alter retained intron levels at a distance from canonical splice sites. This transcriptome-wide inference of intron RNA structures introduces alternative paradigms and model systems for understanding how pre-mRNA folding influences gene expression.

Introns are widely prevalent features of eukaryotic genomes. Many genes contain long stretches of these non-coding RNA sequences, which are excised from mRNA precursors through RNA splicing. In the splicing reaction, the spliceosome precisely recognizes and positions three key intronic sequences termed the 5′ splice site, the branch point and the 3′ splice site, carrying out the two catalytic steps required for removing introns (Fig. 1a)[1]. Despite the prevalence of introns, the functional roles for many of them remain underexplored. In some cases, intron sequences beyond splice sites regulate gene expression by controlling splicing rates and promoting alternative splicing[2,3]. In addition, introns can contain functional non-coding RNAs, alter pre-mRNA decay rates and facilitate the evolution of new genes[4–10].

Intron RNA sequences are complex macromolecules that occupy an ensemble of secondary structures, including single- and double-stranded regions. Intron secondary structure can regulate splice-site selection and splicing efficiency, and structure in pre-mRNA can have a role in numerous other nuclear processes, such as RNA editing and RNA end processing[11,12]. *S. cerevisiae* provides a useful model system for studying the role of pre-mRNA secondary structures because the catalytic steps of splicing, spliceosomal machinery, RNA modification processes, and RNA-decay pathways are highly conserved across eukaryotes, from *S. cerevisiae* to humans[1,13,14]. Research on *S. cerevisiae* has revealed that intron stems called zipper stems can link the 5′ splice site to the branch point[15–18], and hairpins can lower the effective distance between the branch point and 3′ splice site, facilitating

[1]Biophysics Program, Stanford University, Stanford, CA, USA. [2]Department of Biochemistry, Stanford University, Stanford, CA, USA. [3]RNA Center and Department of Molecular, Cell & Developmental Biology, University of California, Santa Cruz, Santa Cruz, CA, USA. [4]Howard Hughes Medical Institute, Stanford University, Stanford, CA, USA. ✉e-mail: rhiju@stanford.edu

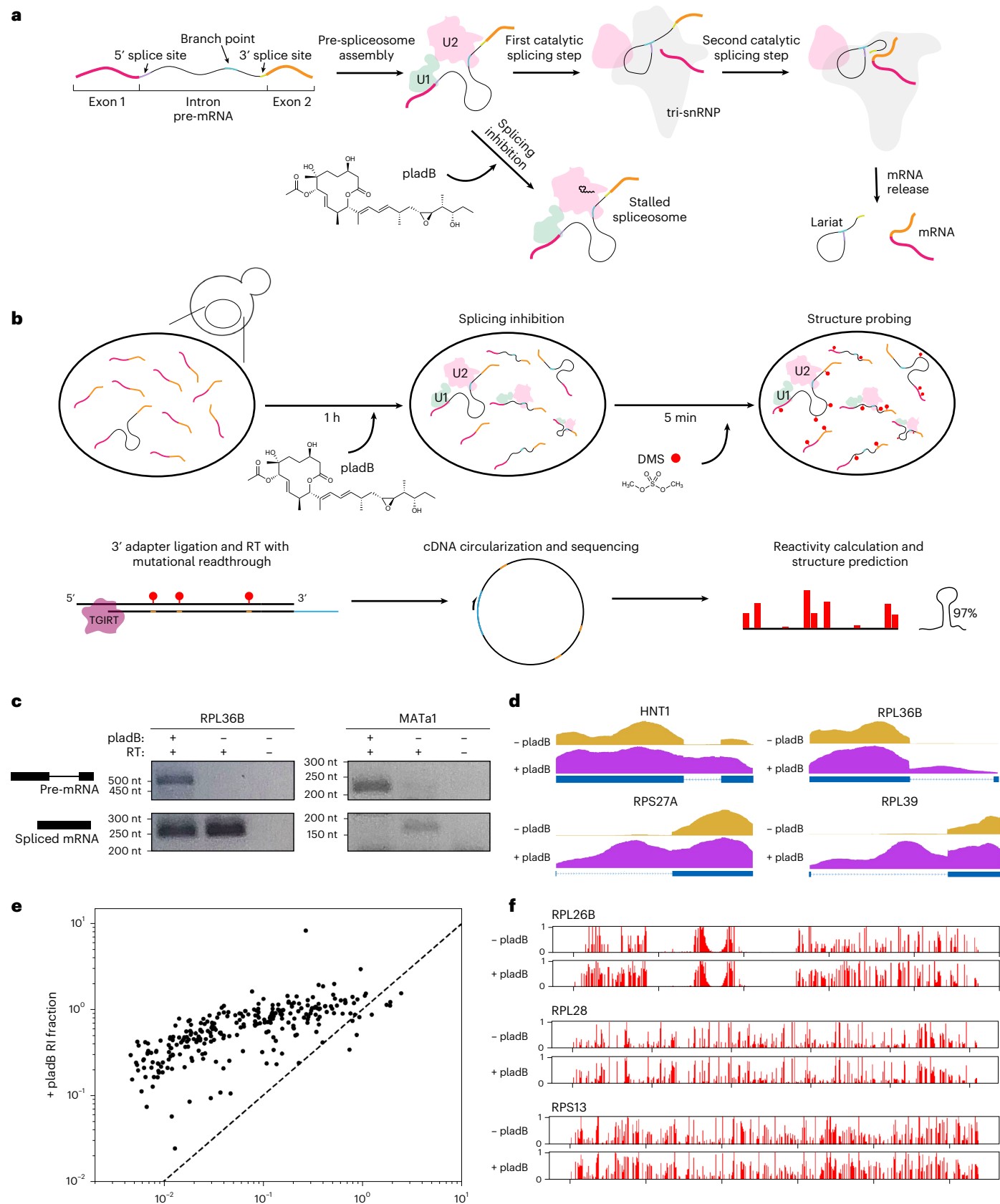

**Fig. 1 | Splicing inhibition by pladB allows for accumulation of pre-mRNA.**
**a**, Schematic of RNA splicing. **b**, Schematic of splicing inhibition, followed by the
DMS-MaPseq experiment. **c**, Accumulation of pre-mRNA for *RPL36B* and *MATa1*,
assessed using RT–PCR. This experiment was repeated with a biological replicate,
with similar results. **d**, Read coverage across intron-containing pre-mRNA with

(purple) and without (orange) pladB treatment. **e**, The RI fraction with and
without pladB treatment. Points are plotted on a log scale, and the equal retained
fraction from both conditions is indicated by the dashed line. RI, retained intron.
**f**, Comparison of reactivity values for three introns with and without pladB
treatment.

efficient splicing[19]. In addition, pre-mRNA structures can bind to their protein products to regulate autogenous gene expression control at the level of splicing[7,20–22].

These studies highlight various functions of pre-mRNA structures, but it remains unclear how generally these findings extend across the transcriptome. A comprehensive experimental survey of structures in introns across the transcriptome of any organism is lacking, and functional data that test the role of native intron structures are limited. Although scans for covariation have identified some potentially functional structures[23], others uncovered in functional studies have been missed by these scans[20–22,24]. Moreover, while sequence variants have been used to test some intron structures[3,25], the depth of mutagenesis has been limited[26]. At this stage, the structural landscape and functional roles for intron secondary structures across any transcriptome are only sparsely determined.

In vivo chemical probing provides an avenue for obtaining deep structural data on RNA across the transcriptome[27]. However, owing to their low abundance, introns typically escape structure detection and quantification by these methods. Here, we use splicing inhibition to enrich for unspliced pre-mRNA in transcriptome-wide structure probing experiments, identifying patterns in reactivity that distinguish structures in *S. cerevisiae* introns from coding regions in vivo. We combine this structural information with phylogenetic analysis across yeast genomes, and we develop a strategy for high-throughput functional analysis called VARS-seq to evaluate levels of unspliced and spliced RNA for variants of 135 sets of stems in 7 introns. Our combined structural, functional and evolutionary analysis provides an atlas of pre-mRNA structures, serving as a foundation for understanding the roles of introns in gene regulation.

## Results

### Transcriptome-wide structure probing with splicing inhibition in yeast

Structure probing experiments such as dimethyl sulfate (DMS) mutational profiling with sequencing (DMS-MaPseq)[27] can be used to evaluate the formation of RNA secondary structures across the transcriptome. However, because splicing proceeds rapidly in yeast[28], the coverage of pre-mRNA is limited in existing DMS-MaPseq datasets for *S. cerevisiae*, preventing the analysis of introns. Recently, *S. cerevisiae* strains with substitutions in the U2 snRNP component encoded by *HSH155* (*SF3B1* in human) have been developed that sensitize yeast to splicing inhibition by pladienolide B (pladB)[29,30]. Typically, splicing proceeds in two catalytic steps: the 5′ splice site and the branch point interact in the first catalytic step, and the 5′ splice site and the 3′ splice site participate in the second catalytic step (Fig. 1a)[1]. Using a pladB-sensitive yeast strain, we stalled the assembling pre-spliceosome (A complex) before the first catalytic step of splicing, chemically inhibiting splicing for 1 h before DMS treatment (Fig. 1b).

Treatment with pladB led to the accumulation of pre-mRNA, increasing the proportion of unspliced mRNA from *RPL36B* and *MATa1* (Fig. 1c). DMS-MaPseq revealed increased intron retention in pladB-treated cells for many genes; intron read coverage increased from negligible levels in untreated cells to levels approaching those of coding regions (Fig. 1d). Most intron-containing genes showed increased intron retention following pladB treatment, yielding a median ratio of retained intron fraction (RI fraction) of 10.5 when comparing pladB and control conditions (Fig. 1e). Only 1 intron had an RI fraction less than 0.05, and 180 introns had an RI fraction above 0.5. Without pladB treatment, 143 of 288 annotated introns had an RI fraction less than 0.05, and only 28 introns had an RI fraction above 0.5. Longer introns (more than 200 nucleotides (nt)) and introns in ribosomal protein genes (RPGs) exhibited increased RI fractions upon pladB treatment compared with those of other introns (Supplementary Fig. 1a,b). We noted no coverage bias between the ends of introns (Supplementary Fig. 1c), suggesting that they are not partial

degradation byproducts of nonsense-mediated decay (NMD), which predominantly involves XRN1-mediated 5′ to 3′ decay[31,32]. RI fractions for introns detected with and without pladB treatment are shown in Supplementary Table 1.

To assess the quality of our DMS-MaPseq reactivity data for introns, we examined control RNAs with known structure. DMS modifies A and C residues, leading to substitution frequencies of 2.7% and 2.3%, respectively, whereas unreactive G and U residues exhibited substitution frequency rates closer to the background level (0.3–0.4%) (Extended Data Fig. 1a). Data for rRNA aligned with prior experiments, showing that substitution frequencies for the 18S and 25S rRNA were highly correlated with values from an earlier DMS-MaPseq study ($r^2 = 0.83$, Extended Data Fig. 1b)[27]. Moreover, DMS reactivity values, a proxy for RNA single-strandedness, permitted successful classification of accessible and inaccessible rRNA residues, with an area under the curve (AUC) of 0.91 (Extended Data Fig. 1b). Finally, the reactivity profiles for stem loops in *HAC1* and *ASH1* mRNA align with well-characterized secondary structures for these regions[27], with base-paired residues being less reactive than residues in loops (Extended Data Fig. 1c).

### Assessing structures from DMS-MaPseq after splicing inhibition

Using data generated from these DMS-MaPseq experiments, we predicted secondary structures with RNAstructure[33,34]. We also assigned confidence estimates for individual helices by performing non-parametric bootstrapping on reactivity values, using an approach that has empirically improved the quality of stem predictions[35]. To calibrate helix confidence estimate cut-offs, we made DMS-guided structure predictions for structured RNAs in our DMS-MaPseq dataset, including rRNAs, small nuclear RNAs (snRNAs), tRNAs and mRNA segments. The implementation of helix confidence estimate thresholds improved the positive predictive value (PPV) and $F_1$ score for stem predictions (Extended Data Fig. 2). When using reactivity data for DMS-guided structure predictions with a helix confidence estimate threshold of 70%, stems with at least 5 base pairs (bp) had a PPV of 82.3% (Supplementary Table 2). This approach remained effective even for larger RNAs, such as the U1 snRNA (Extended Data Fig. 2d,e and Supplementary Table 2). We therefore designated stems with helix confidence estimates of >70% and at least 5 bp as high-confidence stems and focused subsequent analyses on them.

We next assessed the reproducibility of these high-confidence stems. Across all 259 introns, we identified 425 such stems, with 331 (77.6%) agreeing between replicate experiments. Introns with higher sequencing coverage had better replicability between experiments (Extended Data Fig. 3a). In the 88 introns with higher reactivity correlation between replicates ($r^2 > 0.6$), 88.0% of high-confidence stems were consistent between replicates (Extended Data Fig. 3b). To confirm the presence of these stems, we carried out targeted probing for 30 introns to obtain higher-coverage data. DMS profiles for 24 of these introns yielded $r^2 > 0.9$ between replicates of targeted probing (Extended Data Fig. 3c). In these introns, 90.5% of high-confidence stems identified by transcriptome-wide DMS-MaPseq matched structures from targeted probing (Extended Data Fig. 3d and Supplementary Fig. 2). To verify that DMS data informed predicted structures, we extracted RNA from cells and probed it with DMS after heat denaturation, revealing that only 10.4% of these stems were present in denatured RNA (Extended Data Fig. 3e and Supplementary Fig. 2). We focused all further analyses on the 88 introns with $r^2 > 0.6$ from DMS-MaPseq, limiting analysis to high-confidence stems. We identified 301 stems across these 88 introns, with the majority (79 of 88) having at least 1 stem. With additional DMS-guided structure prediction approaches, we found that no introns included high-confidence pseudoknots, and all introns tested for multiple structures were best explained by a single conformation (see Supplementary Text). Supplementary Table 1 details introns'

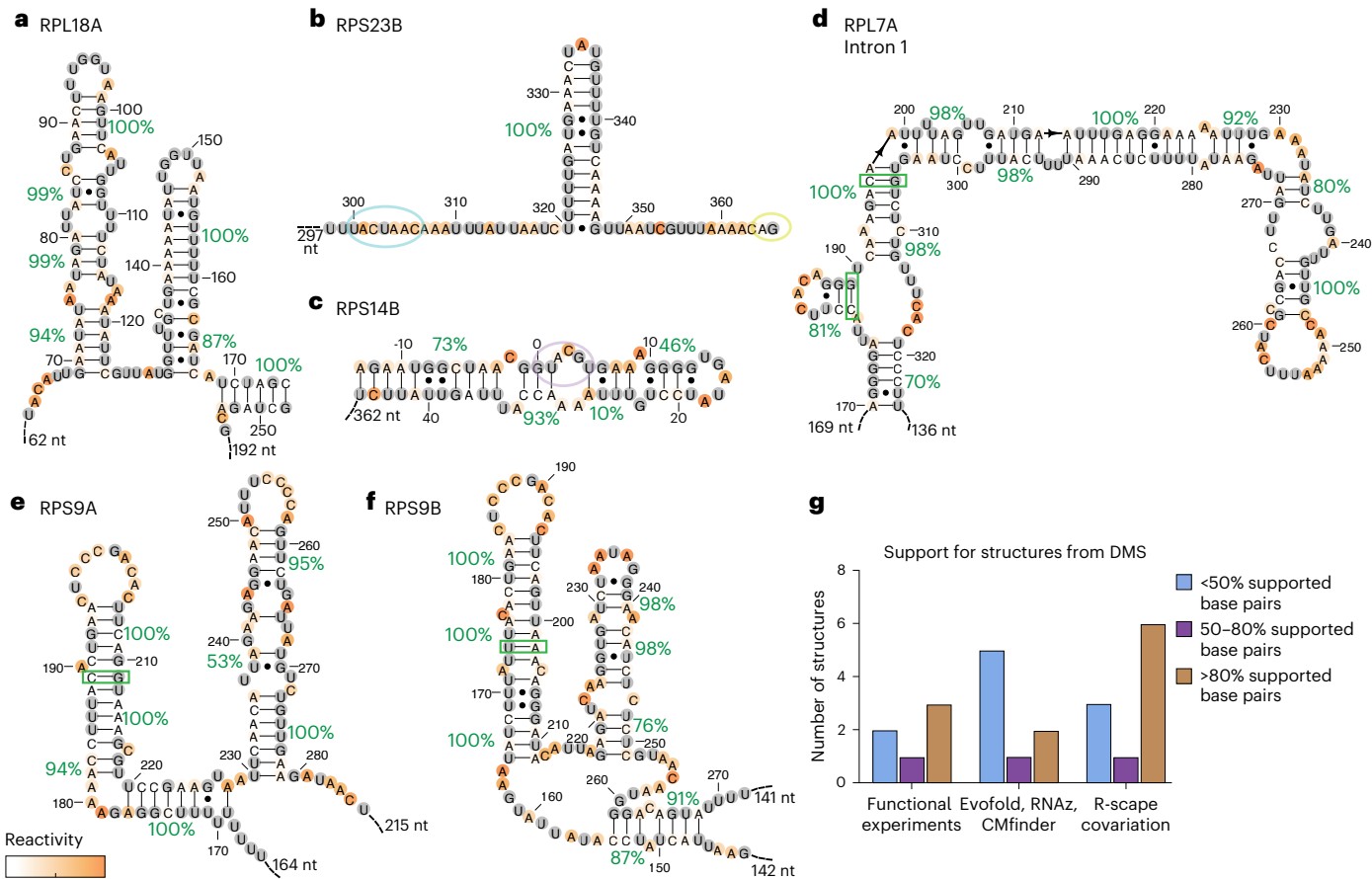

**Fig. 2 | Support from DMS reactivity for in vivo formation of control structures and proposed functional structures. a–f,** Helix confidence estimates and covariation for intron structures reported in *RPL18A*[24] (**a**), *RPS23B*[19] (**b**), *RPS14B*[21] (**c**), the first intron in *RPL7A* (**d**), *RPS9A* (**e**) and *RPS9B*[7] (**f**). Secondary structures are colored according to DMS reactivity, and helix confidence estimates are depicted as green percentages. The 5′ splice site, branch point and 3′ splice site sequences are circled in purple, blue and yellow, respectively. Covarying base pairs in *RPS9A*, *RPS9B* and *RPL7A* are marked with green boxes. **g,** Summary of the percentage of supported base pairs in structures proposed in prior functional studies, R-scape scans for covariation in multiple sequence alignments (MSAs), and other approaches using sequence alignments to pinpoint structures (Evofold, RNAz and cMfinder). A base pair is supported if it is included in a stem whose helix confidence estimate is >70%, and base-pair support statistics are computed on the basis of all base pairs in proposed structures (functional experiments, Evofold, RNAz, cMfinder) or significantly covarying base pairs (R-scape covariation).

replicate correlation, high-confidence stems and structures from transcriptome-wide and targeted probing.

To assess the impact of pladB treatment on intron folding, we examined the three introns that had sufficient DMS-MaPseq coverage, both with and without pladB treatment. The reactivity profiles of these introns were broadly similar, with many highly reactive positions shared between conditions (Fig. 1f). However, some intervals in the introns of *RPL26B* and *RPS13* exhibited shifted reactivity profiles upon pladB treatment, surpassing the variation seen between replicates (Supplementary Fig. 3). When comparing conditions with and without pladB treatment, the reactivities had $r^2$ values of 0.58, 0.86 and 0.71 for the introns in *RPL26B*, *RPL28* and *RPS13*, respectively. To evaluate whether these differences affected structure prediction, we identified high-confidence stems predicted for these three introns with and without pladB treatment. We found that all high-confidence stems were shared across conditions (Supplementary Fig. 4), suggesting that structures obtained after pladB treatment could provide insights into the structures in untreated *S. cerevisiae*.

**DMS support for previously reported yeast intron structures**

Structures have been proposed for some introns in *S. cerevisiae* on the basis of functional experiments[7,15,16,19–22,24], solved spliceosome structures including introns[25,36], covariation scans that pinpoint functional base pairs[23] and de novo prediction methods based on sequence conservation[37]. Using our DMS data, we evaluated the support for the presence of these proposed structures in vivo.

We first focused on seven introns that included regulatory structures identified in functional experiments: introns in *RPS17B*[15,16], *RPS23B*[19], *RPL32* (ref. 20), *RPS9A*[7], *RPS14B*[21,22], *RPL18A*[24] and *RPS22B*[24]. Most of these structures were supported by our DMS data (Fig. 2a–e), with only two exceptions (Supplementary Fig. 5a,b), validating structures found through experiments assessing mutants and compensatory mutants (Fig. 2g and Supplementary Text). By contrast, structures identified through computational predictions in Hooks et al.[37] using CMfinder[38], RNAz[39] and Evofold[40] showed low support from DMS data (Supplementary Fig. 6a–f), suggesting that these approaches are less reliable for structure identification (Fig. 2g and Supplementary Text). DMS-guided predictions include some false negatives (33.8% false negative rate, Extended Data Fig. 2a,b) and will miss structures that represent a minor portion of introns' structural ensembles.

We next evaluated structures detected using R-scape, which identifies pairs of residues with significant covariation compared with phylogenetic sequence backgrounds. In particular, we evaluated DMS support for the covarying residues identified in Gao et al.[23] and those identified in our R-scape scan using less stringent cut-offs (Supplementary Fig. 7 and Methods). Seven of the eight *S. cerevisiae*

introns that encode snoRNAs[41] included significant covariation, suggesting that the remaining introns with covariation could also encode functional structures. Most intron structures with covarying residues were supported by DMS data (Fig. 2d–f and Supplementary Fig. 8), with one exception (Supplementary Fig. 5c and Supplementary Text), suggesting that covariation detected by R-scape[42] can reliably identify structures (Fig. 2g). Thus, we found that, compared with structures predicted by CMfinder[38], RNAz[39] and Evofold[40], those identified through functional experiments and R-scape[42] covariation were more consistently supported by DMS data.

We also used our DMS data to identify high-confidence stems in tRNA introns in *S. cerevisiae*. Given that these tRNA introns are not processed by the standard spliceosome machinery, pladB treatment did not lead to their accumulation. Despite this, ten tRNA introns had sufficient coverage for structure analysis. Each of these introns included at least one high-confidence stem (Supplementary Fig. 9), aligning with prior pre-tRNA structure models, and again confirming structures from prior structural and functional experiments[43,44].

## Structural features found by probing *S. cerevisiae* pre-mRNA

Having established that our DMS-guided structure predictions are in agreement with previously identified intron structures, we next identified enriched structural features in our data. First, in vivo probing supports the widespread formation of zipper stems, that is, stems that reduce the distance between the 5' splice site and branch point. Potential zipper stems have been noted in various introns[17], and a zipper stem in the intron from *RPS17B* has been shown to be essential for efficient splicing[15,16]. To identify zipper stems across introns, we sought to precisely define the positional constraints on zipper stem formation, modeling intronic stems with varying linker lengths in the context of the A-complex spliceosome (Extended Data Fig. 4a,b and Methods). On the basis of this modeling, we defined zipper stems as the longest stem comprising 42 to 85 nt linking the stem, the 5' splice site and branch point sequences. We found that high-confidence zipper stems (those with at least 6 bp and a minimum 70% helix confidence estimate) were predicted for 32 of 88 introns (Fig. 3a). In our targeted DMS probing of 30 introns, zipper stems were present in 25 of them (examples in Supplementary Fig. 2), whereas there were no zipper stems present when DMS data were collected after unfolding with heat denaturation, as expected (Extended Data Fig. 3d and Supplementary Fig. 2).

We next noted the presence of stems between the branch point and 3' splice site across introns, which we term downstream stems. These stems are proposed to facilitate splicing by decreasing the effective distance between the branch point and 3' splice site[19]. We found that downstream stems (those with at least 6 bp and a minimum 70% helix confidence estimate) are present in 21 of 88 introns (Fig. 3b). As with zipper stems, the reactivity profiles for downstream stems (for example, in introns in *RPL40B*, *RPS14A* and *RPS23B*) show higher reactivity in loop or junction residues than in base-paired positions, supporting their in vivo formation (Figs. 2b and 3b).

DMS reactivity patterns in some introns suggest the presence of elaborate extended secondary structures with long stems and multiway

junctions. For example, the intron from *RPL28* features a zipper stem and a downstream stem, and the pre-mRNA includes a three-way junction and long stems of more than 10 bp throughout the intron (Fig. 3c). When modeling the *RPL28* intron in the context of the A-complex spliceosome, we observe that these structures enable internal intronic stems to extend beyond the core spliceosome, potentially allowing binding partners to avoid steric clashes with the splicing machinery (Fig. 3d). We note that the three-dimensional (3D) modeling approach here makes simplifying assumptions and samples only a few of many possible conformations (see Methods). However, sampled conformations suggest that intron structures can extend beyond the spliceosome, even for shorter introns such as that in *RPL36B*, facilitated by stems through the intron (Extended Data Fig. 4c).

## Comparing intron structures with coding regions and unspliced decoys

Our DMS-guided structure predictions revealed that the properties of intron secondary structures are distinct from those of mRNA coding regions. As found previously in mammalian structure probing[45], intron regions have higher Gini coefficients, a measure quantifying the extent to which reactivity values diverge from an even distribution (Fig. 3e). Therefore, introns include more non-random secondary structure elements than do coding regions. Additionally, intron structures predicted using DMS data extend further from their sequence endpoints than do coding-region structures, which we measure as the normalized maximum extrusion from ends (MEE) (Fig. 3f and Methods). Introns also include longer stems, with some high-confidence stems extending to more than 20 bp (Fig. 3g). Finally, intron stems have higher helix confidence estimates on average, suggesting that reactivity values better support intron secondary structures (Fig. 3h). These conclusions were reproduced when analyzing data from each DMS-MaPseq replicate separately (Supplementary Fig. 10).

The secondary structure patterns enriched in introns might distinguish spliced introns from other transcribed sequences that do not splice efficiently. To explore this possibility, we assembled a set of unspliced decoy introns in *S. cerevisiae* that included consensus 5' splice site, branch point and 3' splice site sequences in positions that matched canonical introns' length distributions (Extended Data Fig. 5a). DMS-guided structure predictions for authentic intron sequences enriched for more stable zipper stems with lower folding free energy (ΔG), more stable downstream stems, longer stems and higher MEE (Extended Data Fig. 5b). In contrast to spliced introns, the DMS-guided secondary structures of unspliced decoys were not enriched for these features (Extended Data Fig. 5b). We conclude that, compared with coding regions and unspliced decoys, introns in *S. cerevisiae* adopt extended secondary structures with long stems, as supported by DMS data.

The enrichment of structural features in introns compared with coding regions and unspliced decoys suggests that these structures could have a functional role. To explore this possibility, we analyzed the placement of high-confidence stems in introns relative to the positions of canonical and cryptic splice sites. To evaluate structures

**Fig. 3 | Structural insights from DMS probing of *S. cerevisiae* introns.**
**a**, Reactivity support for zipper stems in *RPL7A* and *RPS11A*, and a pie chart representing the fraction of introns with zipper stems. 5'SS, 5' splice site.
**b**, Reactivity support for downstream stems connecting the branch point and 3' splice site in *RPL40B* and *RPS14A*, and a pie chart representing the fraction of introns with downstream stems. 3'SS, 3' splice site. **c**, The secondary structure of the intron in *RPL28*, predicted by RNAstructure guided by DMS reactivity.
**d**, The top-scoring 3D model for the *RPL28* intron in the context of the A complex spliceosome (PDB ID: 6G90)[74], modeled using the secondary structure derived from DMS-MaPseq. **e–h**, Comparisons between introns and coding regions for the following secondary structure features: the Gini coefficient (**e**), normalized maximum extrusion from ends (**f**), longest stem length (**g**) and average helix confidence estimate (**h**). *P* values were computed using one-sided Wilcoxon

ranked-sum tests to compare classes. In box plots, the median is the center white point, box limits are the 25th (Q1) and 75th (Q3) percentiles, and whiskers extend to the smallest and largest values that fall within 1.5 times the interquartile range below Q1 and above Q3. **i**, Proportion of nucleotides in high-confidence stems from sequence intervals surrounding the canonical 5' splice site, branch point and 3' splice site sequences. Intron sequence positions external to these intervals were marked as 'distal from SS.' **j**, Comparison of the protection by high-confidence stems between nucleotides in cryptic splice sites versus surrounding nucleotides. An example from *RPL34A* is shown with a stem occluding a cryptic 3' splice site (red bracket). Secondary structures are colored by DMS reactivity and are annotated with helix confidence estimates. The 5' splice site, branch point and 3' splice site sequences are circled in purple, blue and yellow, respectively. In **i–j**, *P* values were computed with Chi-squared tests on 2 × 2 contingency tables.

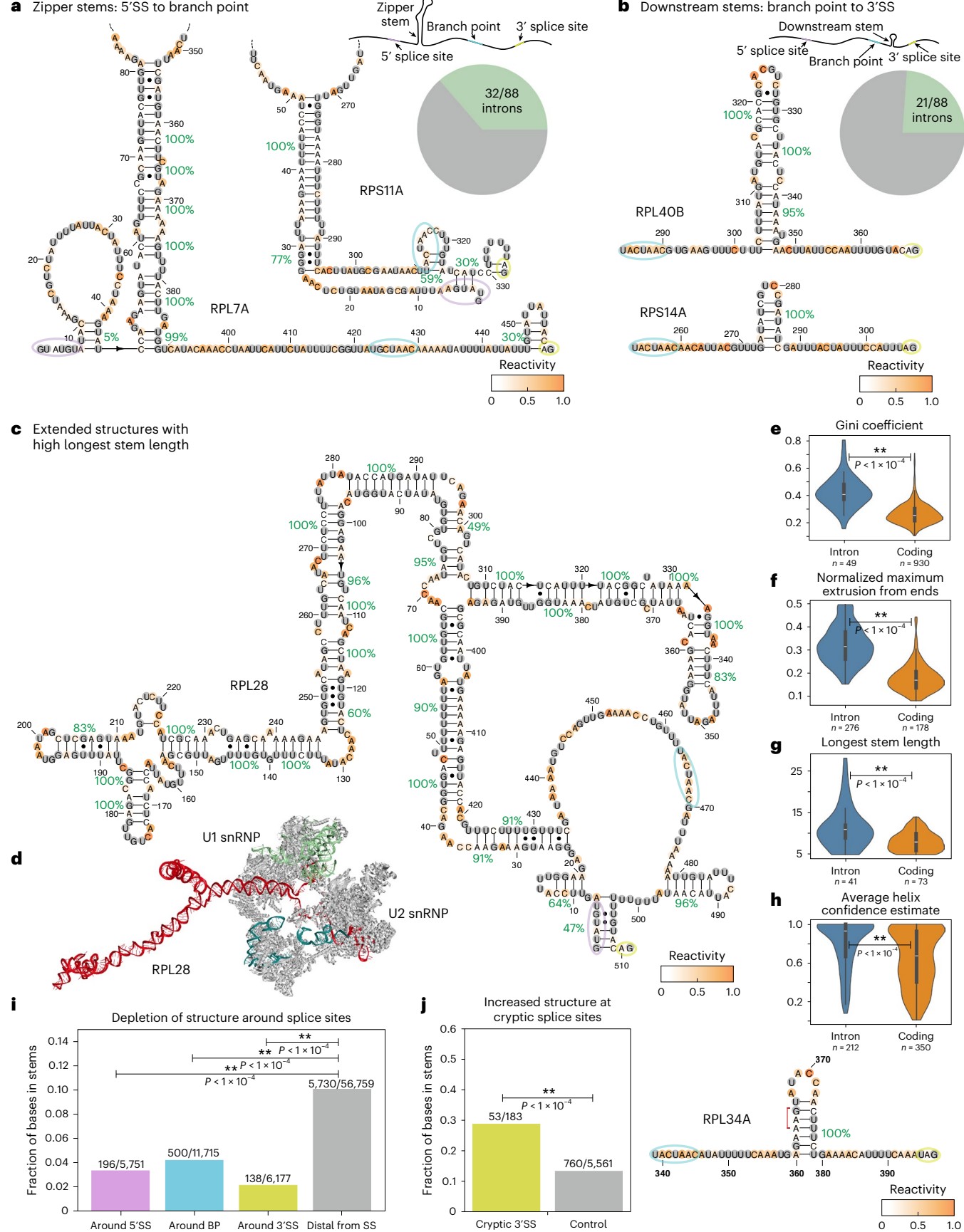

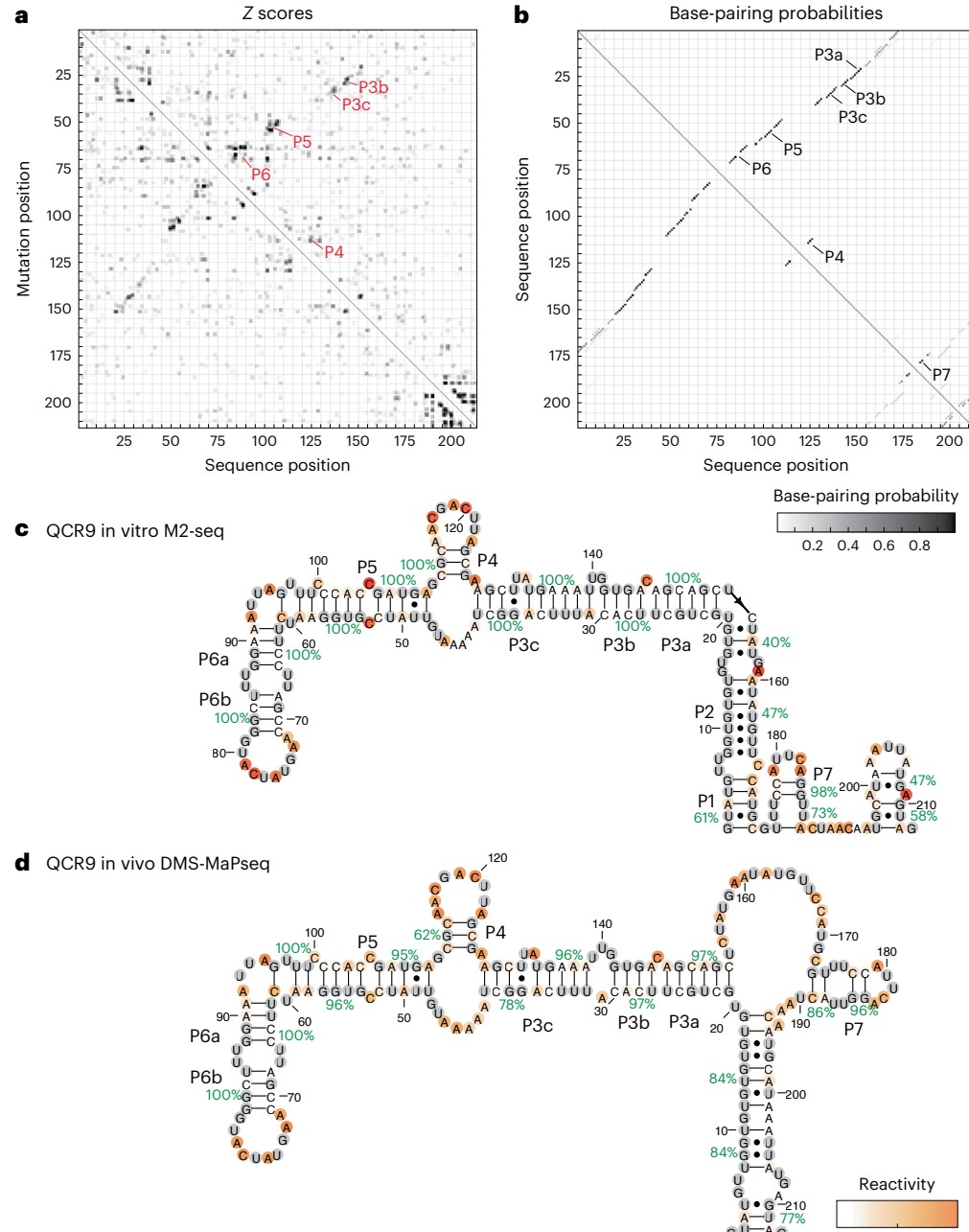

**Fig. 4 | Comparing in vivo and in vitro folding of intron RNA structures. a**, In vitro M2-seq *Z* scores for the intron in *QCR9*, with peaks representing helices annotated in red. **b**, In vitro chemical reactivity base-pairing probabilities for the *QCR9* intron using one- and two-dimensional (1D and 2D) chemical reactivity from M2-seq, with peaks representing helices annotated in black. **c**,**d**, Secondary structure predictions guided by 1D and 2D DMS probing data for the intron in *QCR9* from in vitro M2-seq (**c**) and in vivo DMS-MaPseq (**d**).

surrounding canonical splice sites, we first identified sequence intervals in pre-mRNA surrounding the 5′ splice site, branch point and 3′ splice site, where structural conflicts with the spliceosome are anticipated (Methods). We noted that these intervals were significantly depleted of high-confidence stems when generating structures for introns in the context of the surrounding pre-mRNA sequence (Fig. 3i). However, the presence or absence of high-confidence stems involving these intervals near splice sites was not correlated with the fraction of spliced constructs, suggesting that other factors have a dominant role (Supplementary Fig. 11a). We next identified cryptic splice sites across introns, expecting that these sites could be occluded by structure. Indeed, in multiple introns, downstream stems between the branch point and 3′ splice site occluded cryptic 3′ splice sites (Supplementary

Fig. 11b), aligning with prior work on an intron stem in *RPS23B* that blocks a cryptic 3′ splice site (Fig. 2b)[19]. Across introns, we noted that cryptic 3′ splice sites were enriched for high-confidence stems (Fig. 3j and Supplementary Fig. 11c), suggesting that these stems have a role in enforcing splicing fidelity by repressing the use of incorrect 3′ splice sites.

**Evaluating *S. cerevisiae* intron RNA structures in vitro**

To determine whether intron structures observed in vivo can form in vitro even when potential protein binding partners are missing, we probed isolated introns transcribed in vitro with DMS. For this, we used mutate-and-map readout through next-generation sequencing (M2-seq[46]), which can identify base-pairing residues in addition to

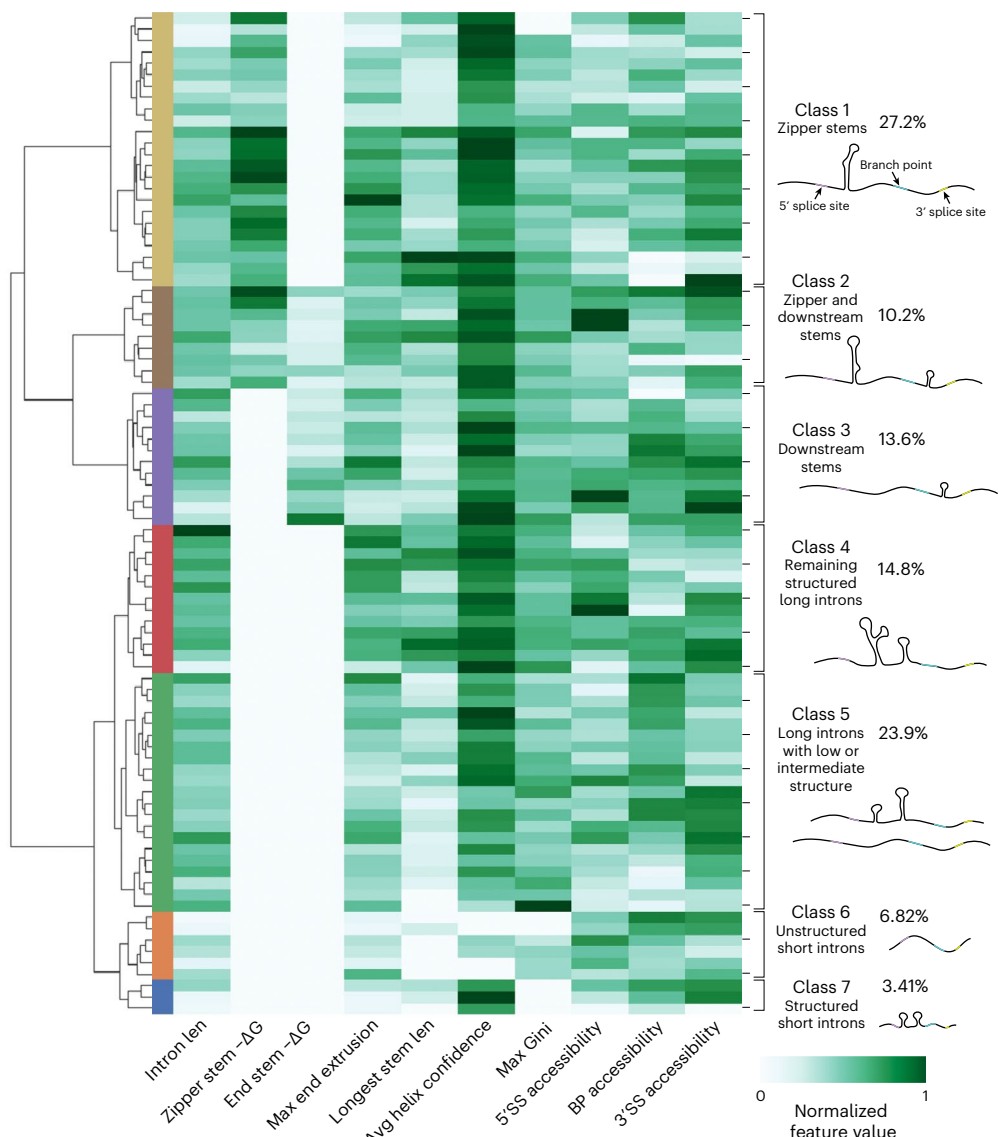

**Fig. 5 | Structural landscape for *S. cerevisiae* introns.** Heatmap and dendrogram summarizing intron structural classes, with hierarchical clustering based on secondary structure features. In addition to the features displayed on the heatmap, flags indicating whether zipper stems and downstream stems were present were included as features for hierarchical clustering. ΔG, folding free energy; len, length; SS, splice site; BP, branch point.

providing average per-residue accessibility data. We applied in vitro M2-seq to five introns with zipper stems in DMS-MaPseq. In the case of the introns in *QCR9* and *RPL36B*, *Z*-score plots from in vitro M2-seq included off-diagonal signals indicative of the presence of stems, and high base-pairing probabilities support the formation of these stems in vitro (Fig. 4a,b, Extended Data Fig. 6a,b and Supplementary Text). Secondary structures from M2-seq agree with the stems observed in vivo, with most high-confidence stems shared between these structures (Fig. 4c,d and Extended Data Fig. 6c,d). For the introns in *RPS11A*, *RPL37A* and *RPS7B*, although M2-seq *Z*-scores did not include off-diagonal signals, helix confidence estimates from M2-seq data supported stems observed in vivo (Extended Data Fig. 7). We additionally assayed intron structures in vitro by refolding RNA extracted from yeast and probing accessible residues with DMS (Methods). However, because only 3 introns reached a between-replicate reactivity correlation of $r^2 > 0.6$ from this experiment (Supplementary Fig. 12), we focused our analyses on cases studied with in vitro M2-seq. Our in vitro M2-seq results suggest that intron sequences can form structures found in vivo, even outside the nucleus and without protein binding partners.

## Structural landscape for *S. cerevisiae* introns

To understand the distribution of structural patterns in introns across the yeast transcriptome, we clustered introns on the basis of their structural features. For each intron, we assembled a full set of features using probing data (Supplementary Table 1): zipper stem and downstream stem free energy (Methods), the MEE, the longest stem length, the average helix confidence estimate, the maximum Gini coefficient window, and the accessibility across the 5' splice site, branch point and 3' splice site.

The clustered heatmap (Fig. 5) depicts the global distribution of secondary structure features across all introns with sufficient sequencing coverage. The first class of introns includes 24 introns with zipper stems; within this class, more stable zipper stems clustered together (class 1, yellow in Fig. 5). Class 2 includes nine introns that have both a zipper stem and downstream stem, with high average helix confidence estimates (class 2, brown in Fig. 5). The next class includes 12 introns with downstream stems (class 3, purple in Fig. 5). In *S. cerevisiae*, intron lengths are bimodal, with a set of shorter introns (shorter than 200 nt) and a set of longer introns (with most between 400 and 500 nt)[47]. Our structure probing data had sufficient coverage for 7 short introns

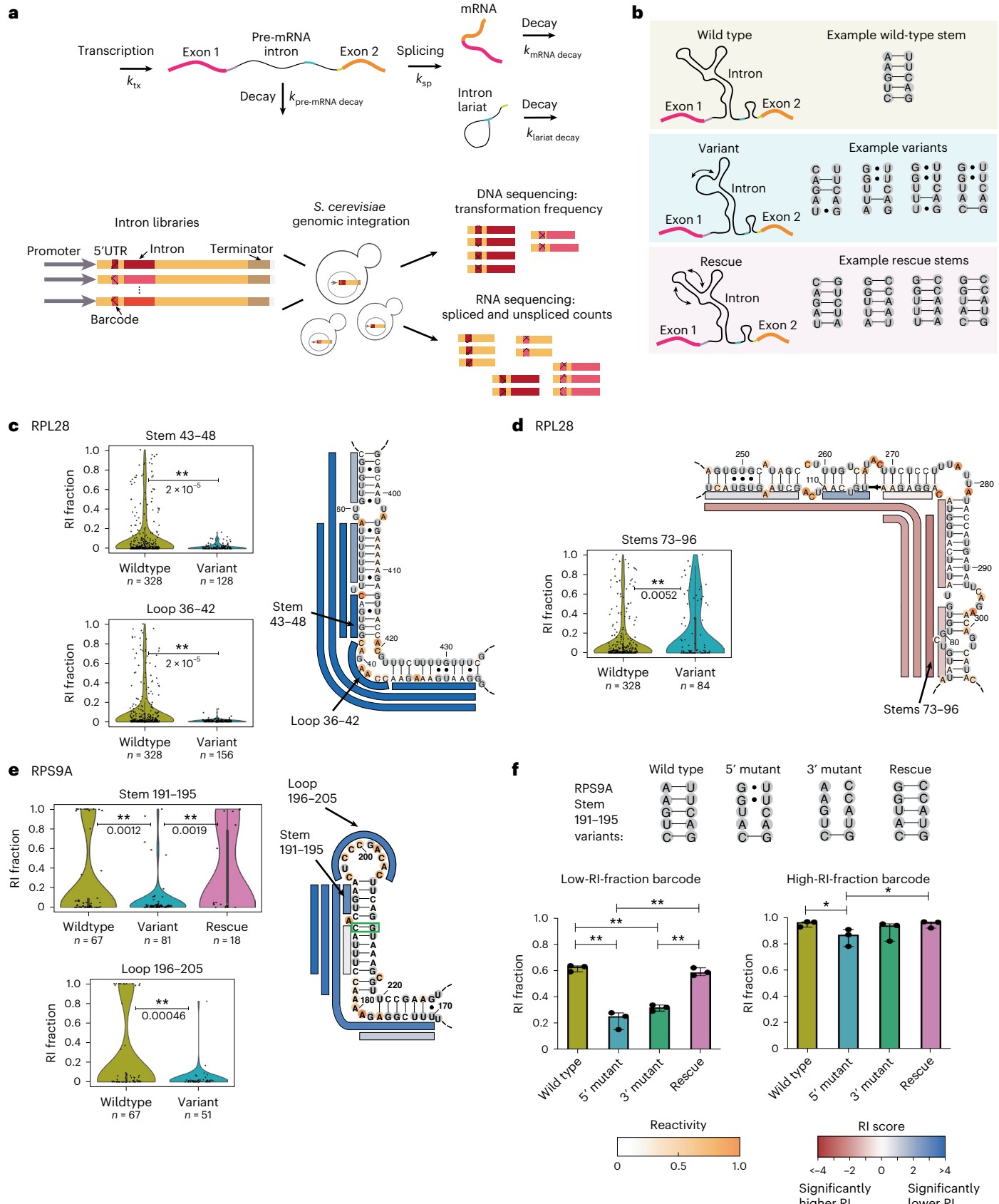

and 81 long introns. Of the long introns, 55.6% include either a zipper stem or downstream stem in classes 1–3. Class 4 includes 13 structured long introns that include other structures with high helix confidence and long stems (red in Fig. 5). By contrast, the next class includes less structured long introns, which have neither high Gini coefficients nor

long stems (Class 5, green in Fig. 5). Class 6 includes most short introns, which tend to be depleted of stems, without zipper stems, downstream stems or high helix confidence estimates (orange in Fig. 5). Finally, class 7 includes structured short introns with at least one high-confidence stem (blue in Fig. 5).

**Fig. 6 | High-throughput structure–function assay for evaluating intron stem variants. a**, An overview of the structure-function experiment. $k_{tx}$, transcription rate; $k_{pre-mRNA}$, pre-mRNA decay rate; $k_{sp}$, splicing rate; $k_{mRNA\ decay}$, mRNA decay rate; $k_{lariat\ decay}$, intron lariat decay rate. **b**, Schematic for an example library design for one intron stem, with variants disrupting the stem and rescue sequences restoring base pairing. **c–e**, The effects of structure variants on the RI fraction for two regions of the *RPL28* intron (**c,d**) and for intron stems in *RPS9A* (**e**). For a given stem set or loop, violin plots depict data for the wild-type sequence and all variant sequences; unique barcodes are shown as black points. Data for rescue sequences are shown when included in the intron library. *P* values are indicated for comparisons between the wild-type and variant sequence sets, and between the variant and rescue sequence sets. *P* values were computed using two-sided permutation tests for the difference in mean statistic. In box plots, the median is the center white point, box limits are the 25th (Q1) and 75th (Q3) percentiles and whiskers extend to the smallest and largest values that fall within 1.5 times the interquartile range below Q1 and above Q3. Secondary structures are colored according to reactivity data. Bars alongside the secondary structure indicate the stem and loop disruption sets, with each bar representing a set of variant

sequences mutating nucleotides across the full extent of the bar. These bars are colored by the RI score for the corresponding stem or loop disruption set. The RI score is computed as the negative log (*P* value) when comparing RI values between wild-type and variant sequences; the sign is used to indicate the effect direction, with positive values (shown as blue) for a lower variant RI fraction compared with the wild type, and negative values (shown as red) for a higher variant RI fraction compared with the wild type. Green boxes in **e** indicate significantly covarying residues. **f**, RI fractions as measured by RT–qPCR for individual strains containing a set of wild-type, variant and rescue sequences for *RPS9A* stem 191–195 (top), with data shown for strains constructed with two different barcode sequences from three biological replicates (bottom). Data are presented as median values with 95% confidence intervals. *P* values are computed with two-way ANOVA tests with multiple comparisons. Exact *P* values for the low-RI-fraction barcode case are as follows: wild type versus 5′ mutant $P < 1 \times 10^{-4}$, wild type versus 3′ mutant $P < 1 \times 10^{-4}$, 5′ mutant versus rescue $P < 1 \times 10^{-4}$ and 3′ mutant versus rescue $P = 0.0003$. Exact *P* values for the high-RI-fraction barcode case are as follows: wild type versus 5′ mutant $P = 0.018$ and 5′ mutant versus rescue $P = 0.021$.

## A high-throughput assay to test the function of intron stems

To assess the influence of intron structures on gene expression, we developed a high-throughput assay for evaluating variants of RNA structure (VARS-seq). Here, we used VARS-seq to measure spliced and unspliced mRNA levels for intron structure variants, anticipating that these structures might influence splicing or pre-mRNA decay rates. Structures that slow pre-mRNA decay or splicing could lead to an accumulation of unspliced mRNA, whereas structures that increase splicing rates could increase spliced mRNA levels (Fig. 6a). We chose seven structured introns to assay with VARS-seq, including introns with zipper stems (in *RPL7A*, *QCR9*, *RPL36B* and *RPL28*), covarying base pairs (in *RPS9A*, *RPS9B* and *RPL7A*), and long stems that distinguished pre-mRNA from coding RNA.

For each intron, we designed variants with systematically mutated secondary structures and rescued them with compensatory substitutions where possible (Supplementary Table 3). More specifically, for each target set of stems, we chose variants that were predicted to disrupt base-pairing in the stem set while maintaining base-pairing elsewhere in the structure (Fig. 6b and Extended Data Fig. 8a,b). When feasible within library length limits (Methods), we also included rescue sequences that were predicted to restore the native secondary structure (Fig. 6b and Extended Data Fig. 8a,b). We tested 4–8 distinct intron variants for each target stem set, with around 200 variant sequences for each intron (Supplementary Table 3). We integrated these intron variant libraries into the yeast genome[48] in their native gene context

(Fig. 6a and Extended Data Fig. 8c), installing unique barcodes to help match spliced RNAs to their original pre-mRNA variant (Extended Data Fig. 8c,d and Supplementary Fig. 13). With multiple barcodes and variant sequences assigned to each set of stems, we could observe subtle effects on gene expression due to changes in intron structure (Supplementary Text). Using our RNA-sequencing data, we computed two key readouts for each variant: the RI fraction (fraction of total RNA that is unspliced) and the normalized mRNA level (spliced mRNA levels normalized by representation in genomic DNA libraries).

Across the 7 tested introns, we detected statistically significant changes in RI and mRNA levels for 52 of 136 tested sets of stems and 4 of 15 tested loops (Extended Data Fig. 9 and Supplementary Fig. 14). Some stem disruptions resulted in significantly decreased RI levels (Fig. 6c), whereas others significantly increased them (Fig. 6d). For many introns, distinct variant sequences designed to disrupt overlapping sets of stems produced similar effects, providing a useful cross-check for our approach (Supplementary Fig. 14a–d). For instance, variants disrupting all 5 sets of stems and loops that included nucleotides 36–48 in the *RPL28* intron significantly decreased the RI, suggesting that these nucleotides reduce splicing efficiency or slow pre-mRNA decay (Fig. 6c and Supplementary Fig. 14a). Similarly, all mutations to nucleotides 73–96 of the *RPL28* intron led to increased RI fractions (Fig. 6d), suggesting that the wild-type nucleotides promote splicing or pre-mRNA decay. Unexpectedly, zipper stem variants in the *RPL28* intron (Fig. 6c) lowered RI fraction, suggesting that not all zipper stems promote

**Fig. 7 | De novo secondary structure feature prediction for *S. cerevisiae* and the *Saccharomyces* genus. a**, Workflow for comparisons between introns' secondary structure ensembles and those of control sequences. **b**, Comparison of secondary structure feature enrichment between introns and control sequences for DMS-guided structure prediction (left; with folding engine RNAstructure, comparing introns to shifted genomic controls), de novo MFE structure prediction (middle; with folding engine RNAstructure, comparing introns with shifted genomic controls), and de novo ensemble structure prediction (right; with folding engine Vienna 2.0, comparing introns with shuffled sequence controls). *\**P* < 0.01 by two-sided Wilcoxon ranked-sum test. Exact *P* values from left to right are as follows for the left panel: $P < 1 \times 10^{-4}$, $P = 0.0003$, $P = 0.0007$, $P < 1 \times 10^{-4}$, $P < 1 \times 10^{-4}$; for the middle panel: $P < 1 \times 10^{-4}$, $P = 0.0072$, $P = 0.0008$, $P < 1 \times 10^{-4}$, $P = 0.0015$; and for the right panel: $P < 1 \times 10^{-4}$, $P = 0.0003$, $P = 0.0011$, $P < 1 \times 10^{-4}$, $P < 1 \times 10^{-4}$. In the left and middle panels, 140 introns are compared; 288 introns are compared in the right panel. MFE, minimum free-energy; SS, splice site; BP, branch point; intron · control score, difference between intron and control score. **c**, Differences in zipper stem (top) and downstream stem (bottom) ΔG between introns and shuffled sequence controls for introns in the *Saccharomyces* genus, using Vienna 2.0 ensemble predictions. *\**P* < 0.01 by two-sided Wilcoxon ranked-sum test. All *P* values for the zipper stem comparisons are $<1 \times 10^{-4}$. For each species, the number of introns compared for both stem types and the *P* value for the downstream stem comparison are as follows: smik ($n = 279$, $P = 0.00150$),

skud ($n = 279$, $P = 0.21$), suva ($n = 278$, $P = 0.07$), cgla ($n = 100$, $P < 1 \times 10^{-4}$), kafr ($n = 216$, $P = 0.83$), knag ($n = 175$, $P = 0.011$), ncas ($n = 250$, $P = 0.10$), ndai ($n = 218$, $P = 0.00056$), tbla ($n = 163$, $P = 0.0017$), tpha ($n = 143$, $P = 0.0005$), kpol ($n = 175$, $P < 1 \times 10^{-4}$), zrou ($n = 166$, $P = 0.28$), tdel ($n = 202$, $P = 0.58$), klac ($n = 151$, $P = 0.0026$), agos ($n = 185$, $P = 0.26$), ecym ($n = 19$, $P = 0.41$), sklu ($n = 229$, $P = 0.27$), kthe ($n = 215$, $P = 0.31$) and kwal ($n = 210$, $P = 0.011$). **d**, Distribution of zipper stems across introns in the *Saccharomyces* genus. Green values on the heatmap indicate a predicted zipper stem; white indicates no predicted zipper stem; gray values indicate deleted introns. Ohnologous introns are combined into a single row, and a zipper stem is annotated if present in either homolog. The species represented in this figure are: *Eremothecium gossypii* (agos), *Candida glabrata* (cgla), *Eremothecium cymbalariae* (ecym), *Kazachstania africana* (kafr), *Kluyveromyces lactis* (klac), *Kazachstania naganishii* (knag), *Vanderwaltozyma polyspora* (kpol), *Lachancea thermotolerans* (kthe), *Lachancea waltii* (kwal), *Naumovozyma castellii* (ncas), *Naumovozyma dairenensis* (ndai), *Saccharomyces kudriavzevii* (skud), *Saccharomyces mikatae* (smik), *Saccharomyces uvarum* (suva), *Tetrapisispora blattae* (tbla), *Torulaspora delbrueckii* (tdel), *Torulaspora phaffii* (tpha) and *Zygosaccharomyces rouxii* (zrou). In box plots, the median is the center white point, box limits are the 25th (Q1) and 75th (Q3) percentiles, and whiskers extend to the smallest and largest values that fall within 1.5 times the interquartile range below Q1 and above Q3.

splicing by colocalizing splice sites[16]. In the context of secondary structures from DMS-MaPseq, these results demonstrate that intronic mutations can influence gene expression, even when they are structurally distant from splice sites (Fig. 6d and Supplementary Fig. 14).

In cases in which we could generate compensatory mutations, stem disruption and rescue variants pinpointed functional intronic structural elements. For instance, in the case of RPL36B, sequence variants in one region reduced normalized mRNA levels and rescue variants restored higher mRNA levels (Supplementary Fig. 15a), implicating the RNA structure, rather than its primary sequence, as the functional element. In the case of the intron in *RPS9A*, we identified a functional stem in which disruptions reduced the RI fraction, which was restored by stem rescue (stem 191–195, Fig. 6e). An analogous stem is present in the *RPS9B* intron, with similar effects (Stems 165–183, Supplementary

Fig. 14b). These *RPS9A* and *RPS9B* stems likely influence gene expression by inhibiting splicing or pre-mRNA decay, aligning with covariation in these stems (Fig. 2e,f) and corroborating a prior study[3] on *RPS9A*.

To validate our findings from VARS-seq, we constructed strains containing individual variant and rescue sequences for intron stems from *RPL36B* and *RPS9A*, and we used quantitative reverse transcription PCR (RT–qPCR) to measure mRNA levels. In the case of *RPS9A*, RT–qPCR data from individual strains recapitulated structure effects from VARS-seq along with the effects of barcode sequences (Fig. 6e,f). Indeed, when assessing a barcode that yielded low wild-type RI fractions (left, Fig. 6f), variant sequences lowered the RI, and rescue sequences restored it to wild-type levels, as determined by RT–qPCR. By contrast, individual strains designed to assess intron stems from *RPL36B* (Supplementary Fig. 15) did not show significantly different normalized

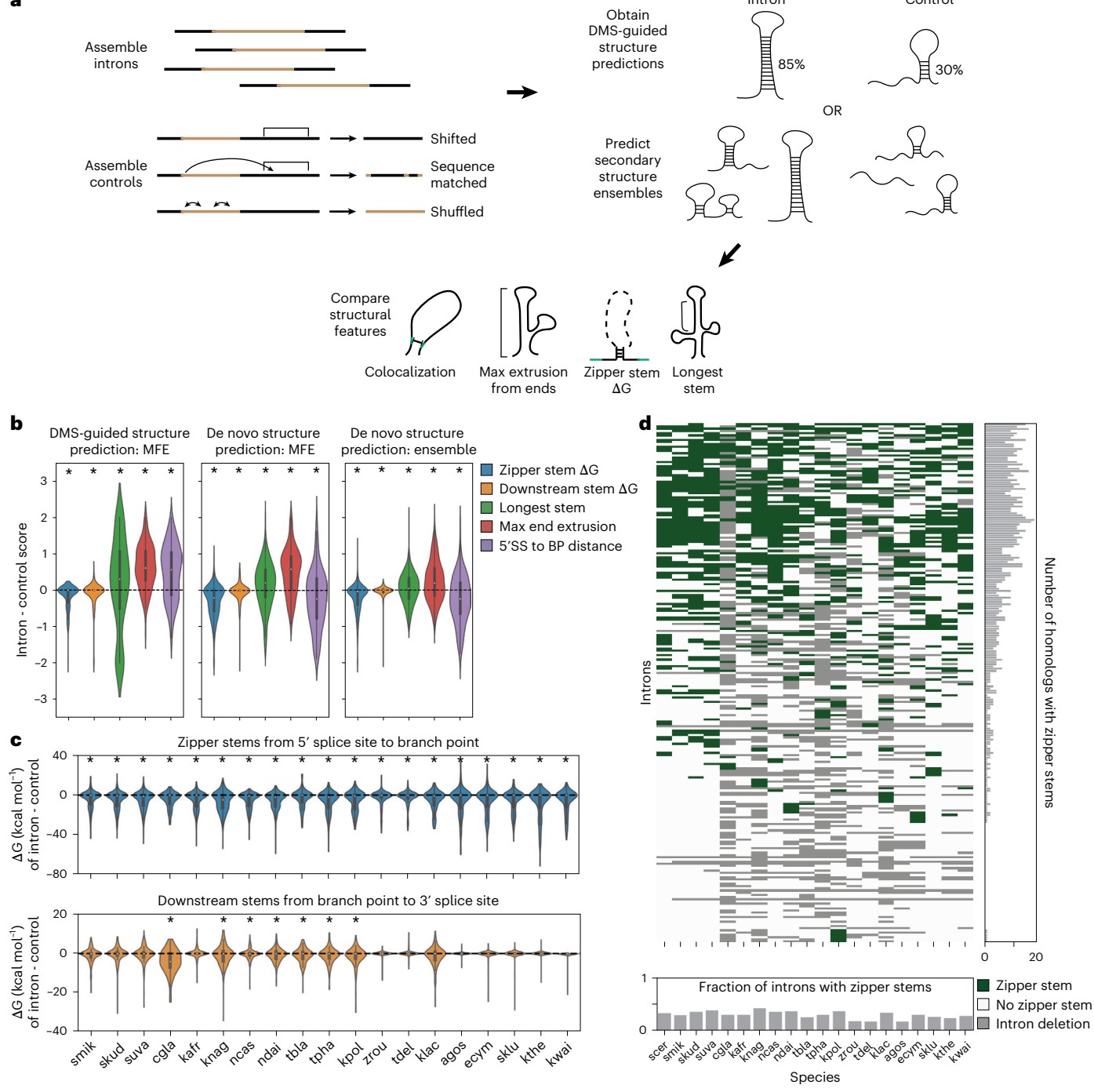

mRNA levels as determined by RT–qPCR (Supplementary Fig. 15b,c). It is possible that effects found by aggregating data across variant and barcode sequences through VARS-seq could not be discerned when analyzing individual *RPL36B* variants in RT–qPCR. Owing to differences in dynamic range between VARS-seq and RT–qPCR, effects for higher RI barcodes on the intron in *RPS9A* are not visible with RT–qPCR (right, Fig. 6f), and significant zipper stem effects from RT–qPCR were not visible with VARS-seq for *RPL36B* (Supplementary Fig. 15b,c). To enable further analysis of intron stem variants, we summarize all variant and rescue comparisons from VARS-seq in heatmaps in Supplementary Figure 16, and we include per-barcode spliced and unspliced RNA counts in Supplementary Table 4.

### Enriched structural patterns across the *Saccharomyces* genus

Our transcriptome-wide structure mapping provides evidence for widespread structure in *S. cerevisiae* introns, and with VARS-seq, we find that intron structures can impact gene expression regulation. To extend our observations to other species in the *Saccharomyces* genus, we turned to computational structure prediction. We first evaluated whether de novo secondary structure prediction could recapitulate structural features observed through DMS-guided analysis, comparing introns with length-matched controls (Fig. 7a and Supplementary Fig. 17). For de novo structure prediction, as a first approach, we predicted minimum free-energy structures[33]; as a second approach, we generated secondary structure ensembles through stochastic sampling of structures[49], which has previously enabled the study of structural patterns in introns[16,19]. With either of these approaches, structural features from de novo secondary structure prediction and DMS-guided structures were largely consistent (Fig. 7b, Supplementary Fig. 18 and Supplementary Text).

We next made de novo structure predictions across species in the *Saccharomyces* genus, focusing on 20 species with intron alignments from Hooks et al.[50]. Enrichment for numerous secondary structure patterns is conserved in introns across the *Saccharomyces* genus. In particular, introns across the genus include more stable zipper stems and downstream stems than do controls (Fig. 7c), along with higher MEE and shorter distances between the 5′ splice site and branch point (Supplementary Fig. 19a). Furthermore, many of these features remain enriched in comparison with phylogenetic controls, which include random sequences constructed to match the mutation and indel frequency between an intron and its homologous *S. cerevisiae* intron (Supplementary Fig. 19b).

Secondary structure patterns are maintained across *Saccharomyces* species despite extreme sequence-level divergence between introns, suggesting that these structures have conserved functions. Complete intron deletions between these species are common, with many introns having orthologs in only a subset of the *Saccharomyces* species (Supplementary Fig. 19c), and most intron sequences have low conservation (50 to 60%) between species (Supplementary Fig. 19d). Some key functional intervals in introns are more conserved across the *Saccharomyces* species, with higher sequence conservation (72.3% to 83.6%) across the seven small nucleolar RNA (snoRNAs) in these intron sequence alignments. By contrast, zipper stem regions in *S. cerevisiae* introns diverge substantially between *Saccharomyces* species; most zipper stems are only 20 to 30% conserved in the primary sequence. Strikingly, this structural motif appears in disparate introns across species, with many zipper stems present in only a small number of orthologous introns (Fig. 7d). It is possible that functional secondary structures in these regions have been challenging to find by covariation due to high intron sequence divergence in closely related fungal genomes.

## Discussion

RNA structures have critical roles in regulating a wide array of nuclear processes, including transcription, RNA modification and editing and splicing. Intronic RNA structures are poised to participate in these processes, for instance by altering splicing kinetics, changing RNA decay rates, or interacting with other nuclear factors. Here, we sought to understand the structural landscape of introns in *S. cerevisiae*, a model system for eukaryotic splicing. First, we evaluated the presence of structural patterns in introns using transcriptome-wide DMS probing after splicing inhibition. This revealed extended secondary structures in introns that distinguished these regions from coding mRNA. These structural data enable the clustering of introns into seven classes, with most falling into classes that contain introns with zipper stems or downstream stems, long introns with intermediate structure, and unstructured short introns. In Extended Data Figure 10 and Supplementary Figure 20, we display secondary structures and reactivity profiles for all introns from DMS-MaPseq, grouped into these classes. With a high-throughput structure-function assay, VARS-seq, we used deep mutagenesis to identify intron structural elements that influence spliced and unspliced mRNA levels. Finally, through computational structure prediction, we identified signals for structure in introns across the *Saccharomyces* genus.

Structure probing experiments have enabled the assessment of RNA structure across the transcriptome, but these experiments often lack sufficient coverage to provide information for low-abundance transcripts, including many unspliced pre-mRNA molecules. Specific low-abundance introns can be probed by target-specific enrichment[27,51] or nuclear RNA enrichment[45,52], but here we enhance detection of pre-mRNA sequences generally by using global splicing inhibition. We expect that most pre-mRNAs in pladB-treated cells remain unspliced, as accumulating pre-mRNAs are expected to outnumber spliceosome components[53]. It is possible that splicing inhibition could alter pre-mRNA structure compared with that in untreated cells, as accumulating pre-mRNA could interact with the nuclear pore complex[54], nuclear exosome[55] or NMD machinery[56]. Nevertheless, for introns with sufficient coverage from untreated cells, we found similar high-confidence stems with and without splicing inhibition. In future work, it would be interesting to directly observe long-range base-pairing and higher-order RNA structures in introns using approaches such as PARIS[57] or KARR-seq[58] in pladB-treated cells.

Our data allowed us to discern structural patterns that distinguish introns from coding regions. For instance, with higher Gini coefficients and longer stems, stable structures are enriched in introns compared with coding regions. It is possible that introns' intramolecular structures are necessary to avoid spurious interactions between intron RNA and other nucleic acids in the crowded nuclear environment, preventing dysregulation of gene expression. Additionally, although there is evidence across species for depletion of double-stranded RNA in coding RNA to avoid activating cytosolic antiviral cellular responses[59], this selective force would not affect introns in the nucleus. Long stems could also be depleted from coding mRNA because cytosolic mRNAs with stable stems might accumulate stalled ribosomes, becoming subjected to decay pathways such as no-go decay[31,60]. Finally, without the evolutionary constraints of adhering to a coding sequence, introns might be free to form extended structures that can regulate a host of nuclear processes.

Given that structural motifs are enriched in *S. cerevisiae* introns, we hypothesized that these introns could harbor functional structures. Although bioinformatic scans for covariation have identified functional structures in a several *S. cerevisiae* introns (including *RPS9A*[7], *RPS9B*, *RPS13* (ref. 23), *RPL7B*[23] and *RPL7A*), sequence-only analyses could miss functional structures owing to the limited power of existing intron alignments[61], with high variation between aligned intron sequences. Indeed, tools such as R-scape can be limited by low sequence conservation in alignments[42,61]. By using the high-throughput VARS-seq analysis to evaluate intron variants and rescue sequences, we identified additional functional structures. Although prior intron variant libraries have been designed to assess the effects of sequence motifs[62,63], libraries assessing the role for structured elements in introns have

been limited. Using VARS-seq, we found that some introns included domains that could influence gene expression despite being distal from splice sites in the intron's secondary structure. These examples add to the complexity of potential functional roles for introns[9,10]. Additional evaluation of individual strains using orthogonal assays could validate the effects for specific stems of interest, which we carried out for intron stem sets from *RPS9A* and *RPL36B*.

Together, our structural, computational and functional experiments point to structural patterns across introns that impact gene expression. In one common pattern, we found an enrichment for structures around cryptic splice sites and depletion of structures around canonical splice sites, potentially indicating a role for structures in encouraging the use of canonical sites. Another pattern was the formation of zipper stems that colocalize the 5' splice site and the branch point. Introns included highly stable zipper stem structures in vivo, and these structures were further confirmed by multi-dimensional chemical mapping (M2-seq) in vitro. These zipper stems could enable efficient splicing by reducing the physical distance between the 5' splice site and branch point[64] or by enhancing specific interaction with the spliceosome, with intron helical density seen interacting directly with the E and pre-A spliceosome complexes[25,36].

To investigate the mechanisms by which these and other structured elements regulate gene expression, customized experiments for individual introns are needed. Structures located close to splice sites could influence spliceosome recruitment, while structures farther away from splice sites might interact with other nuclear factors. Intron secondary structures can regulate splicing by sequestering alternative splice sites[65,66], occluding exonic splicing enhancers[67], physically bridging splice sites[68], facilitating co-transcriptional splicing[69] or mediating protein interactions that influence splicing patterns[70]. Additionally, these intronic structures could have regulatory functions in pathways orthogonal to splicing, much like the snoRNAs encoded in *S. cerevisiae* introns[41]. In fact, RNA secondary structures in introns are associated with RNA-binding proteins involved in transcription, tRNA and rRNA processing, ribosome biogenesis and assembly and metabolic processes[71,72]. Furthermore, secondary structures in introns could influence gene expression in auto-regulatory circuits, as seen previously in the cases of RPS14B[21] or RPS9A[7], with structural elements within a gene's introns binding its protein products and thereby downregulating subsequent gene expression. Intron structures could also influence numerous pre-mRNA decay pathways, including nuclear retention followed by decay by the nuclear exosome, NMD and NMD-independent decay by cytoplasmic exonucleases[31,32,54,55,73]. Finally, structures can have a functional role in regulating adaptation to starvation by influencing the accumulation of introns under nutrient depletion[5,6], and it will be interesting to explore these structures in these saturated-growth conditions.

Our work identifies a set of intron structures with properties distinct from those in coding regions, support for in vivo formation from DMS-MaPseq and signals in related yeast species. Furthermore, we identify structured intervals of introns that modulate gene expression. These functional experiments provide candidates for further mechanistic characterization and provide a glimpse into the broad regulatory potential for intron sequences beyond splice sites. The widespread presence of structured elements in *S. cerevisiae* introns raises the possibility that similar motifs and stable secondary structures play a role in introns in higher-order eukaryotes, perhaps forming regulatory elements in human pre-mRNA.

## Online content

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

## Methods

### Strains, medium and growth conditions for DMS probing

The strain OHY001 was constructed from JRY8012 (ref. [75]), which includes three deletions of ABC transporter genes (prd5::kan[r], snq2::kan[r], yor1::kan[r]) to reduce drug efflux. OHY001 was generated through CRISPR editing of JRY8012 to mutate portions of HSH155 HEAT repeat domains 15–16 to match the sequence found in human SF3B1 (see sequence in Supplementary Table 5).

Strains were grown at 30 °C on YPD plates and in YPD liquid medium. Single colonies of OHY001 were used to inoculate overnight cultures and grown to an optical density at 600 nm ($OD_{600}$) of 0.5–0.6. Biological replicates were obtained from distinct single colonies.

### Splicing inhibition and DMS treatment

We carried out splicing inhibition and DMS treatment for two biological replicates, only splicing inhibition for a no-modification control and only DMS treatment for another control. We cultured 15–60 ml to an $OD_{600}$ of ~0.5 for each replicate of each condition (2 biological replicates of DMS treatment, 1 no-modification control, 1 no-pladB control and 2 biological replicates of in vitro DMS treatment). We treated cultures with 5 μM pladB (Cayman Chemicals), and we incubated cultures at 30 °C for 1 h with shaking. The condition without splicing inhibition was treated with an equal volume of DMSO.

For in vivo DMS modification, we treated cultures with 3% DMS or an equivalent volume of $H_2O$ for the no-modification control. Treated cells were incubated with occasional stirring in a water bath at 30 °C for 5 min, and the reaction was quenched by adding 20 ml stop solution (30% 2-mercaptoethanol, 50% isoamyl alcohol) for every 10 ml of culture. Cultures were mixed and then spun down for 3 min at 1,500g and 4 °C, and washed first with 5 ml wash solution (30% 2-mercaptoethanol) for every 10 ml of culture. A second wash was performed with 3 ml YPD per 10 ml culture. RNA was extracted using the YeaStar RNA Kit (Zymo Research), using 7.5 μl of Zymolase for every 2.5 ml of cell culture and shortening the Zymolase incubation to 15 min at 30 °C.

For in vitro DMS modification, we followed a protocol similar to the one used in Rouskin et al.[76]. We first obtained RNA from cultures after splicing inhibition with the YeaStar RNA Kit (Zymo Research). We then re-folded the RNA in vitro, first denaturing 200 μg of RNA at 95 °C for 2 min, cooling it on ice for 2 min, and then folding it at 30 °C for 30 min in 10 mM Tris HCl pH 8.0, 100 mM NaCl and 6 mM $MgCl_2$. We treated RNA with 3% DMS at 30 °C for 5 min, and we quenched the reaction with 25% 2-mercaptoethanol. RNA was purified by ethanol precipitation and eluted in 20 μl RNase-free $H_2O$.

### RT–PCR for verifying splicing inhibition

As initial verification of splicing inhibition by pladB, we used RT–PCR to compare unmodified RNA extracted after 1 h of either 5 μM pladB or DMSO treatment. We first treated samples with TURBO DNase (Thermo Fisher). We then carried out reverse transcription with the iScript Reverse Transcription Supermix (Bio-Rad), using 10 μl of each RNA sample for a reaction that included the reverse transcriptase, and the remaining 10 μl for a control without the enzyme. PCR was then performed for 30 cycles at an annealing temperature of 56 °C using the NEBNext Ultra II Q5 Master Mix (New England Biolabs (NEB)) with primers RR063 and RR064 for RPL36B, and RR067 and RR070 for MATa1 (Supplementary Table 5).

### DMS-MaPseq sequencing library preparation

To prepare DMS-treated RNA for sequencing, we first depleted the extracted RNA of rRNA using RNase H. We concentrated extracted RNA for each condition using RNA Clean and Concentrator-5 columns (Zymo Research), yielding between 46.1 and 200 μg RNA for each replicate of each condition. We pooled and concentrated 108 fifty-base oligonucleotides (RR-rRNAdep-1-108; Supplementary Table 5) that tiled the 5S, 5.8S, 18S and 25S rRNAs in S. cerevisiae. Up to 40 μg of

total RNA was included in each rRNA depletion reaction, and multiple reactions were performed as needed for each sample. First, a 15-μl annealing reaction was prepared with total RNA, an equal mass of rRNA depletion oligonucleotides, and 3 μl of 5× hybridization buffer (500 mM Tris-HCl pH 7.5 and 1 M NaCl). For the annealing reaction, the reaction mix was heated to 95 °C for 2 min, and the temperature was ramped down to 45 °C by 0.1 °C s$^{-1}$ with a thermocycler. Then, 7.5 μl of Hybridase Thermostable RNase H (Lucigen) with 2.5 μl of 10× digestion buffer (500 mM Tris-HCl pH 7.5, 1 M NaCl, 100 mM $MgCl_2$) was preheated to 45 °C. We combed the annealing mix and the RNase H mix and incubated the reaction at 45 °C for 30 min. For each reaction, we used an RNA Clean and Concentrator-5 column with the size-selection protocol to exclude RNA and oligonucleotides below a size cutoff of 200 nt. We then treated each reaction with TURBO DNase. Reactions were purified and then eluted in 9 μl of RNase-free $H_2O$.

We fragmented each reaction with 10X RNA Fragmentation Reagent (Ambion), incubating at 70 °C for 8 min. We added 1 μl of Stop Solution (Ambion), cleaned up reactions using RNA Clean and a Concentrator-5 column, and removed the 3′ phosphate groups left by fragmentation with 1.5 μl rSAP (NEB). We next ligated a universal cloning linker to the RNA to serve as a handle for reverse transcription. To prepare linker for this reaction, we first phosphorylated 1 nmol of the DNA universal cloning linker with a 3′ amino blocking group (oligo RR118; Supplementary Table 5) with T4 PNK (NEB), and we next adenylated the linker with Mth RNA Ligase (NEB). We then purified the reaction with an Oligo Clean and Concentrator column. We added adenylated linker in twofold molar excess to each RNA sample, with ligation using T4 RNA ligase 2 truncated KQ (NEB). The reaction was incubated at 25 °C for 2 h in a thermocycler. Ligated RNA was purified using an RNA Clean and Concentrator-5 column, followed by elution in 15 μl of RNase-free $H_2O$. The excess DNA linker was then degraded with 1 μl of 5′ deadenylase (NEB) and 1 μl of RecJf (NEB) in a 20-μl reaction. The mixture was incubated at 30 °C for 1 h and purified with an RNA Clean and Concentrator-5 column.

We then proceeded to RT with mutational readthrough. For the RT primer, we used the oligonucleotide RR114 (Supplementary Table 5), which included a sequence complementary to the universal cloning linker, a 5′ phosphate modification that would allow for circularization after RT, a 10-nt randomized unique molecular identifier (UMI) sequence, and sequences complementary to Illumina sequencing primers to allow for PCR amplification of the final library. We added 2 μl of 5× TGIRT buffer (250 mM Tris-HCl pH 8.0, 375 mM KCl, 15 mM $MgCl_2$), 0.5 μl of 2 μM RT primer, and 6 μl of the RNA sample. The reaction was denatured at 80 °C for 2 min and left at room temperature for 5 min. Then, 0.5 μl TGIRT enzyme (InGex), 0.5 μl SUPERase inhibitor, 0.5 μl 100 mM DTT, and 1 μl 10 mM dNTPs were added to the reaction. The reaction was incubated at 57 °C for 1.5 h in a thermocycler for RT. RNA was then degraded by adding 5 μl of 0.4 M NaOH for 3 min at 90 °C, and the reactions were neutralized by adding 5 μl of an acid quench mix (from a stock solution of 2 ml of 5 M NaCl, 2 ml of 2 M HCl and 3 ml of 3 M sodium acetate). The reaction was purified using an Oligo Clean and Concentrator column, with elution in 7.5 μl of RNase-free $H_2O$. The cDNA was then purified with a denaturing PAGE (dPAGE) gel to remove excess RT primer, selecting the 200–400-nt size range. Purified cDNA was eluted using the ZR small-RNA PAGE Recovery Kit (Zymo Research).

The size-selected cDNA was circularized and then amplified for sequencing. For circularization, we used CircLigase ssDNA Ligase (Lucigen), with overnight incubation at 60 °C followed by 10 min at 80 °C. The circularized cDNA was purified with an Oligo Clean and Concentrator column and eluted in 7.5 μl of RNase-free $H_2O$. Residual RT primer was removed by degrading linear DNA, first with treatment with RecJf, followed by ExoCIP A and ExoCIP B (NEB) treatment. The circularized cDNA was purified and eluted in 10 μl. We then carried out 9 cycles of indexing PCR to add i5 and i7 index sequences to the sequencing library, using the NEBNext Ultra II Q5 Master Mix (NEB)

with 10 µl of cDNA and 2.5 µl of 10 µM primers including different index sequences for each sample (i5_4 and i7_4 for replicate 1, i5_3 and i7_3 for replicate 2, i5_5 and i7_5 for the no-modification control, and i5_2 and i7_2 for the -pladB control; Supplementary Table 5). Nine PCR cycles were carried out, with annealing at 70 °C for 30 s. The reaction mix was purified with the DNA Clean and Concentrator-5 kit (Zymo Research), with elution in 10 µl of H₂O. To obtain sufficient material, we then carried out two more cycles of PCR for in vivo replicates 1 and 2 and the no-modification control, five more cycles for the no-pladB control, nine more cycles for in vitro replicate 1, and three more cycles for in vitro replicate 2. For the final PCR reaction, we used the same reaction conditions and cycling parameters as the first indexing PCR with primers P5 and P7 (Supplementary Table 5). We purified the final library with two rounds of bead purification with size selection to remove remaining excess RT primer. We used RNACleanXP beads (Beckman Coulter), mixing 42.5 µl beads with the 50 µl PCR reaction. For the first round of bead-based size selection, we performed elution in 50 µl of H₂O, and for the second round, we performed elution in 20 µl of H₂O. The sequencing libraries were quantified with a Qubit high sensitivity dsDNA Assay Kit (Invitrogen) and Bioanalyzer HS DNA assay. The dsDNA libraries for in vivo replicate 1, the no-modification control, and the no-pladB control were sequenced across three Illumina HiSeq lanes with paired-end reads of length 150. The dsDNA library for in vivo replicate 2 and the in vitro replicates were sequenced with NovaSeq S4 partial lanes; these also had a paired-end read length of 150.

## DMS-MaPseq sequencing data analysis

We obtained 557 million reads for in vivo replicate 1, 1.30 billion reads for in vivo replicate 2, 375 million reads for the no-modification control, 169 million reads for the control without pladB, 335 million reads for in vitro replicate 1, and 174 million reads for in vitro replicate 2. We used UMI-tools[77] to extract UMI tags from reads, and we used cutadapt to trim low-quality reads (Q-score cut-off of 20) and remove adapters. We then aligned sequencing reads to sequence sets of interest, including rRNA, introns, pre-mRNA ORFs, coding mRNA, decoy intron sequences (see below), and sequences for structured controls (the ASH1 and HAC1 mRNA sequences). *S. cerevisiae* intron annotations, including genomic coordinates and branch point positions, were obtained from Talkish et al.[78]. Coding ORF annotations were obtained from the Saccharomyces Genome Database[79] for the S288C reference genome, and coding sequences corresponding to introns were identified for all cases except the two introns in snoRNAs (SNR17A and SNR17B). Paired-end alignment was performed with Bowtie2 (ref. [80]) using the following alignment parameters in ShapeMapper 2 (ref. [81]): –local –sensitive-local –maxins=800 –ignore-quals –no-unal –mp3,1 –rdg5,1 –rfg 5,1 –dpad 30. Alignments were merged, sorted, and indexed using samtools[82]. Reads with matching UMI tags were then deduplicated using the UMI-tools dedup function.

Mutational frequencies, coverage values and normalized reactivities were obtained by processing alignments using RNAframework[83] executables. We first ran rf-count with the flag -m to compute mutation counts, using all other default parameters. We obtained coverage statistics for each sequence with rf-rctools stats, per-position mutation counts and coverage with rf-rctools view, and reactivity values with rf-norm (flags: -sm 2 -nm 2 -ow -rb AC -dw).

## DMS-MaPseq data quality assessment

For each construct, per-base coverage was computed as the number of reads obtained from rf-rctools stats multiplied by the total read length and divided by the construct length. To compare the retained intron fraction across all introns before and after pladB treatment, these coverage values were obtained for all introns and all coding regions for genes containing introns in the conditions without and with pladB. The retained intron fraction was the ratio of these values. We removed introns in *GCR1* from further consideration as the gene

includes multiple distal alternative 5′ splice sites[84]. We combined data from two introns in *SRC1* from nearby alternative 5′ splice sites differing by 4 nt. We additionally consolidated data from two-intron genes with multiple annotated isoforms (in *RPL7B*, *VMA9*, *DYN2* and *SUS1*), in which annotated introns included both single individual introns and the longer intron representing the skipped isoform[85]. We used annotations for the ribosomal protein-coding genes from Hooks et al.[50]. We excluded eight snoRNA-containing introns from further analysis[41], as the majority of these constructs' reactivities represented the excised snoRNA structure rather than the complete intron structure.

We next found the Pearson correlation between intron reactivities for replicate 1 and replicate 2. Reproducibility between replicates is reported as the square of the Pearson correlation coefficient ($r^2$) through the manuscript. A linear fit for log coverage versus replicate correlation for each intron indicated that the replicate correlation $r$ was best approximated as 0.2 log (coverage) – 1.03. On the basis of this relationship, to reach $r^2 = 0.6$, we used a coverage cut-off of 7,673, averaged between replicate 1 and replicate 2 in subsequent analyses.

To obtain the correlation between rRNA mutational frequencies in Zubradt et al.[27] and our data, we obtained the paper's DMS-MaPseq reads for *S. cerevisiae* with TGIRT reverse transcriptase (replicate 1 accession number SRX1959209, run number SRR3929621). We aligned these reads to the 18S and 25S yeast rRNA sequences with Tophat v2.1.0 (ref. [86]) using the alignment parameters stated in Zubradt et al.[27]: -N 5 –read-gap-length 7 –read-edit-dist 7 –max-insertion-length 5 –max-deletion-length 5 -g 3. We obtained mutational frequencies with rf-count, as described above, and we compared them with our dataset.

We evaluated whether reactivity values could differentiate surface-accessible, unpaired residues from base-paired residues across the 18S and 25S rRNA. To identify surface-accessible residues in rRNA, we followed a similar protocol to that in Rouskin et al.[76]. With the *S. cerevisiae* ribosome structure (PDB ID 4V88 ref. [87]), we computed solvent-accessible surface area (SASA) values for the rRNAs' N1 atoms on A residues and N3 atoms on C residues in PyMOL, approximating DMS as a sphere with solvent_radius 3 and with dot_solvent and dot_density parameters set to 1. We determined the receiver operating characteristic curve for distinguishing unpaired and solvent-accessible rRNA residues from Watson–Crick base-paired positions (found with DSSR[88]), using a SASA cut-off of 2 to determine solvent accessibility.

## DMS-guided structure prediction and validation

For introns with between-replicate $r^2 > 0.6$, we performed secondary structure prediction by RNAstructure[33] guided by DMS reactivity with 1,000 bootstrapping iterations, using the package Biers[46] with default parameters for RNAstructure to obtain minimum free-energy structures and base-pair confidence matrices from bootstrapping. Structures were visualized with VARNA, Biers and RiboDraw (https://github.com/ribokit/RiboDraw).

To assess DMS-guided structure prediction, we performed structure prediction for a set of controls with known secondary structures. We obtained ground-truth secondary structures for the 5S, 5.8S, and 18S rRNA with DSSR[88] from a eukaryotic ribosome structure (PDB ID: 4V88)[87]; for U5 snRNA from Nguyen et. al.[89]; for U1 snRNA from Li et. al.[90]; for tRNA from Rfam-derived secondary structures[91]; and for mRNA segments from Zubradt et al.[27]. We did not include control RNAs with known pseudoknots (for example, RNase P RNA[92]) because our structure predictions did not include pseudoknots. Additionally, we excluded control cases with multiple conformations, such as the U2 snRNA[93]. For DMS-guided structure prediction, 1,000 bootstrapping iterations were performed for all cases except the 18S rRNA, for which we performed 100 bootstrapping iterations. Positive stem predictions were cases in which predicted stems included at least 5 bp above the helix confidence estimate threshold. False positive predictions occurred when less than 50% of the predicted stem's base pairs were included in the ground-truth structure. False negative predictions

included stems of at least 5 bp in length in the native structure that had either low confidence or fewer than 50% of their base pairs correctly predicted.

We additionally explored other methods for structure prediction. First, using Arnie (https://github.com/DasLab/arnie) we ran ShapeKnots[94] (allowing for pseudoknot prediction) on all 88 introns with sufficient coverage, guiding predictions with DMS data in 100 bootstrapping iterations. As a control, we predicted the RNase P RNA structure with ShapeKnots, comparing to the ground-truth secondary structure from PDB ID 6AGB ref. 92. We also generated predictions using DREEM[95] for the following intron regions with high coverage from DMS-MaPseq (numbered from intron start): *RPL28* 75–300, *RPL7A* 120–340 (first intron), *ECM33* 35–270, *RPL26B* 300–400, *RPS13* 140–212, *RPL25* 148–255, *RPS9B* 147–266 and *RPL30* 25–165.

### Targeted DMS probing of RNA in *S. cerevisiae* cells and heat-denatured RNA

To evaluate structures identified from transcriptome-wide DMS-MaPseq, we carried out targeted DMS probing for 30 introns that contained zipper stems (listed in Supplementary Table 6). For each of two biological replicates of targeted DMS probing, 5 ml of culture at an $OD_{600}$ of ~0.5 were treated with 3% DMS, and RNA was extracted as described above. We additionally obtained DMS profiles for heat-denatured RNA. More specifically, for two biological replicates, we denatured 20 µg of RNA extracted from *S. cerevisiae*, denaturing RNA in a 500 µl volume with 1 mM EDTA by heating at 90 °C for 3 min. For these denatured RNA samples, we then incubated with 1.5% DMS for 1 min at room temperature, quenched the reactions with 500 µl 2-mercaptoethanol and purified RNA with RNA Clean and Concentrator-5 (Zymo Research). We depleted rRNA from all targeted probing samples with RNase H, as described above.

We then prepared sequencing libraries from the DMS-treated RNA samples, using targeted primers for RT and PCR to specifically assess the structure of our 30 target introns. We designed primers to amplify 220–270-nt intron segments, with overlapping intervals covering the full length of the intron (primers RR400–RR523 in Supplementary Table 5). Primers were designed to overlap exon and intron boundaries to ensure that unspliced pre-mRNA was specifically amplified. These primers were pooled for multiplexed PCR into four pools with Primer Pooler[96] (setting the $Mg^{2+}$ concentration to 0 mM, the Na concentration to 50 mM and the dNTP concentration to 0.8 mM), with the aim to minimize the formation primer dimers and other incorrect amplicons. Primers in Supplementary Table 5 are annotated with the resulting pool number.

DMS-treated RNA was reverse transcribed with each of the four RT primer pools. Each RT reaction was conducted with Induro reverse transcriptase (NEB) using 2 µl of 10 µM RT primer pool with 1 µg RNA in a 20-µl reaction. Reactions were incubated at 55 °C for 1 h for RT, and NaOH was used to degrade RNA, as described above. Purified cDNA was first amplified in a pooled format with a separate reaction for each primer pool, combining 20 µl of 10 µM PCR primer pool with 5 µl cDNA template and 25 µl Q5 master mix for an 11-cycle PCR reaction (annealing temperature 61 °C). DNA was purified with the DNA Clean and Concentrator-5 kit (Zymo Research). We next added adapters for Illumina sequencing, using primers RR524–RR647 (Supplementary Table 5) in individual PCR reactions with 15 cycles at an annealing temperature of 64 °C. Finally, i5 and i7 primers were added with 8 cycles of PCR with annealing temperature 70 °C (primers RR281, RR282 for targeted DMS probing replicate 1; RR283, RR284 for targeted DMS probing replicate 2; RR285, RR286 for denatured RNA probing replicate 1; and RR287, RR288 for denatured RNA replicate 2). To select amplicons with final library size between 350 and 400 bp, we used AMPure beads. Specifically, we used 28.5 µl of beads for a 50 µl sample volume to remove large fragments, and then added 13 µl of beads to the supernatant to capture the correct amplicon size range. The resulting

library was quantified with Qubit and Bioanalyzer, and sequenced as a part of a NovaSeq X+ lane with a 2×150 cycle kit.

Data were processed using a similar workflow as described above for DMS-MaPseq. Targeted primers were removed using cutadapt, and primer-binding regions at the 5′ and 3′ end of each intron were masked during DMS-guided structure prediction. We obtained coverage of at least 100,000 for each of the 30 tested introns with targeted DMS probing.

### Two-dimensional chemical mapping (M2-seq)

We carried out two-dimensional chemical probing to assess the formation of base pairs from DMS-MaPseq. DNA sequences for the introns from *QCR9*, *RPL36B*, *RPS11A*, *RPL37A* and *RPS7B* were obtained as gene fragments from Twist Biosciences (Supplementary Table 5). Each gene fragment included the full intron sequence, a T7 promoter, reference hairpins for normalizing structure probing data, and adapters for universal RT and PCR primers. To generate a pool of DNA variants through error-prone PCR, we first assembled a reaction mix with the following for each intron: 10 µl of 2 ng µl$^{-1}$ template, 10 µl of 100 mM Tris pH 8.3, 2.5 µl of 2 M KCl, 3.5 µl of 200 mM $MgCl_2$, 4 µl of 25 mM dTTP, 4 µl of 25 mM dCTP, 4 µl of 5 mM dATP, 4 µl of 5 mM dGTP, 2 µl of 25 mM $MnCl_2$, 1 µl of Taq polymerase (Thermo Fisher), 2 µl of 100 mM primers (RR1 and RR107; Supplementary Table 5) and 51 µl of $H_2O$. We carried out 24 cycles of PCR with a 64 °C annealing temperature, and samples were purified using RNACleanXP beads (Beckman Coulter) with an AMPure bead to sample volume ratio of 1.8. We then transcribed RNA in vitro with 5 µl of 1× T7 RNA polymerase (NEB), 2 µl of 1 M DTT, 6 µl of 25 mM NTPs, 5 µl of 40% PEG-8000, 5 µl of T7 transcription buffer (NEB) and template DNA in a final reaction volume of 50 µl. Reactions were incubated at 37 °C overnight. Samples were treated with TURBO DNase (Thermo Fisher). RNA was purified with RNACleanXP beads using a 70:30 ratio of beads to 40% PEG-8000 as the beads mixture.

We proceeded with structure probing, RT, and library preparation for these RNA pools. We prepared 3 µl of 12.5 pmol RNA for each intron for DMS treatment and a no-modification control. We denatured the RNA by unfolding at 95 °C for 2 min, and left it on ice for 1 min. To fold the RNA, we added 5 µl of 5× folding buffer (1.5 M sodium cacodylate pH 7.0. 50 mM $MgCl_2$) and 14.5 µl of RNase-free $H_2O$, and incubated the mixture for 30 min at 37 °C. We modified the RNA by adding 2.5 µl of 15% DMS (DMS condition) or 100% ethanol (no-modification control), and heating at 37 °C for 6 min. The reaction was quenched by adding 25 µl of 2-mercaptoethanol and purified by AMPure bead purification, followed by elution in 7 µl of RNase-free $H_2O$. We reverse-transcribed the modified RNA with mutational read-through using TGIRT, as described above, in a 10 µl reaction volume. For reverse transcription, we used FAM-labeled primers that included a distinct index sequence for each construct and condition (RTB primers in Supplementary Table 5). The RT reaction was incubated at 57 °C for 3 hours, RNA was degraded with NaOH, and the solution was neutralized with an acid quench (see above). cDNA was purified using RNACleanXP beads and then amplified with PCR to add Illumina adapters for sequencing, using Phusion high-fidelity DNA polymerase (NEB) for 20 cycles at annealing temperature 65 °C with MaP forward and reverse primers (Supplementary Table 5). The resulting libraries were sequenced in two partial sequencing runs using Illumina MiSeq v3 600-cycle reagent kits, providing 300-nt paired-end reads.

M2-seq sequencing reads were processed using the M2seq package[46]. First, barcodes for the different introns and conditions were demultiplexed, obtaining at least 500,000 reads for each intron. The ShapeMapper 1.2 (ref. 81) software was used to align reads using Bowtie2 and compute mutation rates. The output from ShapeMapper was processed by the simple_to_rdat.py script to obtain RDAT files for both DMS and no-modification conditions, and processed with the Biers package[46] to generate *Z*-score plots, base-pairing probability matrices, and predicted secondary structures. These secondary

structure predictions were made using RNAstructure's Fold executable[33] with 500 bootstrapping iterations, guiding predictions with both one-dimensional and two-dimensional DMS reactivity data. Structures were visualized using VARNA and Biers.

## Assessing proposed functional structures with DMS data

Using our DMS data, we assessed previously proposed intron structures. Structures from Hooks et al.[37] were obtained by request. For structures with covarying residues, DMS-guided structure predictions were displayed when agreeing with proposed covarying base pairs (in *RPS9A*, *RPS9B*, and *RPL7A*), and structures from CaCoFold[97] were displayed otherwise (*RPS13*).

To expand the set of covarying residues identified across introns, we used the protocol and software developed by Gao et al.[23] with thresholds as follows. We obtained all complete and chromosome-level assemblies of genomes in the Ascomycota phylum from NCBI GenBank[98] (107 sequences, accessed on 17 August 2021). For each *S. cerevisiae* intron, we generated multiple-sequence alignments in flanked mode with a non-coding threshold of 50. We then applied R-scape[42] with an *E*-value threshold of 0.05, and we generated structures with CaCoFold[97]. We identified all introns with at least one significantly covarying pair between two non-consecutive residues in a stem with at least three base pairs. We evaluated the presence of these covarying residues in minimum free-energy secondary structures from ViennaRNA 2.0 (ref. 49). The set of snoRNA-containing introns was obtained from the *S. cerevisiae* snoRNA database[41].

## Zipper stem identification and stability calculation

We identified the positional constraints for forming a stem between the 5′ splice site and branch point ('zipper stems') by modeling 3D structures for introns in the context of the spliceosome using the Rosetta protocol FARFAR2 (ref. 99). For test constructs, we used variants of the *RPS17B* intron, which includes a previously characterized zipper stem[16]. To identify the minimum linker length for which a zipper stem is sterically compatible with the spliceosome, we built models of the *RPS17B* intron with a range of linker lengths between the 5′ splice site, zipper stem, and branch point. We based models on an A-complex spliceosome structure (PDB ID: 6G90)[74]. The tested variants included 36, 40, 45, 50, and 56 total nucleotides between the 5′ splice site, zipper stem strands, and the branch point. All models were built using Rosetta (release v3.10) for RNA homology modeling[100].

More specifically, for each modeled *RPS17B* variant, we used Rosetta's rna_thread to replace the pre-mRNA nucleotides resolved in the A-complex structure with the *RPS17B* sequence. We docked the *RPS17B* zipper stem into the binding site for an intron stem in the U1 snRNP by aligning the E-complex structure including a zipper stem (PDB ID: 6N7R ref. 25) with the A-complex structure in PyMOL. For each variant, 1,250 structures were sampled using Rosetta's rna_denovo. The A-complex spliceosome, docked zipper stem, and rethreaded *RPS17B* residues were treated as rigid bodies, and remaining linker residues in *RPS17B* were sampled freely. To accelerate structure sampling, we generated models using three scoring terms, only including rna_vdw at weight 10.0 (the RNA van der Waals score term), rnp_vdw at weight 50.0 (the RNA–protein van der Waals score term), and linear_chainbreak at weight 10.0 (a score term penalizing chain breaks). We collected linear chain break scores for each of the top ten sampled structures, monitoring whether linker lengths were too short to extend between docked nucleotides.

The 3D modeling approaches used here make numerous approximations that could lead to inaccuracies. First, components such as the docked zipper stem are included as a rigid body, which precludes capturing local dynamics in these components. Second, we used a simplified score function and did not model explicit waters and ions in our system. Without additional score terms (for example electrostatics), we cannot model specific interactions accurately. Finally, these

intron structures are highly flexible, but we analyzed only a small set of structures to obtain statistics. These structural models can be used to estimate pre-mRNA length constraints, rather than to analyze detailed structural features and interactions. Therefore, despite modeling limitations, this structural modeling can provide guidelines for zipper stem formation.

On the basis of the linear chain break scores, to designate a zipper stem, we required at least 42 nt connecting the zipper stem to the 5′ splice site and branch point. Additionally, these stems were required to begin at least 10 nt from the 5′ splice site and end at least 20 nt from the branch point, as closer positions could not form base pairs in the context of the A complex. Finally, we required that the linker between zipper stems and the 5′ splice site and branch point be at most 85 nt total, as zipper stems must colocalize the 5′ splice site and branch point. For a given secondary structure, we first identified the longest zipper stem satisfying these constraints with at most 5 bulge nucleotides, and we computed the stem's ΔG stability with RNAcofold[101]. Vienna 2.0 carries out free-energy calculations using a nearest-neighbor energetic model, which computes the total free energy by summing energies for stacking neighboring base pairs (including enthalpic and entropic contributions). As a caveat, these calculations will not capture non-local interactions that might be modeled by sampling 3D structures. Free energies were computed at the default temperature (37 °C) and salt concentration (1.021 M).

## Analyzing intron secondary structures surrounding canonical splice sites

We designated intervals surrounding the 5′ splice site, branch point, and 3′ splice site where pre-mRNA secondary structures would clash with the spliceosome. We identified the number of pre-mRNA nucleotides that threaded through structures of the E, A, pre-B, B, Bact, C and C* state spliceosome[25,74,102–106]. We noted that the following nucleotides threaded through the spliceosome in at least one state: 16 nt upstream and 10 nt downstream of the 5′ splice site; 33 nt upstream and 21 nt downstream of the branch point; and 8 nt upstream and 20 nt downstream of the 3′ splice site. We expected that these sequence intervals would be depleted of structure. We generated DMS-guided secondary structure predictions for each intron along with 50 upstream and downstream nucleotides, and we analyzed the proportion of nucleotides occluded by high-confidence stems.

## Analyzing intron secondary structures surrounding cryptic splice sites

We identified cryptic splice sites by searching for alternative 5′ splice sites, branch points, and 3′ splice sites in relevant intron regions. More specifically, we searched for any splice site sequences that had appeared in either canonical introns or previously annotated proto introns[78]. We noted that at least 42 nt were required between the 5′ splice site and branch point for pre-mRNA to be compatible with spliceosome structures (see above). Thus, we located cryptic 5′ splice site sequences by searching for alternative 5′ splice sites up to 42 nt upstream of the canonical branch point. Similarly, we located cryptic branch point sequences between 42 nt downstream of the 5′ splice site and the canonical 3′ splice site. Finally, we identified cryptic 3′ splice sites within the intron that were at least 10 nt downstream of the branch point sequence. Using intron structures derived from DMS probing data, including 50 nt of surrounding context, we evaluated the protection of these cryptic sites by high-confidence stems, comparing with the background protection of nucleotides in each region.

We additionally tested whether cryptic 3′ splice sites used only upon Prp18p inactivation (GAG, UG, CG and GG splice sites) were protected[107], finding a significant enrichment of secondary structure at these cryptic 3′ splice sites relative to surrounding background ($291/1,559 = 18.6\%$ protection at these sites relative to $526/4,280 = 12.3\%$ protection at background sites; $P < 0.001$). However, cryptic 3′ splice

sites matching the standard consensus sequence (UAG, CAG or AAG) were more protected (53/183 = 29.0% protection).

## DMS structure analysis for introns and coding regions

We made reactivity-guided secondary structure predictions to identify structural features of *S. cerevisiae* introns and coding sequences for intron-containing genes. To count the number of introns that included zipper stems, we used the zipper stem criteria identified through spliceosome modeling as described above. We additionally identified 'downstream stems' between the branch point and 3′ splice site with at least 6 base pairs and bootstrapping support at least 70%.

To calculate secondary structure properties, we used both fine- and coarse-grained graphs. Fine-grained graphs included a node for every base pair and single-stranded position, with edges between consecutive positions or base pairs. Coarse-grained graphs included a single node for each stem, junction, loop, single-stranded 5′ end, and single-stranded 3′ end. Fine-grained graphs were used to compute the maximum extrusion from ends, using the arnie package[108] to find the shortest path length from each position to either end of the sequence. The maximum value for this shortest path across all nodes yielded the maximum extrusion from ends, providing the effective distance between the sequence ends and the position farthest from these ends. The values for maximum extrusion from the ends were normalized by the length of each construct. Coarse-grained graphs were used to identify the longest stem with at most 10 total loop nucleotides and at least 90% bootstrapping probability. Additionally, coarse-grained graphs were used to compute average helix confidence estimates for all stems with a length of at least 6. Finally, Gini coefficients were calculated on the basis of mutational frequency values for intron and coding regions, using windows of 20 nt and scanning in intervals of 10 nt. To calculate the Gini index, we required that every position had per-position coverage at least 15,346 as computed from rf-rctools view, corresponding to 0.6 Pearson correlation of reactivity profiles between replicates.

## Modeling of intron stems in spliceosome structure

We modeled the *RPL28* and *RPL36B* introns in the context of the A-complex spliceosome (PDB ID: 6G90)[74], using a protocol similar to that for *RPS17B* zipper stem modeling, described above. Intron nucleotides resolved in the PDB structure were replaced by the corresponding sequences from *RPL28* and in *RPL36B* with the rna_thread application[109]. The secondary structure used for modeling was obtained from RNAstructure prediction guided by DMS data, and helices were supplied as rigid bodies. 4564 models were sampled using the Rosetta application rna_denovo[99] with the simplified score function described above.

## Comparison of introns with decoys

To determine whether secondary structure features can distinguish authentic introns from unspliced genomic sequences that match splicing motifs, we assembled a set of decoy introns. To find these decoy introns, we first computed the position weight matrix (PWM) for the 5′ splice site (6 nt with the consensus sequence GUAUGU), branch point (8 nt with the consensus sequence UACUAACN) and 3′ splice site (3 nt with the consensus sequence YAG) for all annotated *S. cerevisae* introns. We then obtained three length distributions from annotated introns: the complete intron length, the distance between the 5′ splice site and branch point, and the distance between the branch point and 3′ splice site. To find decoy introns, we scanned all genes for intervals containing 5′-splice-site, branch-point, and 3′-splice-site sequences matching the PWMs and three length distributions for introns, using cut-offs that captured at least 95% of canonical introns. We filtered these decoy sequences to exclude annotated introns. For each authentic intron and decoy sequence, we assembled length-matched control sequences that were shifted in the genome by 500 nt. DMS-guided structure predictions and structural features were computed for each intron, decoy and matched control sequence with sufficient coverage, as described above. Additionally, we computed the distance between the 5′ splice site and branch point as the length of the shortest path in the secondary structure graph between these sites. We then determined whether authentic introns and decoy intron sequences enriched for these structural features more than shifted genomic controls.

## Visualizing the structure landscape for *S. cerevisiae* introns

We used secondary structure features from DMS-MaPseq to classify *S. cerevisiae* introns. For each intron, we included the following metrics: the sequence length, the presence of a zipper stem, zipper stem free energy, the presence of a downstream stem, downstream stem free energy, maximum extrusion from ends, the longest stem length, the average helix confidence estimate, the maximum Gini coefficient, and average DMS accessibility across splice sites and branch points. Hierarchical clustering was performed using scipy's hierarchy module[110] with Ward linkage and optimal leaf ordering, and Seaborn's clustermap was used to generate a dendrogram and heatmap.

## Designing intron structure variants for structure-function assay

We chose seven structured introns to mutate systematically by identifying secondary structures with zipper stems, covarying residues, and long stems. Intron library sequences were constrained to 300 nt, including fixed primer binding sites, leaving around 200–250 nt downstream from the 5′ splice site as the variable region. For each intron, we designated stem and loop sets, with each stem set containing one or more continuous stems, and each loop set containing a single junction or loop.

For each stem set and loop set, we generated candidate variants and scored these candidates on the basis of the desired secondary structure properties. For each stem set, we first generated 10,000 randomized and 10,000 shuffled sequences mutating the 5′ strands of the stems. Additionally, if the stem's 3′ end fell within the library's length constraints, we generated rescue sequences for each variant, installing compensatory mutations in the 3′ strand. For each loop set, we generated 20,000 random and shuffled variants. For each variant and rescue sequence, we calculated the minimum free energy structure with Contrafold[111] within the full intron context. We then assessed whether the variant disrupted any targeted stems without altering the remainder of the secondary structure. More specifically, for each variant, we computed two penalty scores: a variable region penalty and a constant region penalty. The variable region penalty was the sum of the number of paired residues in the variable region along with the number of base pairs present in this region. The constant region penalty focused on nucleotides outside the variable region, computing the number of native base pairs disrupted by the variant, and adding half of the number of base pairs gained in the variant. If applicable, we calculated the rescue penalty as the number of base pairs altered between the native secondary structure and the rescue sequence. We chose the top 4–8 unique variants per region on the basis of the sum of the variable region, constant region and rescue penalties. Our variants did not affect the 5′ splice site, branch point or 3′ splice site. We additionally included the wild-type sequence for each set. The final chosen variant sequences and their penalty scores are in Supplementary Table 3.

## Designing gene context for intron variants

Introns were cloned into their native gene context to better mimic the effects of structures in native *S. cerevisiae* on gene expression. Each construct included the promoter *TDH3* (which has been shown to induce high expression levels[48]), the 5′ untranslated region (UTR) beginning at the transcription start site, the full gene including the intron sequence and the *ADH1* terminator (shown to yield high mRNA half life[112]) (Extended Data Fig. 8c). For each gene, we used the YeasTSS database to identify the most prominent transcription start site to

start the 5′ UTR[113]. When cloning intron libraries into this gene context, we included a 8-nt fixed sequence in the 5′ UTR to distinguish the inserted gene from the endogenous copy (AGCGGACG for *QCR9*, *RPL28*, *RPS9A* and *RPL36B*; AGAAGACG for *RPS14B*; AGAAGAGC for *RPL7A* and AACTGCCC for *RPS9B*). Additionally, we included a random 12-nt (12N) barcode in the 5′ UTR to identify variants even after introns are spliced out. We ran simulations sampling 12N barcodes to confirm that, for our expected number of clones (10,000–50,000), most barcodes would have an edit distance of at least two from all others (Extended Data Fig. 8d). We therefore expected that these barcodes would enable unambiguously assigning spliced reads to intron variants.

### Growth media for structure variant library assay
We used the following growth media and plates for our structure–function assay:

- YPD medium: YPD powder (Difco; yeast extract 10 g l$^{-1}$, peptone 20 g l$^{-1}$, dextrose 20 g l$^{-1}$)
- YPD plates: Difco YPD powder 50 g l$^{-1}$, Difco agar powder 20 g l$^{-1}$
- SD-LEU plates: premix from Takara Bio
- SD-HIS plates: premix from Takara Bio

### Constructing yeast background strain for integrating structural variant libraries
We integrated our variant library into the yeast genome using homologous recombination and CRISPR–Cas9-induced double-stranded breaks. We chose the *ARS416* locus as our integration site, because it is a highly efficient site[114]. To ensure that our library was integrated in the expected locus, we first constructed the background strain PLH001, which inserted a partial *LEU2* selection marker at this locus, beginning at the second codon of the coding sequence. We then designed our structure variant library constructs to include the *LEU2* promoter and start codon, selecting for integration at the correct locus with SD-LEU.

The strain PLH001 was constructed from background strain BY4741 (Mata, his3Δ1, leu2Δ0, met15Δ0, ura3Δ0). A partial *LEU2* selection marker and a complete *HIS3* selection marker were integrated into the *ARS416* gene locus[114]. *LEU2* and *HIS3* were obtained from the pHLUM plasmid[115], adding homology arms by PCR for genomic integration into *ARS416*. These constructs were cloned into the pUC19 backbone[116] using NEBuilder Hifi DNA Assembly (NEB) (Extended Data Fig. 8c and insert sequence RRLH in Supplementary Table 5). We obtained a linear insert for transformation with PCR, which was integrated into the *ARS416* locus using SD-HIS selection[117].

### Constructing plasmid backbones for structure variant library
We first cloned a plasmid backbone for each intron variant containing the intron's gene context (pRR1–pRR7), and we then cloned our intron libraries into this background. For each intron, the background gene consisted of the *TDH3* promoter, the 5′ UTR, the gene and intron excluding the variant library, the *ADH1* terminator and homology arms for insertion into the *ARS416* locus (Extended Data Fig. 8c). All PCR reactions in the following sections were carried out with the NEBNext Ultra II Q5 Master Mix (NEB). We used the following gene fragments from Integrated DNA Technologies to build gene background constructs: (1) gene blocks with the homology arm and *TDH3* promoter; (2) a gene block with the *ADH1* terminator, *LEU2* promoter, *LEU2* start codon and *LEU2* homology arm; and (3) gene blocks with the target gene and constant regions in the intron (gblockRR4-12; Supplementary Table 5). Gene blocks were amplified with PCR using primers RR222–RR233 (Supplementary Table 5) and cloned into the background plasmid using a NEBuilder Hifi DNA Assembly (NEB) reaction.

### Cloning structure variant library with high-efficiency transformation
Intron libraries were obtained as one oPool per intron from Integrated DNA Technologies, and the seven sub-libraries were amplified by PCR. For each intron, we carried out eight parallel PCR reactions with limited PCR cycling to reduce bias. We used the following primer pairs: RR273 and RR249 for *RPS14B*; RR274 and RR251 for *RPS9B*; RR275 and RR253 for *RPS9A*; RR276 and RR255 for *QCR9*; RR277 and RR257 for *RPL36B*; RR279 and RR261 for *RPL28*; and RR280 and RR263 for *RPL7A* (Supplementary Table 5). Primers included overhangs for plasmid insertion, and forward primers included 12N random barcodes. PCR was carried out with 30 ng template and 14 amplification cycles. The eight PCR reactions were concentrated together with the QIAQuick PCR Purification Kit (Qiagen) and gel purified with a MinElute PCR Purification Kit (Qiagen). Plasmid backbones pRR1–pRR7 were linearized using PCR primers RR223 and RR239–RR245 using a 3.5-min extension time, gel purified, and treated with DpnI (NEB). We transformed NEB Stable Competent *E. coli* cells (NEB) with NEBuilder HiFi DNA Asssembly (NEB) reactions for each of our 7 sub-libraries, using 2.5 µl of ligation product for 32 µl of competent cells. We obtained 6,000–46,000 clones per sub-library (30–230× library coverage) and validated 8–16 clones per sub-library with Sanger sequencing. Plasmid DNA was extracted using the ZymoPURE II Plasmid Midiprep Kit (Zymo Research).

### Yeast genomic integration for structure variant library assay
The plasmid library was linearized by restriction digest. For the initial digestion, we combined 3 µl of XcmI (NEB) with 10 µl of 10X R2.1 buffer (NEB) and added 5 µg of each plasmid sub-library in a total volume of 100 µl. Reactions were left at 37 °C for 60 min and at 65 °C for 20 min. Then, for the second digestion, we added 2.0 µl of 2.5 M NaCl, 4.16 µl of Tris HCl pH 7.9, and 1.5 µl of the BglII enzyme, followed by an incubation at 37 °C for 60 min. To increase the efficiency of integration, the linearized plasmid library was transformed with Cas9. We obtained 50 µg of plasmid p426 (ref. [114]) with the ZymoPURE II Plasmid Maxiprep Kit (Zymo Research) and linearized with AhdI digest.

We used heat-shock transformation in PLH001 with the linearized Cas9 plasmid and library insert for genomic integration with the lithium acetate (LiAc), single-stranded DNA (ssDNA) and polyethylene glycol (PEG) transformation protocol[117]. For each sub-library transformation, we used 5 µg linearized plasmid library and 5 µg linearized p426 plasmid to transform 25 ml of cells at an OD$_{600}$ of 0.5. We plated transformed cells on SD-LEU selection plates. As negative controls, we transformed DNA from one sub-library into BY4741, and we transformed p426 alone without a library insert into PLH001. We saw no colony growth for either of these negative control conditions on SD-LEU selection plates. We obtained 9,000–35,000 colonies for each sub-library, yielding 45–175× library coverage for each intron. We verified correct integration for four to eight colonies per sub-library with Sanger sequencing.

### Genomic DNA sequencing for structural variant library assays
We sequenced genomic DNA to assign barcodes to each designed variant sequence and to obtain transformation frequencies. We extracted DNA using the YeaStar Genomic DNA Kit (Zymo Research). We then carried out an indexing PCR with limited cycles to amplify target intron library regions and add i5 and i7 indices for Illumina sequencing with the primers RR281–RR294 (Supplementary Table 5). These primers amplified the target sub-libraries including barcodes, with forward primers binding to the fixed 8-nt sequences that distinguished the inserted intron library from the endogenous gene copy. For each indexing PCR, we used 50 ng genomic DNA template and 7 amplification cycles. We then amplified our indexing PCR sample with P5 and P7 primers, now with 11–14 PCR cycles and annealing at 66 °C. Samples were purified using RNACleanXP beads (Beckman Coulter) with a AMPure

bead to sample volume ratio of 0.8 to remove primer dimers from the sample. The sequencing libraries were quantified and the libraries were sequenced using an Illumina MiSeq v3 600-cycle kit.

## Targeted RNA sequencing for structure variant library assay

We aimed to sequence RNA to obtain levels of spliced and unspliced RNA. We started overnight cultures from 100 µl of yeast sub-library glycerol stocks, and we used 5 ml culture per sub-library in two columns from the YeaStar RNA Kit (Zymo Research), with 2.5 µl of Zymolase for every 2.5 ml starting cell culture. We pooled RNA extracted from each sub-library, combining 50 µg total RNA from each sub-library to yield 350 µg total RNA. For RNA sequencing, as described previously, we first depleted rRNA from our total RNA with RNase H, using 50-mer oligos to tile rRNA sequences. We then carried out size selection with RNA Clean and Concentrator-5 kit (Zymo Research), followed by treatment with TURBO DNase, and purified reactions with the RNA Clean and Concentrator-5 kit (Zymo Research).

To assess the removal of contaminating genomic DNA from our total RNA, we carried out a reverse transcription reaction with and without reverse transcriptase and checked for amplification of control RNA regions with PCR. For reverse transcription, we used the iScript Reverse Transcription Supermix (Bio-Rad). We purified samples with Oligo Clean and Concentrator columns (Zymo Research), with elution in 15 µl $H_2O$. We then performed PCR with the MATa1 and RPL36B primers as described above.

We next generated cDNA using the DMS-MaPseq library-preparation strategy described above. In brief, first, we fragmented our RNA and purified with the RNA Clean and Concentrator-5 kit (Zymo Research). We then removed 3′ phosphate groups from fragmentation with rSAP (NEB) treatment followed by rSAP inactivation. We prepared adenylated linker from RR118 (Supplementary Table 5), and we then ligated this linker to our RNA sample with a T4 RNA ligase 2 truncated KQ (NEB) reaction, adding adenylated linker in 2× molar excess of the RNA sample. Excess linker was degraded with 5′ deadenylase (NEB) and RecJf (NEB) exonuclease. For reverse transcription, we used the TGIRT enzyme (InGex), using the reverse transcription primer RR301 which includes a 10-nt UMI and Illumina adapters (Supplementary Table 5). We degraded RNA and then purified cDNA with an Oligo Clean and Concentrator column (Zymo Research).

Next, we amplified cDNA with targeted PCR primers for each sub-library to generate final sequencing samples. We first carried out a short indexing PCR for each sample. For these PCR reactions, we used the following primer pairs: RR281 and RR302; RR283 and RR303; RR285 and RR304; RR287 and RR305; RR289 and RR306; RR291 and RR307; and RR293 and RR308 (Supplementary Table 4). Our forward PCR primers included the 8-nt fixed sequence that distinguished the inserted library from the endogenous gene. These primers allowed for the amplification of cDNA, including barcode sequences and the splice site junction. Additionally, these primers included unique i5 and i7 index sequences. For indexing PCR, we used Q5 with seven amplification cycles. We then carried out 24–28 additional PCR cycles with P5 and P7 primers (Supplementary Table 5). Finally, to obtain double-stranded DNA for sequencing, we carried out two rounds of size-selection and purification with RNACleanXP beads (Beckman Coulter), using an AMPure bead to sample volume ratio of 0.7, followed by elution in 15 µl $H_2O$. The sequencing libraries were sequenced using a partial Illumina NovaSeq S4 lane, providing around 700 million paired-end reads of length 150.

## Genomic DNA sequencing data analysis for structure variant library

Genomic DNA sequencing provided links from barcodes to intron variants along with quantification for the number of transformants for each variant. To prepare genomic DNA sequencing reads for analysis, we first used UMI-tools[77] to extract barcodes and then error-corrected

paired-end reads using bbmerge[118], removing Illumina adapter sequences and ensuring consistency between paired-end reads.

We used the following pipeline to assemble consensus intron variant sequences for each barcode using the fgbio toolkit (fulcrumgenomics.github.io/fgbio/). First, paired-end reads were aligned to introns using BWA[119], alignments were sorted using fgbio's SortBam, and mate-pairs were annotated with fgbio's SetMateInformation. We then grouped reads with the same barcode using fgbio's GroupReadsByUmi with the adjacency method. Consensus reads were called using fgbio's CallMolecularConsensusReads with the flags -m 20 -M 5 to require that bases have a $Q$ score above 20, and that at least 5 reads covered each consensus sequence. We then filtered these sequences using fgbio's FilterConsensusReads with the flags -M 5 -q 20 -N 20 -E 0.01, additionally requiring that consensus bases have a quality of at least 20, and reads have average error rates lower than 0.01. We merged paired-end reads using bbmerge[118], obtaining consensus length distributions. Finally, to ensure that barcodes are mapped primarily to single consensus reads, we kept the primary consensus sequence for each barcode and required that at most 5% of reads map to secondary consensus sequences, removing 0.4–4.7% of barcodes.

From our consensus sequences, we obtained barcode mappings, along with transformation frequencies, for designed variants. We collated barcodes that mapped to designed variant sequences, and found at least ten unique barcodes for many of the designed variants. We additionally found that some colonies had consensus sequence variants that were mutations of wild-type sequences or designed variants. From these sequences, we retrieved mutated *QCR9* branch point sequences as a control sample. To obtain transformation frequencies, we found read coverage using the output from fgbio's FilterConsensusReads.

## RNA-sequencing data analysis for structure variant library

To process RNA-sequencing data, we first extracted barcodes (one for each transformant) and UMIs (one for each cDNA molecule) from our sequencing reads with UMI-tools[77]. We then removed Illumina adapters from the reads using cutadapt, additionally trimming bases with a $Q$-score <20. We then aligned reads to spliced isoform reference sequences. For our reference sequence annotations, we collated all possible isoforms capturing spliced and unspliced transcripts (four isoforms for two-intron *RPL7A* and two for all other genes), and we generated gff annotation files with gmap[120]. We used TopHat2 (ref. 86) to align sequencing reads, using the following flags:–no-novel-juncs -T –b2-mp 3,1 –b2-rdg 5,1 –b2-rfg 5,1 –segment-length 20 –segment-mismatches 3 –read-gap-length 7 –read-edit-dist 50 -m 1 –max-insertion-length 19 –max-deletion-length 19. We generated fasta files from the resulting alignment files with samtools[82].

We classified each aligned sequence into one of three categories: unspliced, spliced at the expected junction, or other, requiring 14 nt of agreement at the unspliced or spliced junction. Sequences that did not include canonical spliced or unspliced junctions were classified as alternative splicing events if they appeared in at least ten UMIs per barcode and at least three barcodes per variant. For canonical spliced or unspliced junctions, we counted reads after deduplicating UMIs. We then computed two metrics for each barcode: the retained intron fraction and the normalized mRNA level. The retained intron fraction was the ratio between the unspliced read counts and total read counts. The normalized mRNA level was the ratio between the spliced read count and the transformation frequency from genomic DNA sequencing. For each intron sub-library, we used the barcode-variant correspondence from genomic DNA sequencing to collect retained intron fractions and normalized mRNA levels.

For all the stem and loop sets assessed in each intron sub-library, we compared RI fractions and normalized mRNA levels between the wild-type sequence versus stem disruption variants, and the stem disruption variants versus the rescue sequences. Comparisons were

made using two-sided permutation tests (SciPy[110]) for the difference in mean statistic, with 100,000 resamples of data labels per comparison.

**RT–qPCR from individual variant strains to validate VARS-seq**

We first cloned intron variants of *RPS9A* and *RPL36B* individually into the corresponding plasmid backbones to generate complete gene constructs with individual variant introns. For *RPS9A*, we cloned in eight intron variants, including the wild type, 5′ mutant, 3′ mutant, and rescue sequences for two different barcodes (from gblock_RR15-18 with two barcodes subsequently added by PCR). For *RPL36B*, we cloned eight intron variants including the wild type, 5′ mutant, 3′ mutant, and rescue sequences for two different stem sets labeled 'short' and 'long' (cloning from gblock_RR19-26). For *RPS9A*, gblocks were amplified by PCR with primers RR253 and RR337 to include the barcode expected to have high retained intron levels, and with primers RR253 and RR338 to include the barcode expected to have low levels of retained introns. For *RPL36B*, gblocks were amplified with primers RR343 and RR345 for the short stem set and with primers RR345 and RR344 for the long stem set. Intron library inserts were assembled into the corresponding linearized backbones (pRR5 and pRR7) using NEBuilder HiFi DNA Assembly (NEB) as described above, with cloning in NEB Stable Competent *E. coli* cells (NEB).

We next integrated these intron variant constructs into *S. cerevisiae* to analyze the effects of these variants on splicing. Plasmids extracted from correct clones were linearized by restriction digest, with a first restriction digest by XcmI (NEB) and a second digest with BglII (NEB), as described above. These linearized plasmids included complete gene constructs for *RPS9A* or *RPL36B* with intron variants, homology arms to the genomic integration site, and a partial *LEU2* selection cassette that would be completed by genomic integration into PLH001. Linearized plasmids were transformed into PLH001 and plated on SD-LEU selection plates. We verified clones with Sanger sequencing of extracted genomic DNA.

We next extracted RNA from each strain and evaluated levels of spliced and unspliced transcripts with qPCR. To extract RNA, we used the YeaStar RNA Kit (Zymo Research) with 5 ml culture with an $OD_{600}$ of 0.5–0.6, as described above. We treated the samples with TURBO DNase (Thermo Fisher) and cleaned up samples with Zymo5 RNA columns. We then carried out reverse transcription with the iScript Reverse Transcription Supermix (Bio-Rad) along with a no-reverse-transcriptase control, and we purified samples with Oligo Clean and Concentrator columns (Zymo Research).

With this reverse transcription product, we then carried out qPCR with the iTaq Universal SYBR Green Supermix (Bio-Rad), using 1 μl of forward and reverse primer in a 20-μl reaction. Forward primers were designed to avoid the endogenous gene locus by including the fixed 8-nt sequence inserted into the constructs. Reverse primers were designed to amplify only the spliced construct (binding to the exon junction) or only the unspliced construct (binding within the intron). For *RPS9A* we used primers RR313 and RR314 to amplify the spliced product, and RR317 and RR319 to amplify the unspliced product. For *RPL36B* we used primers RR346 and RR347 to amplify the spliced product, and RR346 and RR348 to amplify the unspliced product. A region of *ACT1* was used as a control interval, with primers RR341 and RR342. All primers were verified by checking amplicons on agarose gels.

**De novo computational structure prediction and structure metrics**

We predicted secondary structure ensembles for *S. cerevisiae* introns and control sequence sets with de novo structure prediction. For direct comparison between DMS-guided RNAstructure prediction and de novo RNAstructure prediction, we computed features in both cases only for the 161 introns with a coverage of 1,971 (corresponding to $r^2 > 0.25$) from DMS-MaPseq; for other cases, we made predictions for the complete set of introns filtered to exclude sequences smaller than 50 nt or larger than 600 nt. Three control sequence sets were assembled. First, for the 'shifted control,' the genome coordinates for each intron were shifted 500 nt downstream of the 5′ splice site. For the 'sequence-matched control,' the shifted control's sequences were replaced at the positions of splice sites to match the corresponding intron. The final set, the 'shuffled control,' was obtained by shuffling each intron's sequence randomly.

Secondary structure ensemble predictions for each set of sequences were obtained either through minimum free energy predictions or by sampling suboptimal secondary structures. For suboptimal structure sampling, 1,000 structures were sampled using the Arnie package (https://github.com/DasLab/arnie) to call structure prediction executables from Vienna 2.0 (ref. 49) and RNAstructure[33]. We additionally made structure predictions with 50 nt of surrounding context upstream and downstream of the intron. For each intron and matched control, we computed average values across these structure ensembles for the zipper stem free energy, downstream stem free energy, length of the longest stem, maximum extrusion from ends, and distance between the 5′ splice site and branch point. As described previously, we compared the intron and control values using the non-parametric Wilcoxon ranked-sum test.

**Structure prediction for *Saccharomyces* genus**

We obtained multiple sequence alignments for introns in 20 species in the *Saccharomyces* genus from Hooks et al.[50]. To count the number of orthologs for each intron, we found all the species with non-empty sequences aligned to the *S. cerevisiae* intron. We additionally counted the number of zipper stem orthologs by finding all the non-empty regions aligned to the longest zipper stem in each *S. cerevisiae* intron. Finally, we computed the average percentage sequence conservation across each complete intron and across the intron's zipper stem, if one was present.

We next evaluated secondary structure feature enrichment for the introns across the *Saccharomyces* genus. We tested each intron in these 20 species against matched control sets, with each intron having a matching shuffled sequence control. We additionally compared the secondary structure ensembles of all the introns in each of these species' genomes to phylogenetic controls. To construct phylogenetic control sequence sets, we measured mutation and indel rates between each intron and its homologous *S. cerevisiae* intron, and we created an intron's matched control sequence by inserting mutations and indels randomly into the homolog *S. cerevisiae* intron at this measured average frequency. As described earlier for the analysis of *S. cerevisiae* introns, for each of these comparisons, we predicted secondary structure ensembles for the intron set and control set, computed secondary structure features, and compared between sequence sets with a Wilcoxon ranked-sum test.

**Reagent and resource sharing**

Yeast strains are available upon request from the authors. Requests for resources and reagents should be directed to the corresponding author, Rhiju Das (rhiju@stanford.edu).

**Reporting summary**

Further information on research design is available in the Nature Portfolio Reporting Summary linked to this article.

## Data availability

All sequencing data are available at the Gene Expression Omnibus (GEO) accession number GSE209857. DMS-derived secondary structures are included in Supplementary Table 1. Processed DMS reactivity data, control RNA secondary structures and base-pairing probabilities from structure prediction with bootstrapping can be found at https://github.com/ramyarangan/DMS_intron_analysis. VARS-seq intron variant sequences along with spliced and unspliced read counts are

included in Supplementary Tables 3 and 4. Source data for Figs. 1–3 and 5–7, along with Extended Data Figs. 1–5, 8 and 9 are provided with this paper. Source data for the Supplementary figures have been provided as a Supplementary Data File. Source data for Supplementary Fig. 14 is included in Supplementary Table 4.

## Code availability

Code for assessing DMS-MaPseq reactivity values, evaluating control RNA structures, and analyzing DMS-derived secondary structures is available at https://github.com/ramyarangan/DMS_intron_analysis. Code for computational secondary structure prediction, secondary structure feature calculation, statistical comparisons, and evolutionary analyses is available at https://github.com/ramyarangan/pre-mRNA_Secstruct. Code for designing VARS-seq sequences and analyzing gDNA and RNA sequencing data from VARS-seq is available at https://github.com/ramyarangan/VARSseq.

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

## Acknowledgements

We thank K. B. Hooks (Faculty of Life Sciences, University of Manchester, M13 9PT, United Kingdom) and D. Delneri (Faculty of Life Sciences, University of Manchester, M13 9PT, United Kingdom) for providing secondary structure data upon request. We thank A. Hoskins, J. Puglisi, T. Lan and S. Rouskin for illuminating discussions on this project, and we thank V. V. Topkar and I. Zheludev for advice on experimental protocols. We thank Stanford University for providing computational resources and support for the Sherlock 2.0 cluster that contributed to these results. This work was supported by the National Science Foundation Graduate Research Fellowship Program grant no. 1650114 (R.R.); the Gerald J. Lieberman Fellowship (R.R.); the National Institutes of Health grant R35GM145266 (M.A.), a Stanford Bio-X Interdisciplinary Initiative Award (R.D.); the National Institutes of Health grant R35GM122579 (R.D.) and the Howard Hughes Medical Institute (HHMI) Investigator Program (R.D.). This article is subject to HHMI's Open Access to Publications policy. HHMI lab heads have previously granted a nonexclusive CC BY 4.0 license to the public and a sublicensable license to HHMI in their research articles. Pursuant to those licenses, the author-accepted manuscript of this article can be made freely available under a CC BY 4.0 license immediately upon publication. The funders had no role in study design, data collection and analysis, decision to publish or preparation of the manuscript.

## Author contributions

R.R. and R.D. conceived the project. R.R. designed and performed experiments and carried out computational analyses. O.H. and M.A. developed the *S. cerevisiae* strain OHY001 used in the project. R.H. conducted in vitro and targeted DMS probing experiments. P.P. tested alternate structure prediction approaches. R.R., M.A. and R.D. interpreted the results. R.R. wrote the initial draft, and R.R., M.A. and R.D. wrote the final manuscript with input from all authors.

## Competing interests

The authors declare no competing interests.

## Additional information

**Extended data** is available for this paper at https://doi.org/10.1038/s41594-025-01565-x.

**Correspondence and requests for materials** should be addressed to Rhiju Das.

**Peer review information** *Nature Structural & Molecular Biology* thanks Aaron Hoskins and the other, anonymous, reviewer(s) for their contribution to the peer review of this work. Primary Handling Editor Dimitris Typas was the primary editor on this article and managed its editorial process and peer review in collaboration with the rest of the editorial team. Peer reviewer reports are available.

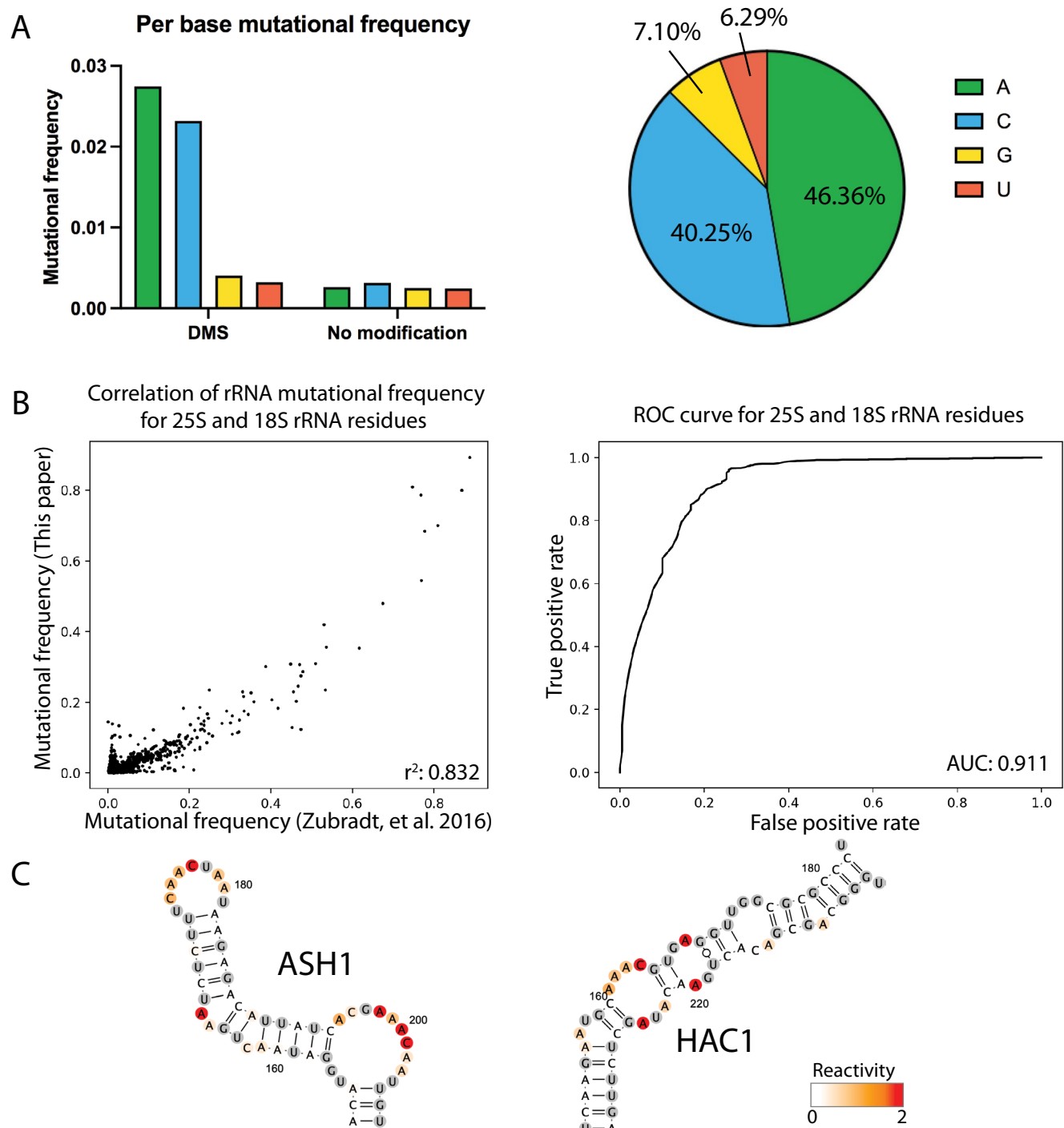

**Extended Data Fig. 1 | DMS-MaPseq data quality. A)** Per-base mutational frequencies. **B)** Accuracy of mutational frequency values for rRNA residues. **C)** *HAC1* and *ASH1* positive control structures with overlaid reactivity profiles.

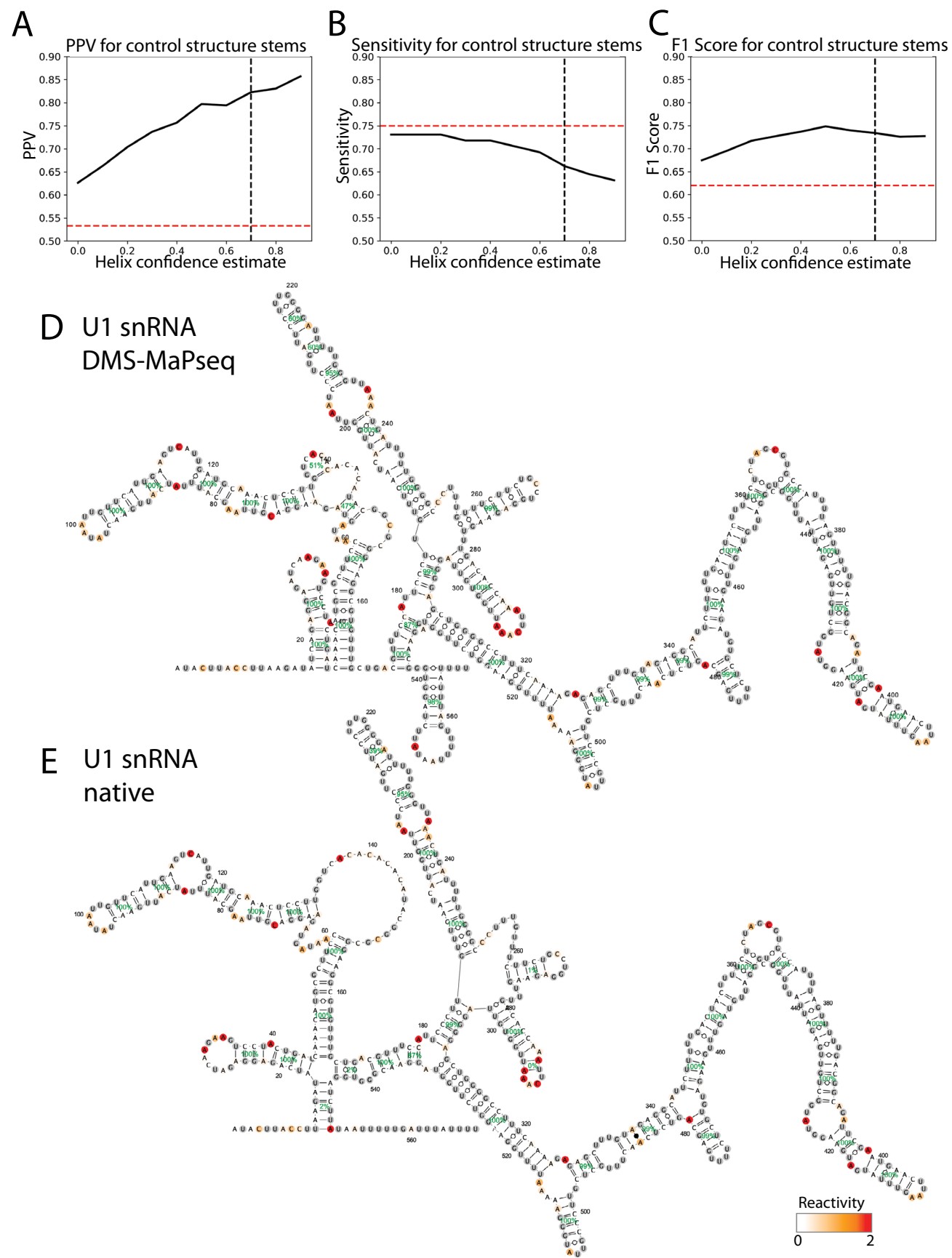

**A** PPV for control structure stems

**B** Sensitivity for control structure stems

**C** F1 Score for control structure stems

**D** U1 snRNA DMS-MaPseq

**E** U1 snRNA native

Reactivity

0    2

**Extended Data Fig. 2 | See next page for caption.**

**Extended Data Fig. 2 | Support from DMS reactivity for *in vivo* formation of control structures. A)** PPV, **B)** sensitivity, and **C)** F1 score for structure prediction of a set of control RNA structures (rRNAs, tRNAs, snRNAs, and mRNAs; Supplementary Table 2), using RNAstructure guided by DMS with varying helix confidence estimate cutoffs for calling stems. The black dotted line represents the helix confidence estimate 0.7 chosen in this paper. The red dotted line represents the PPV, sensitivity, and F1 score for Vienna RNA structure prediction without using DMS data. **D)-E)** DMS-MaPseq structure prediction for the U1 snRNA compared to the native secondary structure. **D)** DMS-guided secondary structure prediction for the U1 snRNA, with reactivity values overlaid and helix confidence estimates indicated in green percentages. **E)** Native secondary structure for the U1 snRNA, with DMS reactivity overlaid along with helix confidence estimates. For the native structure, helix confidence estimates were computed as the percent of bootstrapping iterations where the helix was recovered when sampling DMS reactivity values and making structure predictions.

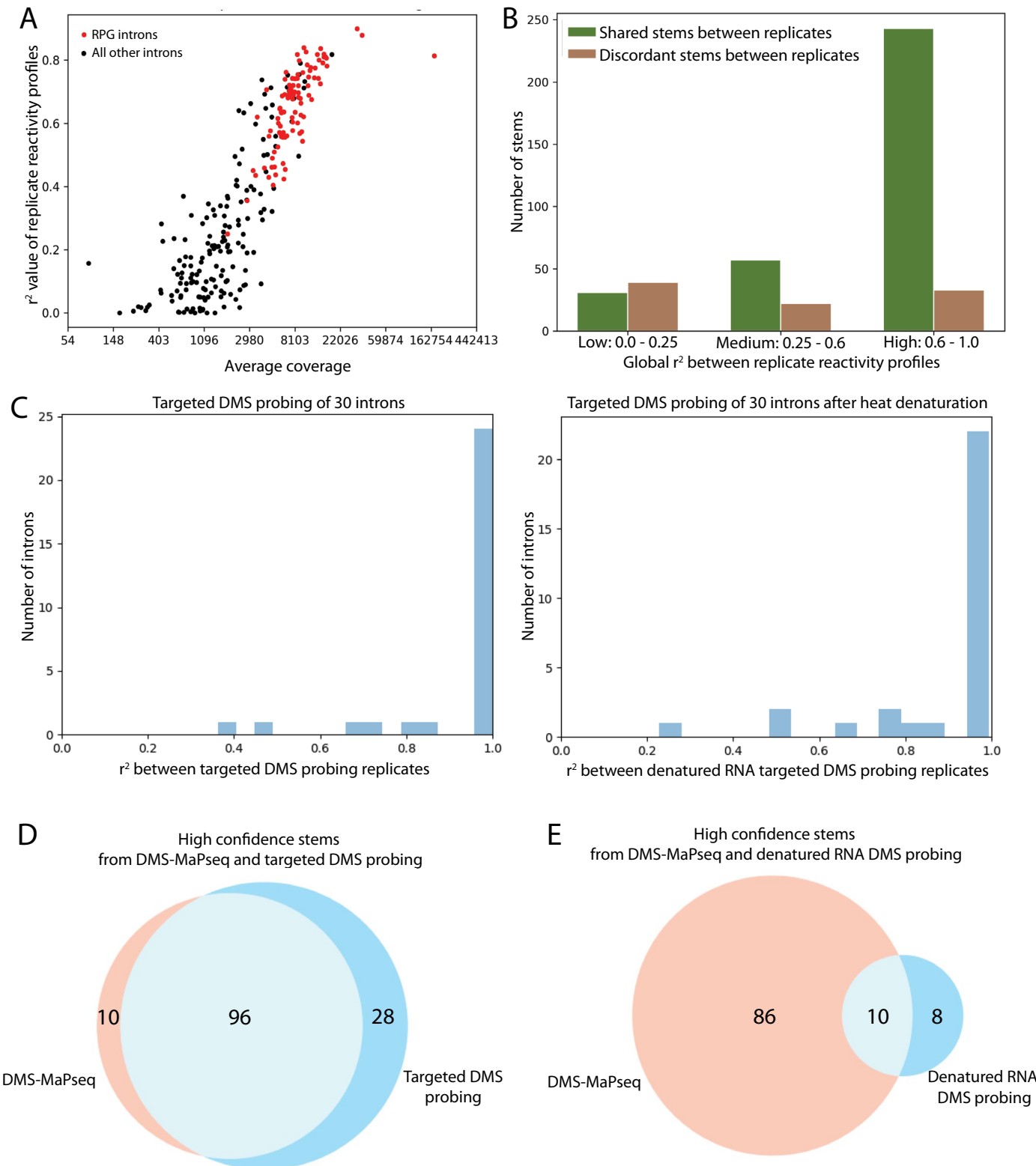

**Extended Data Fig. 3 | DMS-MaPseq and targeted DMS probing reproducibility between replicates. A)** $r^2$ between replicates for each intron in *S. cerevisiae* versus the average sequencing coverage between replicates. **B)** Bar graph comparing the number of high confidence stems that are shared vs discordant between replicates for introns in different ranges of replicate correlation. **C)** $r^2$ between

replicates of targeted DMS probing for 30 introns without (left) or with (right) heat denaturation. **D)** The number of high confidence stems found in 24 introns with $r^2 > 0.9$ from targeted DMS probing and $r^2 > 0.6$ from DMS-MaPseq. **E)** The number of high confidence stems found in 22 introns with $r^2 > 0.9$ from targeted DMS probing of denatured RNA and $r^2 > 0.6$ from DMS-MaPseq.

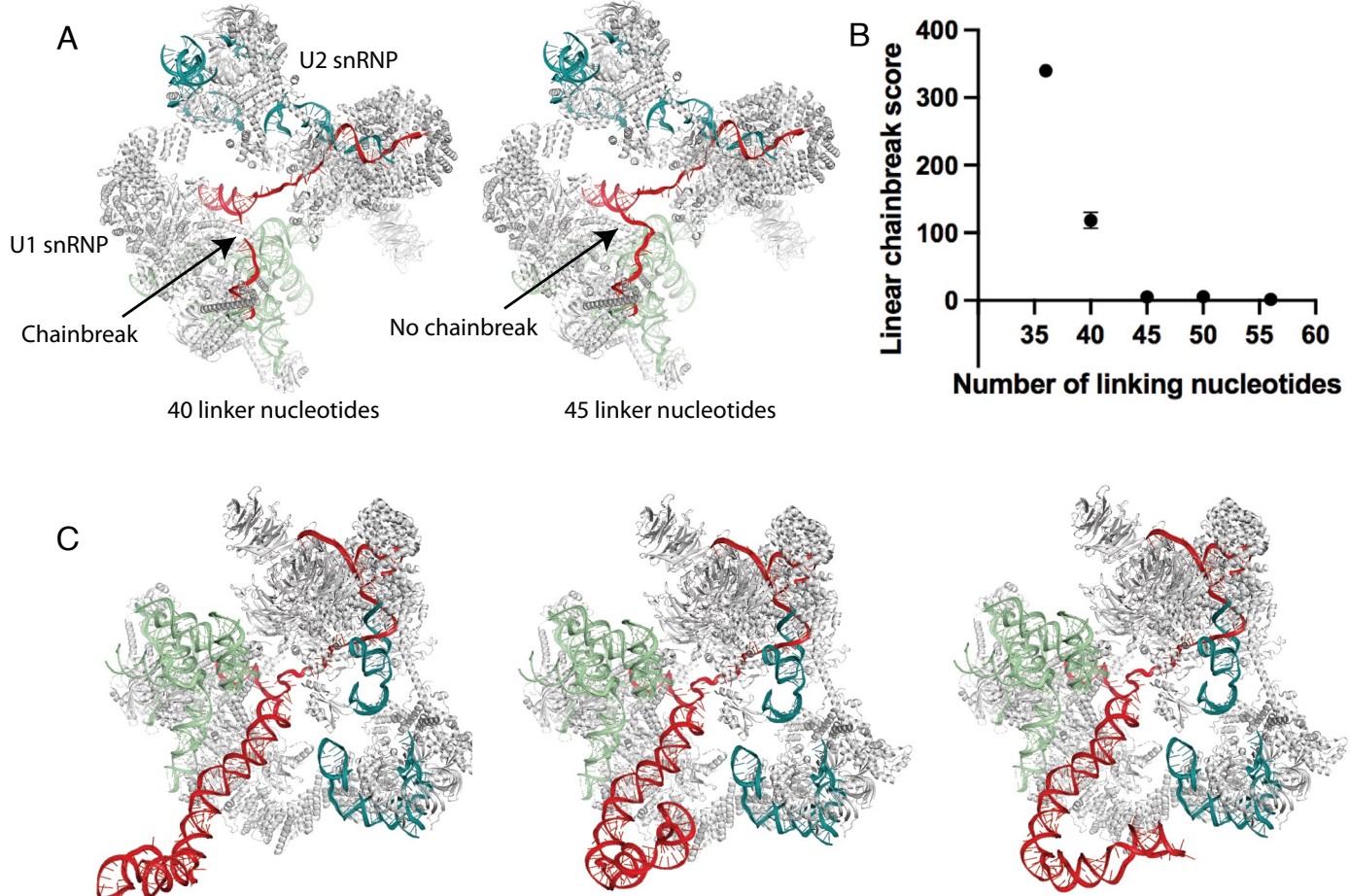

**Extended Data Fig. 4 | Structural modeling with the A state spliceosome.**
**A)** Sample Rosetta models of introns with varying linker lengths to identify linker lengths for zipper stems compatible with the A state spliceosome structure. **B)** Penalty for chain breaks as modeled linker length increases. Data are presented as mean values +/- standard deviation. **C)** Top 3 models for *RPL36B* intron in the context of the A state spliceosome, with the *RPL36B* intron secondary structure specified from DMS-MaPseq.

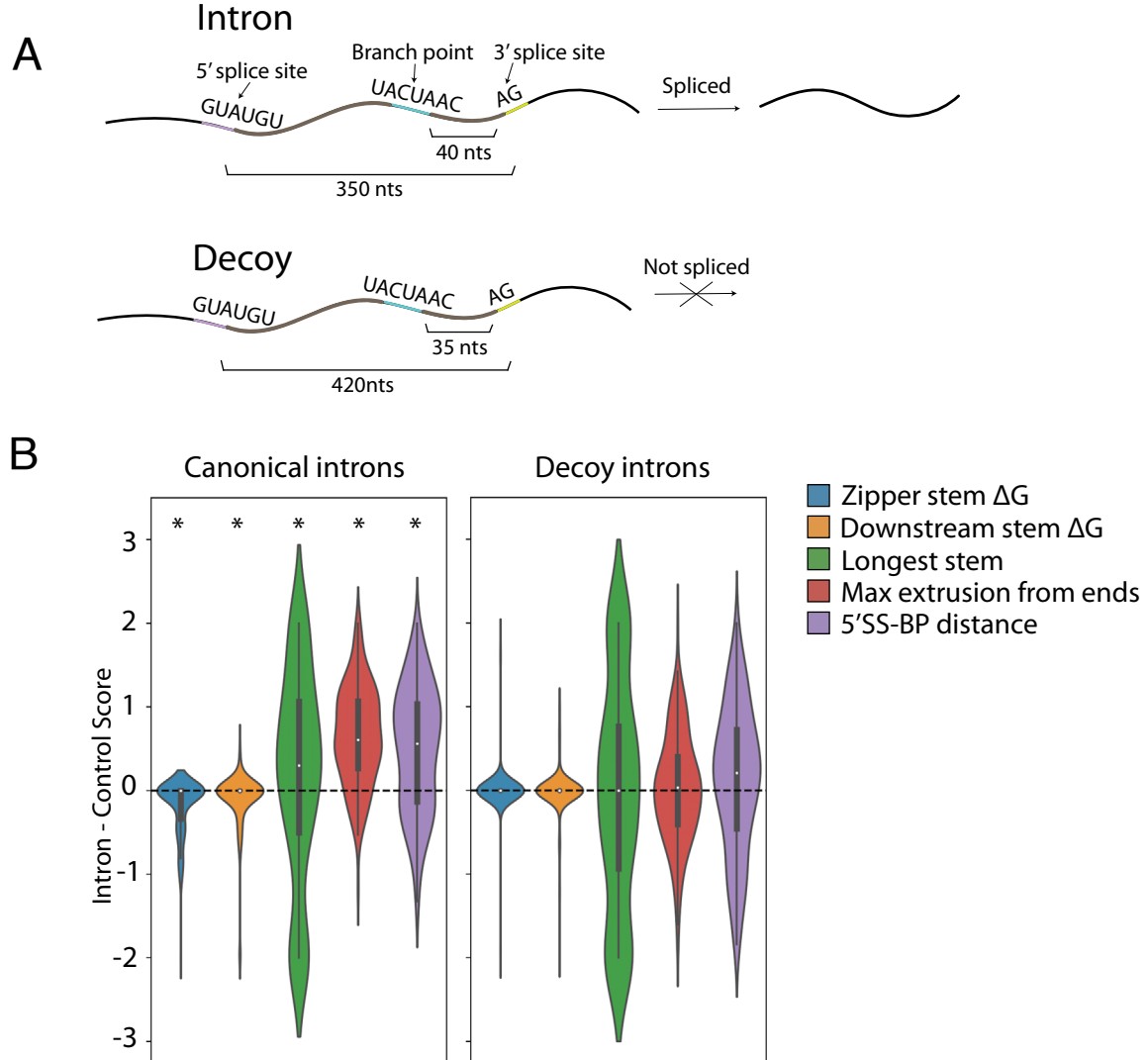

**Extended Data Fig. 5 | Secondary structure features for introns and non-splicing decoy sequences. A)** Intron and decoy sequence schematic. Nucleotide lengths depicted are representative lengths for introns and decoy sequences, with decoy sequences chosen to have 5' splice site, branch point, and 3' splice site placements matching length distributions from canonical introns (see Methods). **B)** Secondary structure features are enriched in standard canonical spliced introns in yeast, but not in decoy sequences (genomic intervals which match splice site sequences and yet do not splice). *p-value < 0.01 by two-sided Wilcoxon ranked-sum test. Left: N = 140 canonical introns were compared to controls, yielding p-values (left to right): <1E-4, 0.00027, 0.00072, <1E-4, <1E-4. Right: N = 167 decoy introns were compared to controls, yielding p-values (left to right): 0.74, 0.23, 0.68, 0.37, 0.10. Embedded box plots mark the median as the center white point and include a box from the 25th (Q1) to 75th (Q3) percentile, extending whiskers to the smallest and largest value that fall within 1.5 times the interquartile range below Q1 and above Q3.

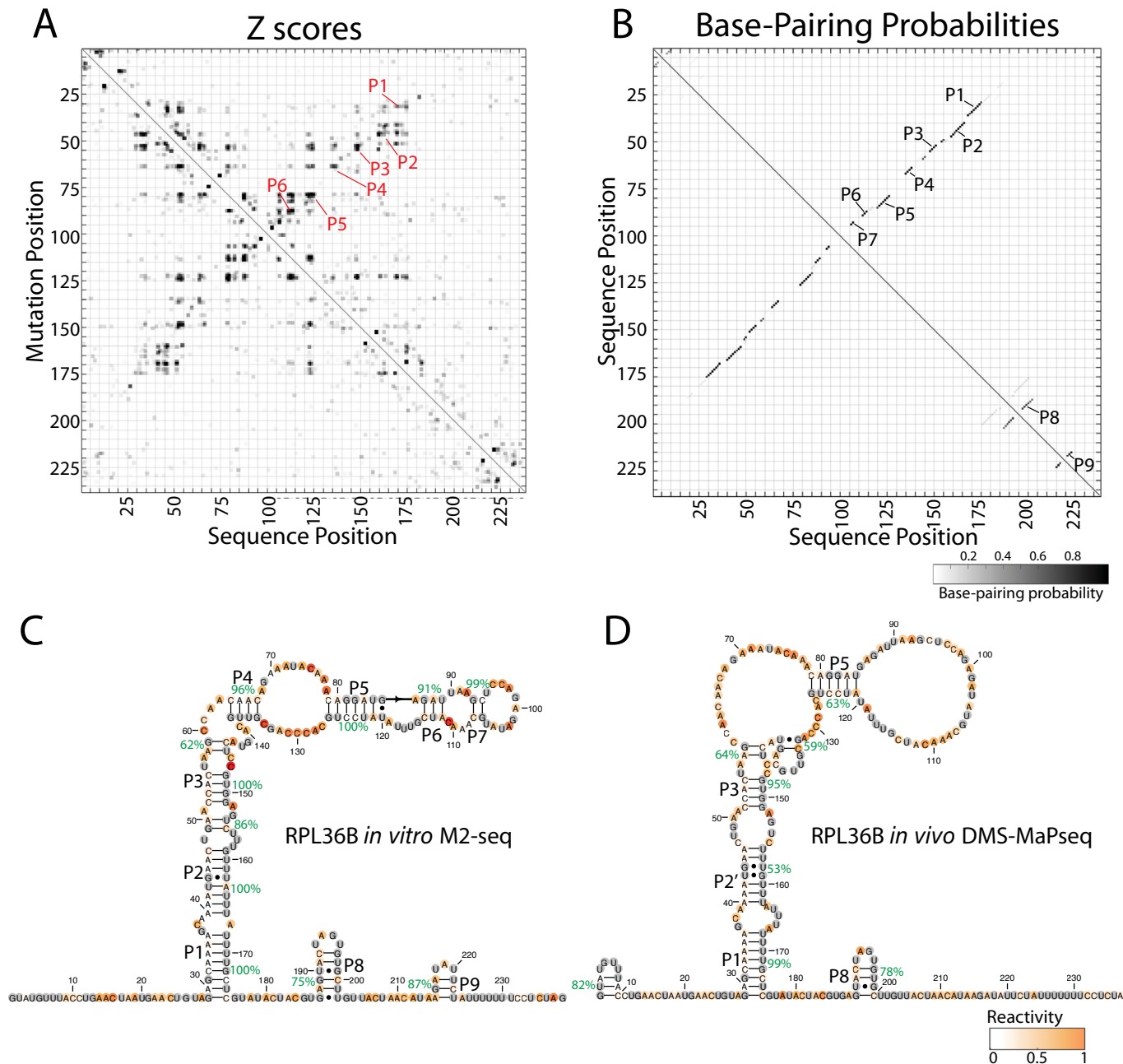

**Extended Data Fig. 6 | Multidimensional chemical mapping for *RPL36B*. A)** In vitro M2-seq Z scores for the intron in *RPL36B*, with peaks representing helices annotated in red. **B)** In vitro chemical reactivity base-pairing probabilities for

RPL36B using 1D and 2D chemical reactivity from M2-seq. Secondary structure predictions guided by 1D and 2D DMS probing data for the intron in *RPL36B* **C)** from in vitro M2-seq, and **D)** from in vivo DMS-MaPseq.

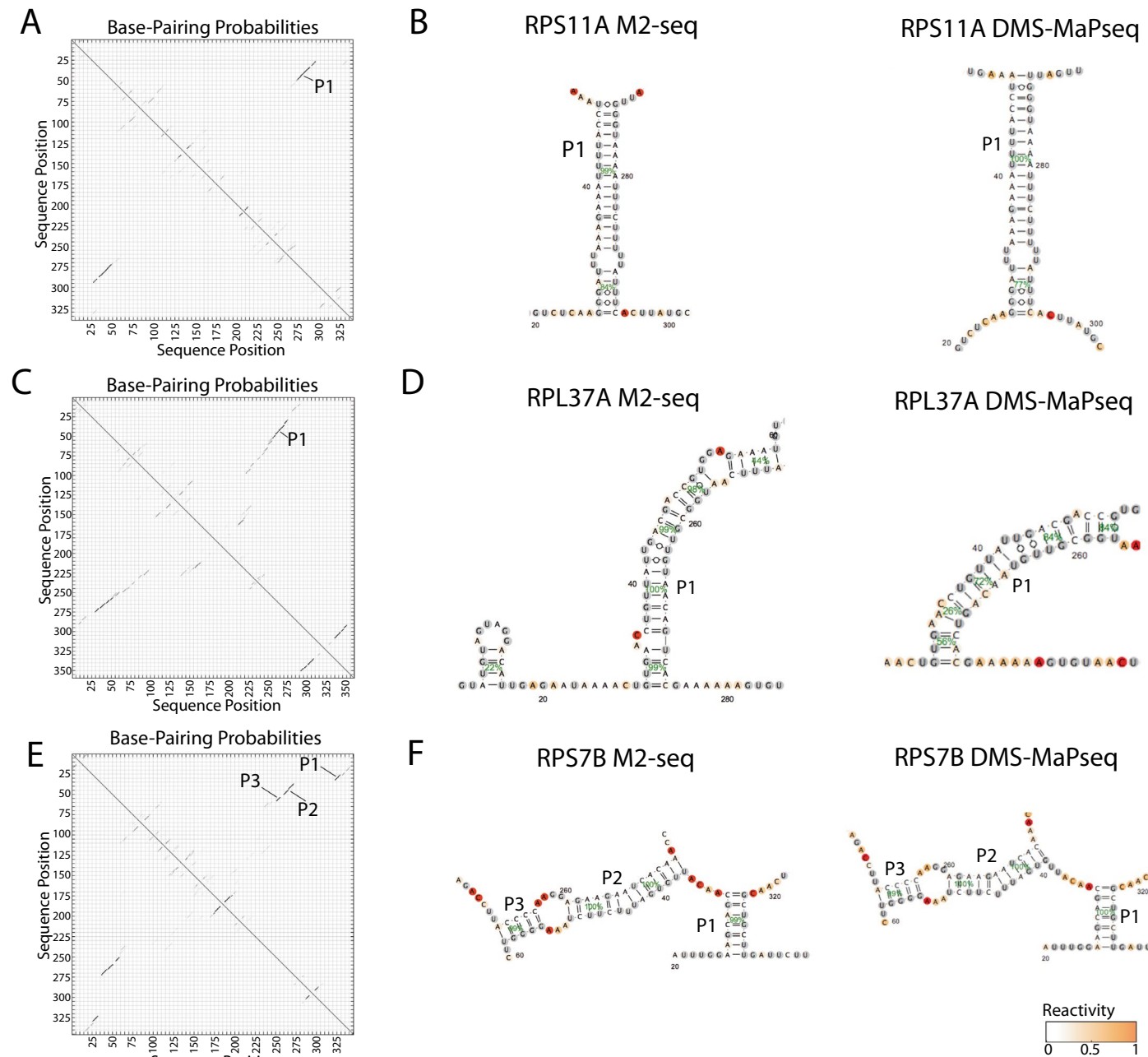

**Extended Data Fig. 7 | Base-pairing probabilities and secondary structure predictions from in vitro M2-seq and in vivo DMS-MaPseq.** Base-pairing probabilities and secondary structure predictions are shown for introns in *RPS11A* (**A, B**), *RPL37A* (**C, D**), *RPS7B* (**E, F**). (**B, D, F**) Secondary structures are guided by 1D and 2D DMS probing data from M2-seq (left) or by 1D DMS probing data from in vivo DMS-MaPseq (right).

## A

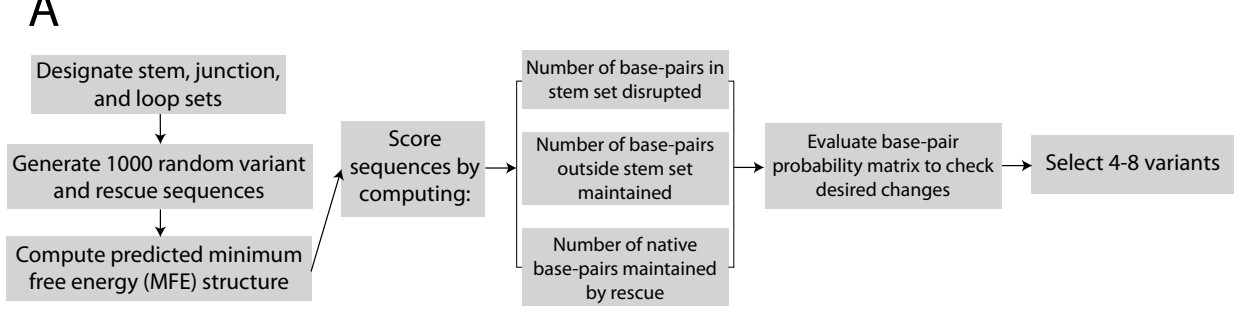

## B

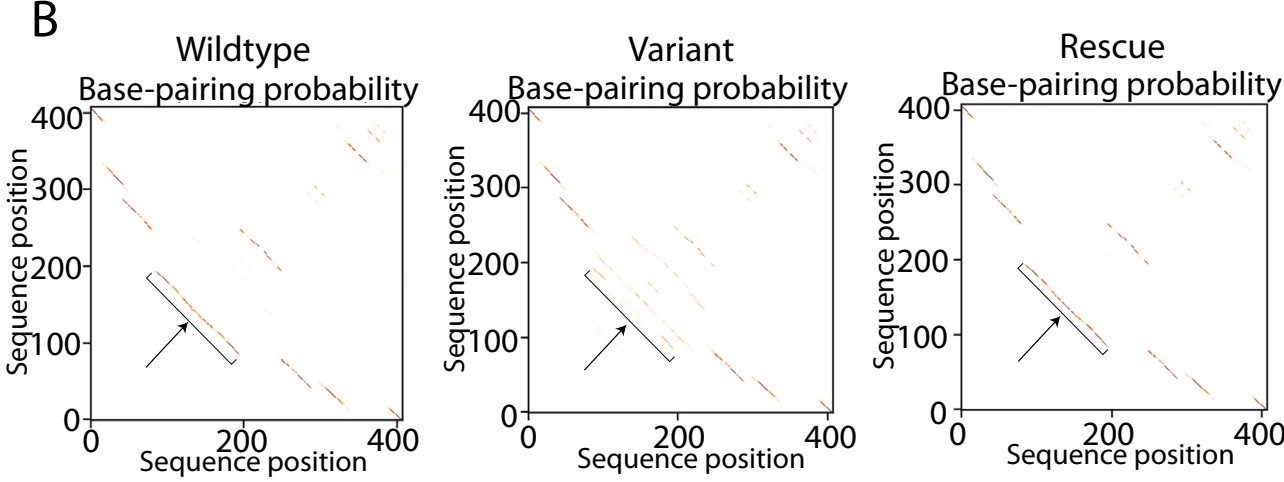

## C

Construct for landing pad transformation:
Transform BY4741 with SD-HIS selection to make LPL001

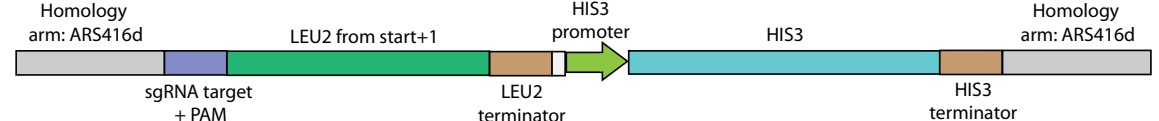

Construct for intron library transformation:
Transform LPL001 with SD-LEU selection to make yeast library

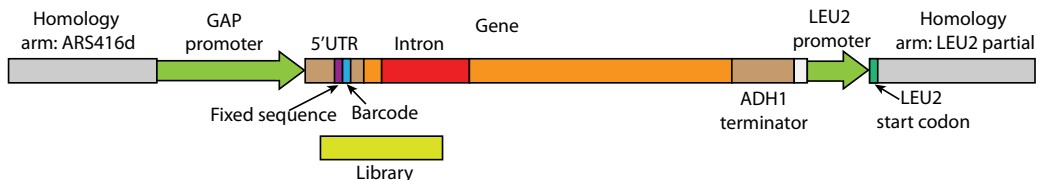

## D

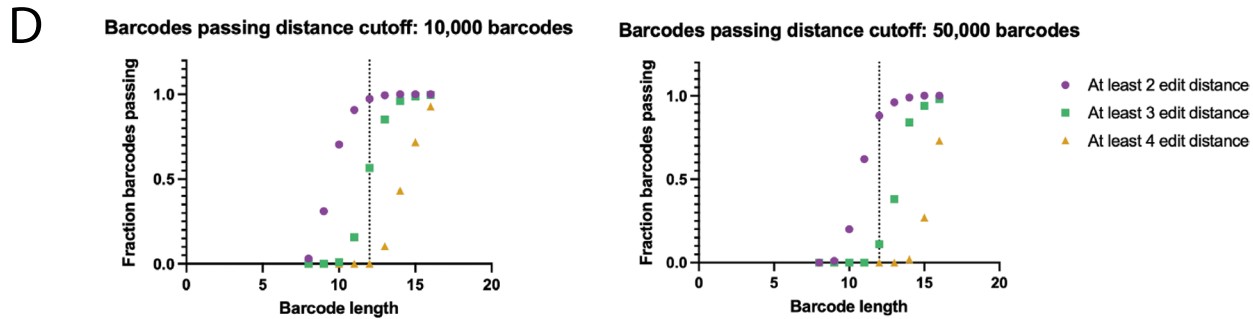

**Extended Data Fig. 8 | See next page for caption.**

**Extended Data Fig. 8 | Structure variant library design to measure spliced and unspliced RNA levels for intron variants. A)** Computational pipeline for designing variant and rescue sequences to assess stem and loop sets in an intron. **B)** Sample base-pair probability matrix comparison between a set of wildtype (left), variant (middle), and rescue (right) sequences. The stem set targeted by this variant and rescue sequence are bracketed. **C)** Constructs used for constructing the landing pad strain LPL001 (top) and for transforming the intron library via genomic integration into LPL001 (bottom). **D)** Based on simulations sampling 10,000 or 50,000 random barcodes, the number of barcodes within 2, 3, or 4 edit distance of another barcode. The vertical line indicates the chosen barcode length (12).

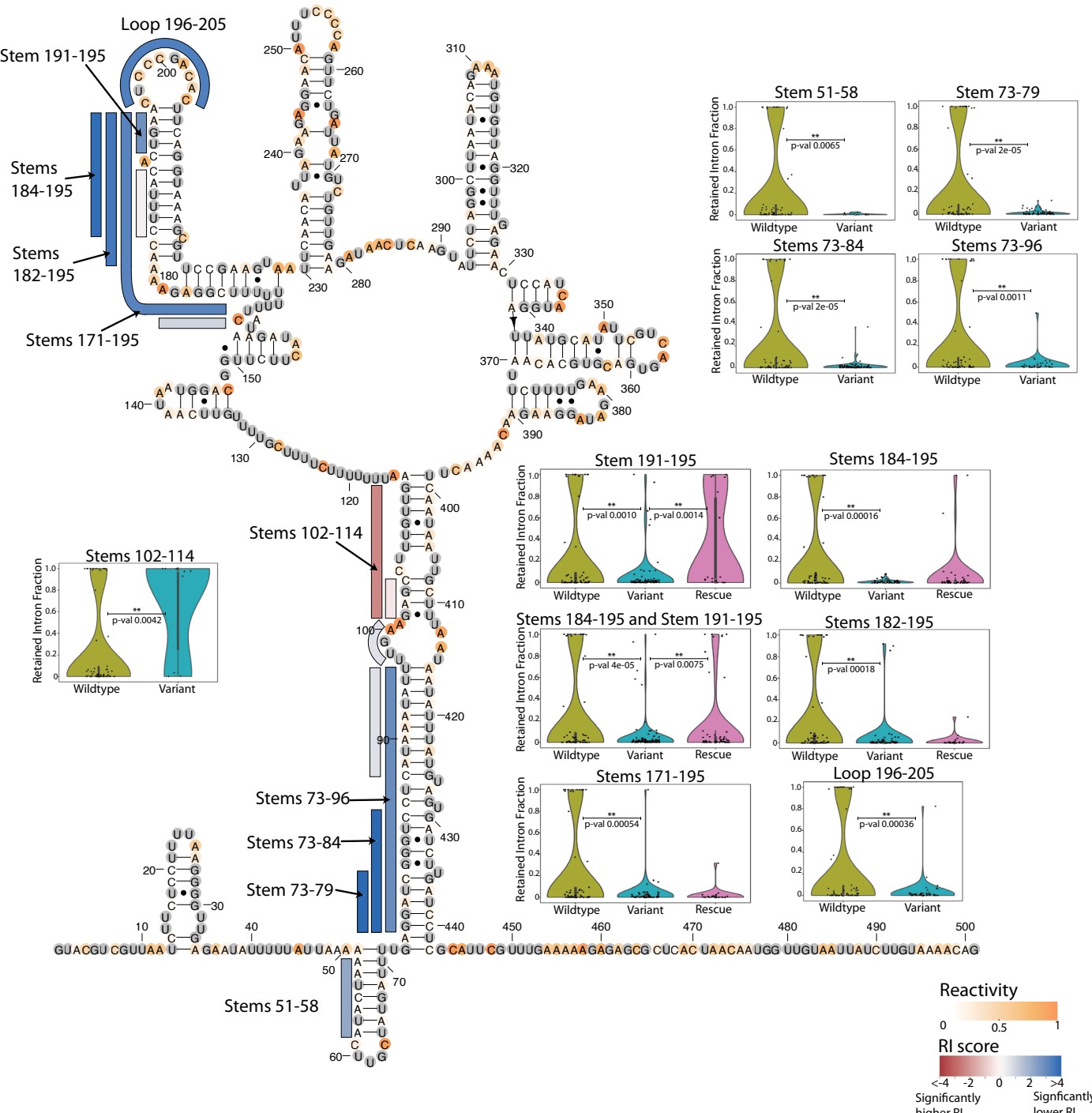

**Extended Data Fig. 9 | Effects of structure variants on retained intron levels for _RPS9A_ variants.** For a given stem set or loop, violin plots depict data for the wildtype sequence and all variant sequences, with points for each unique barcode. Data for rescue sequences are shown when included in the intron library. p-values (computed by two-sided permutation tests) are indicated for comparisons between wildtype and variant sequences, and between variant and rescue sequences. There are 67 wildtype sequences compared to variant sequences in each plot. The numbers of variant sequences are as follows: 12 for stem 51-58, 75 for stem 73-79, 11 for stems 73-84, 39 for stems 73-96, 10 for stems 102-114, 81 for stem 191-195, 44 for stems 184-195, 60 for stems 182-195, 74 for stems 171-195, and 51 for loop 196-205. The numbers of rescue sequences are as follows: 18 for stem 191-195, 46 for stems 184-195, 25 for stems 182-195, and 24 for

stems 171-195. Embedded box plots mark the median as the center white point and include a box from the 25th (Q1) to 75th (Q3) percentile, extending whiskers to the smallest and largest value that fall within 1.5 times the interquartile range below Q1 and above Q3. Secondary structures are colored by reactivity data, and bars alongside the secondary structure indicate stem and loop disruption sets, with each bar representing variant sequences mutating nucleotides across the full extent of the bar. These bars are colored by the retained intron (RI) score for the interval. The RI score is the negative log(p-value) comparing RI values between wildtype and variant sequences, and the sign indicates the effect direction, with positive values (shown as blue) for lower variant RI compared to wildtype, and negative values (shown as red) for higher variant RI.

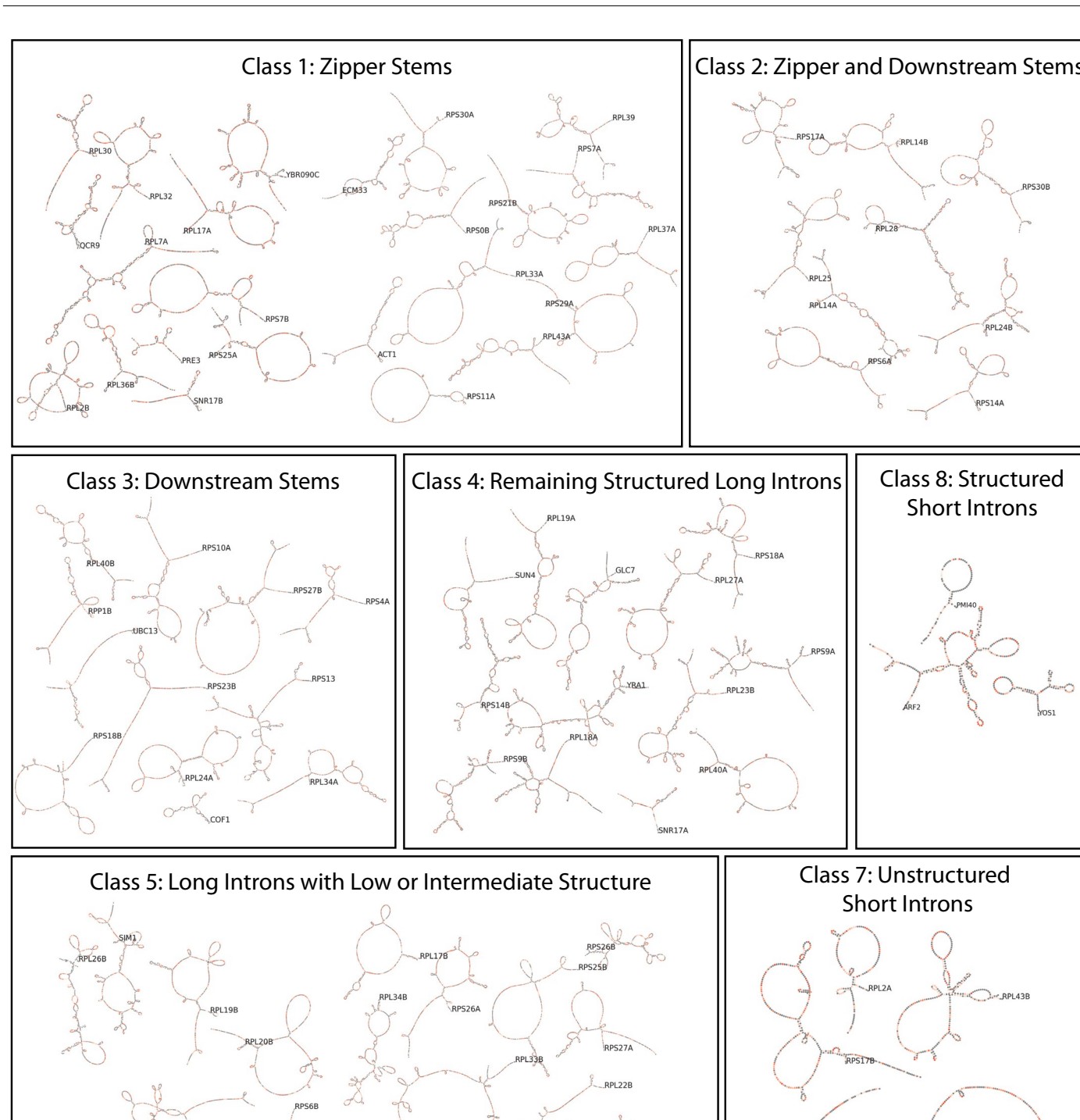

**Extended Data Fig. 10 | One-page overview of secondary structures and overlaid DMS reactivity profiles for all introns with sufficient coverage from DMS-MaPseq.** Introns are grouped into classes from hierarchical clustering (Fig. 5). Secondary structures are depicted schematically using RiboGraphViz (https://github.com/DasLab/RiboGraphViz).

# Reporting Summary

## Statistics

For all statistical analyses, confirm that the following items are present in the figure legend, table legend, main text, or Methods section.

| n/a | Confirmed | |
|---|---|---|
| ☐ | ☒ | The exact sample size ($n$) for each experimental group/condition, given as a discrete number and unit of measurement |
| ☒ | ☐ | A statement on whether measurements were taken from distinct samples or whether the same sample was measured repeatedly |
| ☐ | ☒ | The statistical test(s) used AND whether they are one- or two-sided *Only common tests should be described solely by name; describe more complex techniques in the Methods section.* |
| ☒ | ☐ | A description of all covariates tested |
| ☒ | ☐ | A description of any assumptions or corrections, such as tests of normality and adjustment for multiple comparisons |
| ☐ | ☒ | A full description of the statistical parameters including central tendency (e.g. means) or other basic estimates (e.g. regression coefficient) AND variation (e.g. standard deviation) or associated estimates of uncertainty (e.g. confidence intervals) |
| ☐ | ☒ | For null hypothesis testing, the test statistic (e.g. $F$, $t$, $r$) with confidence intervals, effect sizes, degrees of freedom and $P$ value noted *Give P values as exact values whenever suitable.* |
| ☒ | ☐ | For Bayesian analysis, information on the choice of priors and Markov chain Monte Carlo settings |
| ☒ | ☐ | For hierarchical and complex designs, identification of the appropriate level for tests and full reporting of outcomes |
| ☐ | ☒ | Estimates of effect sizes (e.g. Cohen's $d$, Pearson's $r$), indicating how they were calculated |

*Our web collection on statistics for biologists contains articles on many of the points above.*

## Software and code

Policy information about availability of computer code

| | |
|---|---|
| Data collection | No software was used for data collection. |
| Data analysis | Sequencing data processing pipelines are detailed at the GEO accession GSE209857. The following software was used for sequencing data analysis: bowtie2 (v2.3.4.1), cutadapt (v1.18), UMI-tools (v1.1.1), samtools (v1.16.1), fgbio (v2.0.2), gmap (2015-12-31), TopHat (v2.1.1), Biers (v1.2), RNAframework (v2.6.9), and ShapeMapper (v1.2). The following tools were used for structure prediction and structure analysis: Rosetta (v3.5), RNAstructure (v6.1), Vienna (v2.4.13), Contrafold (v2.00), DREEM (v0.0.2), Arnie (v0.2.6), and R-scape (v1.6.0). RiboDraw (v1.0.0) and RiboGraphViz (https://github.com/DasLab/RiboGraphViz) were used for structure visualization. Structural feature calculation, statistical comparisons, and evolutionary analyses are included at https://github.com/ramyarangan/pre-mRNA_Secstruct. Code for designing VARS-seq sequences and analyzing gDNA and RNA sequencing data from VARS-seq is included in https://github.com/ramyarangan/VARSseq. Analysis of intron evolution and computational predictions for intron secondary structures and structural features were performed using the code in: https://github.com/ramyarangan/pre-mRNA_Secstruct. |

For manuscripts utilizing custom algorithms or software that are central to the research but not yet described in published literature, software must be made available to editors and reviewers. We strongly encourage code deposition in a community repository (e.g. GitHub). See the Nature Portfolio guidelines for submitting code & software for further information.

## Data

Policy information about availability of data

All manuscripts must include a data availability statement. This statement should provide the following information, where applicable:
- Accession codes, unique identifiers, or web links for publicly available datasets
- A description of any restrictions on data availability
- For clinical datasets or third party data, please ensure that the statement adheres to our policy

All sequencing data are available at the Gene Expression Omnibus (GEO) accession number GSE209857. DMS-derived secondary structures are included in Table S1. Processed DMS reactivity data, control RNA secondary structures, and base-pairing probabilities from structure prediction with bootstrapping are included in https://github.com/ramyarangan/DMS_intron_analysis. VARS-seq intron variant sequences along with spliced and unspliced read counts are included in Tables S3-S4. PDB IDs 6G90, 4V88, 6AGB, and 6N7R were used for analysis. We compared our data with prior DMS-MaPseq data included at GEO accession GSE84537 (sample SRX1959209, run number SRR3929621). We additionally used annotations from the Saccharomyces Genome Database, S. cerevisiae snoRNA database, and the YeasTSS database.

## Research involving human participants, their data, or biological material

Policy information about studies with human participants or human data. See also policy information about sex, gender (identity/presentation), and sexual orientation and race, ethnicity and racism.

| | |
|---|---|
| Reporting on sex and gender | N/A |
| Reporting on race, ethnicity, or other socially relevant groupings | N/A |
| Population characteristics | N/A |
| Recruitment | N/A |
| Ethics oversight | N/A |

Note that full information on the approval of the study protocol must also be provided in the manuscript.

# Field-specific reporting

Please select the one below that is the best fit for your research. If you are not sure, read the appropriate sections before making your selection.

☒ Life sciences ☐ Behavioural & social sciences ☐ Ecological, evolutionary & environmental sciences

For a reference copy of the document with all sections, see nature.com/documents/nr-reporting-summary-flat.pdf

# Life sciences study design

All studies must disclose on these points even when the disclosure is negative.

| | |
|---|---|
| Sample size | Structure probing experiments were carried out in two replicates per condition, aligning with prior standards in the field. |
| Data exclusions | No data were excluded. |
| Replication | For all transcriptome-wide structure probing experiments (in vivo, in vitro refolded, and denatured conditions) along with targeted DMS-MaPseq, we performed experiments with two replicates to confirm replicable structure profiles. Biological replicates were additionally performed for RT-qPCR experiments. In all cases, replicate experiments were performed by growing overnight cultures from independent yeast clones, with separate cell treatment and RNA extraction. All attempts at replicating experiments were performed successfully. |
| Randomization | Randomization was not relevant to this study design, as experiments were conducted where treatments were applied to some sequences but not others. |
| Blinding | Blinding was not relevant to this study design, as we did not expect that bias would influence outcomes of experiments. |

# Reporting for specific materials, systems and methods

We require information from authors about some types of materials, experimental systems and methods used in many studies. Here, indicate whether each material, system or method listed is relevant to your study. If you are not sure if a list item applies to your research, read the appropriate section before selecting a response.

## Materials & experimental systems

| n/a | Involved in the study |
|-----|-----------------------|
| ☒ ☐ | Antibodies |
| ☒ ☐ | Eukaryotic cell lines |
| ☒ ☐ | Palaeontology and archaeology |
| ☒ ☐ | Animals and other organisms |
| ☒ ☐ | Clinical data |
| ☒ ☐ | Dual use research of concern |
| ☒ ☐ | Plants |

## Methods

| n/a | Involved in the study |
|-----|-----------------------|
| ☒ ☐ | ChIP-seq |
| ☒ ☐ | Flow cytometry |
| ☒ ☐ | MRI-based neuroimaging |

## Plants

| | |
|---|---|
| Seed stocks | *Report on the source of all seed stocks or other plant material used. If applicable, state the seed stock centre and catalogue number. If plant specimens were collected from the field, describe the collection location, date and sampling procedures.* |
| Novel plant genotypes | *Describe the methods by which all novel plant genotypes were produced. This includes those generated by transgenic approaches, gene editing, chemical/radiation-based mutagenesis and hybridization. For transgenic lines, describe the transformation method, the number of independent lines analyzed and the generation upon which experiments were performed. For gene-edited lines, describe the editor used, the endogenous sequence targeted for editing, the targeting guide RNA sequence (if applicable) and how the editor was applied.* |
| Authentication | *Describe any authentication procedures for each seed stock used or novel genotype generated. Describe any experiments used to assess the effect of a mutation and, where applicable, how potential secondary effects (e.g. second site T-DNA insertions, mosaicism, off-target gene editing) were examined.* |

