## [Peer Review File · Nature Structural & Molecular Biology]

Comprehensive analysis of *S. cerevisiae* intron structures *in vivo*

Corresponding Author: Professor Rhiju Das

Version 0:

Decision Letter:

30th Aug 2023

Dear Dr. Das,

Thank you again for submitting your manuscript "RNA structure landscape of *S. cerevisiae* introns". We now have comments (below) from the 3 reviewers who evaluated your paper. In light of these reports, we remain interested in your study and would like to see your response to the comments of the referees, in the form of a revised manuscript.

You will see that all referees appreciate the novelty of the findings, the importance of novel functional assays such as VARS-seq, and the potential for follow-up studies that this work presents. However, the experts raise important issues that need to be addressed for a successful peer-review process. More specifically, reviewer #2 expresses technical concerns on multiple levels (reproducibility of data, sensitivity of VARS-seq and limitations due to high variability between barcodes, accordance of *in vitro* and *in vivo* findings, etc) that must be addressed in their entirety. In addition, all experts voice concerns, though to differing extent and with independent guidelines, about the mechanistic findings, which we editorially agree that would significantly boost the study, if fully addressed. Finally, the referees also request at places clarifications, additional discussion, and contextualising the novel findings within existing literature.

Please be sure to address/respond to all concerns of the referees in full in a point-by-point response and highlight all changes in the revised manuscript text file.

We appreciate the requested revisions are extensive. We thus expect to see your revised manuscript within 6 months. If you cannot send it within this time, please let us know. We will be happy to consider your revision as long as nothing similar has been accepted for publication at NSMB or published elsewhere. Should your manuscript be substantially delayed without notifying us in advance and your article is eventually published, the received date would be that of the revised, not the original, version.

Reporting Summary:

When submitting the revised version of your manuscript, please pay close attention to our <https://www.nature.com/nature-portfolio/editorial-policies/image-integrity> Digital Image Integrity Guidelines and to the following points below:

Finally, please ensure that you retain unprocessed data and metadata files after publication, ideally archiving data in

perpetuity, as these may be requested during the peer review and production process or after publication if any issues arise.

We require deposition of coordinates (and, in the case of crystal structures, structure factors) into the Protein Data Bank with the designation of immediate release upon publication (HPUB). Electron microscopy-derived density maps and coordinate data must be deposited in EMDB and released upon publication. Deposition and immediate release of NMR chemical shift assignments are highly encouraged. Deposition of deep sequencing and microarray data is mandatory, and the datasets must be released prior to or upon publication. To avoid delays in publication, dataset accession numbers must be supplied with the final accepted manuscript and appropriate release dates must be indicated at the galley proof stage. Please find the complete NRG policies on data availability at <http://www.nature.com/authors/policies/availability.html>.

Link Redacted

Sincerely,

Dimitris Typas
Associate Editor
Nature Structural & Molecular Biology
ORCID: 0000-0002-8737-1319

Referee expertise:

Referee #1: RNA processing/splicing in cerevisiae

Referee #2: RNA structural biology

Referee #3: RNA structural biology

Reviewers' Comments:

Reviewer #1:

Remarks to the Author:

In their manuscript, Rangan et al. carry out a comprehensive analysis of pre-mRNA intron structure in *S. cerevisiae* using in vivo probing and test the functional impact of several of these with respect to splicing. This work is important for several reasons. First, the dogma in the field is that these structures are present because they facilitate splicing by decreasing 3 dimensional distances between splice sites. Rangan and coworkers show that this is not true—these structures can either enhance or inhibit splicing (or RNA decay, see below). Second, they have essentially provided a resource of well-characterized RNA structures that form in vivo that can be combined with the awesome power of yeast molecular genetics to assess structure/function relationships. I suspect many more papers to result from this in the future! Third, these structures may themselves be the source of new RNA structural motifs, which could be important for expanding the RNA structural repertoire and our ability to predict these. Fourth, they have developed new tools (VARS-Seq) and made new observations (the intron structures also form in vitro) that will likely spur future research in other areas related to RNA structure or in vitro

mechanistic analysis of pre-mRNA splicing.

This work will be of high interest to anyone studying mechanisms of pre-mRNA splicing, regulation of gene expression (especially in yeast) and RNA structure. The experiments are expertly designed, conducted, and analyzed and the results presented clearly. This is really an outstanding piece of science.

The main criticism of the work is that the mechanistic conclusions, other than RNA structure in introns not being “neutral”, are a bit nebulous. As a result, the work is more of a “resource” for the community (but a terrific one!) than one that reveals a new mechanistic paradigm. In some sense this is function of the system—the structures appear to have idiosyncratic effects. It would be interesting to pin down, at least for one case, the molecular mechanism by which the structure is enhancing or inhibiting splicing (or RNA decay). Based on this, I have a few comments or suggestions that the authors could consider at their discretion prior to publication that may be able to increase the mechanistic impact.

1. The authors rightly conclude that changes in RI could be due to changes in splicing OR RNA decay. Can this be tested? Perhaps a UPF1 deletion strain or something similar could be used to answer this more definitively.
2. There is potentially an interesting connection between the structure of the intron and the 3 dimensional structure of the spliceosome. Can the authors predict effects on splicing based on whether or not the intron stem is compatible with *published* structures of the spliceosome?
3. It occurred to me that one function of these stems could be to occlude cryptic splice sites. For example, there is a nonconsensus BS at the base of the stem in RPS23B (UAUUAAU). I'm sure there are others. Perhaps the recent paper from the Chanfreau lab on Biorxiv on activation of cryptic 3' SS could prove useful.
4. It is interesting to note that introns apparently have more structure than mRNAs. This made me wonder about the translational consequences of these structures. Perhaps this is a NMD “trigger” and/or produces ribosome collisions?

Minor Points:

1. Do tRNA introns (the longer ones) have any interesting structures? Or are these specific to spliceosomal-processed introns?
2. Can the authors comment on why some introns (11?) are apparently more resistant to pladB inhibition than others? I didn't see a list of these, it would be nice to know what these introns are.
3. Some intron structures have also been confirmed by cryo-EM structures of spliceosomes assembled on those introns. I don't think those are included in the references on lines 196-197.
4. Do any of the VAR-Seq variants introduce new splice sites that are used by the spliceosome?
5. No scale for DMS reactivity in Figures S2 and S3
6. -Parenthesis where there should not be, p. 4 “...using reactivity data to make DMS-guided structure predictions) with a helix confidence estimate...”
7. -p. 6, 2nd paragraph under “New structures found by probing *S. cerevisiae* pre-mRNA” Fig. 2E is referenced when it should be Fig. 2B for RPS23B
8. -Fig S10A: Are the nucleotide lengths given the averages of all used?
9. -Fig S18D: one plot not labeled with intron interval it is for

Reviewer #2:

Remarks to the Author:

This study examines the role of pre-mRNA secondary structures in RNA processing pathways, using the yeast species *S. cerevisiae*. The research used a technique called dimethyl sulfate probing to examine these structures, finding widespread formations known as “zipper stems” and “downstream stems”, and identified new long stems that differentiate pre-mRNA from spliced mRNA. A new method called VARS-seq was developed to analyze variations in RNA structure. Interestingly, the study suggests that some structured elements could increase levels of spliced mRNA despite not being close to canonical splice sites, while others, like zipper stems, can increase levels of retained introns. This study brings forth new perspectives on how pre-mRNA folding affects gene expression.

Overall, I find this manuscript interesting with broad implications for splicing regulation, but the quality of the current data is not high enough to support the conclusions. The authors also present contradictory data that go against their conclusions (Fig.S11) as discussed in more detailed below.

Major concerns:

1. It is unclear what happens during 1h of pladB treatment, are some RNAs degrading? What is the half-life of the RNA? Could this readout be measuring the state of splicing or degrading some RNAs preferentially?
2. My biggest concern is data reproducibility.
 - a. Starting off, when discussing reproducibility, the use of the r^2 value, which provides clear information about the proportion of the signal distribution that is consistent across experiments, is more informative and intuitive. I highly recommend that you report all values as r^2 . Although there is no universally accepted threshold for categorizing results as “same” versus “different”, the current cutoff set by you at $R=0.5$, which equates to $r^2=0.25$, seems too low. This implies that the two patterns are 25% similar and 75% different, which is not a strong enough correlation to be considered “the same”. I suggest increasing this cutoff to at least $r^2=0.8$. With the current cutoff, the two signals might yield completely different structures,

which could confound the interpretation of your results.

b. It's important to understand the cause of the poor reproducibility seen in many introns. Is it because of their low abundance, leading to a limited number of RNA molecules being evaluated? Or does it stem from inadequate sequencing coverage? Or could it possibly be a combination of both? Supplementary Figure S2 indicates a strong correlation with coverage. If increasing sequencing coverage is not technically feasible, the authors should consider excluding data that exhibits low coverage and reproducibility. By focusing on high-quality data, the conclusions drawn from the study will be more reliable and impactful.

c. Regarding Figure 1F, it's evident that the patterns between the pladB treated and untreated samples differ, leading to distinct structural models as demonstrated in Figure S4. The authors should explicitly acknowledge the observed changes associated with the addition of pladB. It's concerning to see such significant differences induced by pladB. I'm uncertain about how the authors are able to determine the proportion of helices that remain unchanged, especially when they only have sufficient coverage for a limited number of introns without pladB.

3. The data shown in Figure S11, depicting strikingly different structures in vivo and in vitro, contradicts the authors' assertion in the abstract stating that "intron structures can form in vitro without the presence of binding partners". There appears to be a discrepancy in the presented data and its explanation, which is not adequately addressed. The authors' hypothesis that "introns could be unfolded during in vitro refolded total RNA due to intermolecular interactions" seems contradictory. Intermolecular interactions would presumably reduce DMS accessibility and lead to the appearance of more structured RNAs. As the results of this experiment deviates considerably from the rest of the manuscript, further testing and explanation are needed, especially since it challenges a key point of the manuscript.

4. Figure 6, focusing on the Var-seq functional assay, shows that the presence of different barcodes leads to the greatest variation in RIFraction. The impact of the barcode, which should ideally be minimal, appears bigger than the difference between the Wildtype and the Variant. This effect needs to be understood and minimized before making comparisons. The authors should also provide information about:

- a. The actual effect size in terms of fold change in mRNA expression or RI.
- b. The consistency of barcode distributions. A few outlier barcodes could potentially shift the overall distribution significantly.

5. For validation purposes, it would be more convincing to provide individual well-characterized examples with fold change and effect size of structure on function, even if it's just one for each, to support the discovery of new regulatory mechanisms. Specifically, the findings that certain structured elements can increase spliced mRNA levels despite their distance from canonical splice sites, and others, like zipper stems, can enhance levels of retained introns.

Reviewer #3:

Remarks to the Author:

The authors attempt to elucidate the landscape of RNA secondary structure in the yeast transcriptome. This landscape is largely unknown for any species and is important for improved understanding of the regulatory role of RNA in living systems. This problem is critically important, both from the perspective of basic research and from the perspective of mRNA therapeutic and vaccine development. Specifically, the team focuses on splicing and pre-mRNA secondary structures. Using pladB, the team is able to inhibit splicing, effectively trapping for analysis pre-mRNA secondary structures across the transcriptome. This is a key step, since without doing this, a large ensemble of spliced and unspliced transcripts exists, making it difficult to disentangle. Using chemical probing strategies developed by the authors, the team finds highly structured upstream regions ('zipper stems' towards the 5') and downstream regions ('downstream stems' towards the 3'). They find interesting differences in the structure of the mRNA between coding and noncoding regions. They perform a careful comparison between in vitro and in vivo systems and detailed phylogenetic analysis. Importantly, they develop a high throughput functional assay, where they produce variants of secondary structure elements and test the effect of these variants on splicing activity. The paper is well written and well organized. The figures are excellent and the data is compelling. There is an excellent supplementary information section. The results have broad and general implications for life sciences as a whole, for the readership of NSMB and beyond. The results should be published without delay.

Comments:

1. The hypothesis that 'introns could be unfolded in vitro and refolded total RNA due to intermolecular interactions in the complex RNA mixture' lacks support and is unclear. It seems more probable that non-equilibrium effects, kinetic effects, and environmental perturbations to the free energy landscape are more likely culprits in remodeling the folding landscape.

2. For the 'zipper stem and downstream stem free energy', a brief discussion of the limitations of the free energy calculations should be included, especially in terms of how the entropic contribution is captured, the importance of sampling 3D configuration space, and the importance of magnesium.

3. While R-scape has merits, the limitations of R-scape could also be discussed (e.g., Tavares, et al., 2019).

4. More details should be included in the description of the 3D models. For example, 'we built models for variants'. How were the models 'built'? Which software/code was used, what algorithms did the code use to do the building? How were ends of fragments handled? How were conformations determined? If multiple options model options (there a models that can fit the

given constraints), how were the choices made? How were the nucleotides 'replaced'? Which alignment algorithm was used? How was the alignment performed? How was the 'docking' performed? Which docking algorithm was used? Why this algorithm and why not a different algorithm? What are the limitations of using a rigid body approximation? It looks like electrostatics were neglected (or that a simple electrostatic relation was used). What are the limitations of this? What are the limitations of not included explicit solvent and explicit magnesium? How were nucleotides 'replaced by corresponding sequences'? Since you don't have an experimentally determined 3D structure for these sequences, what are the limitations of using a model?

5.It would be interesting to discuss the implications of this study of mRNA secondary structure for riboswitches and long non-coding RNAs (e.g., Novikova, et al., JMB 2013), as well as how a similar approach could be attempted in 3D (e.g., Kim, et al., Nat. Comm. 2020) and if useful information could be gained from a 3D transcriptome wide approach.

Version 1:

Decision Letter:

26th Jul 2024

Dear Professor Das,

Thank you again for submitting your manuscript "RNA structure landscape of *S. cerevisiae* introns". I am very sorry for the delay in responding, which resulted from the difficulty in obtaining suitable referee reports. As you will note, despite our efforts, we were unable to obtain a report from Reviewer #3 that had originally assessed the manuscript. Nevertheless, as expertise 2 are covered and Reviewer #3 had provided what read as a supportive review in the opening round, we can continue with the comments (below) from the 2 remaining reviewers who evaluated your paper. In light of these reports, we remain interested in your study and would like to see your response to the comments of the referees, in the form of a revised manuscript.

You will see that though Reviewer #1 signs off with a minor request, assessing that the manuscript is ready for publication, Reviewer #2 retains conceptual and technical concerns. While our editorial team deems that certain conceptual concerns (e.g. point 2 of Reviewer #2) can be overruled as work that is very interesting but probably out of scope for this manuscript, all technical issues (points 3-5) must be addressed in a way that alleviates the major concerns that Reviewer #2 retains. In addition, addressing the remaining conceptual concern (point 1) would significantly boost the chances of success of the work. Please note that we are reluctant to see multiple rounds of revisions to avoid both reviewers' and authors' fatigue, leading to diminishing returns. Thus we urge you to please fully address the indicated, remaining concerns and please be sure to respond to all concerns of the referees in full in a point-by-point response and highlight all changes in the revised manuscript text file. If you have comments that are intended for editors only, please include those in a separate cover letter.

We expect to see your revised manuscript within 3 months. If you cannot send it within this time, please contact us to discuss an extension; we would still consider your revision, provided that no similar work has been accepted for publication at NSMB or published elsewhere.

Reporting Summary:

When submitting the revised version of your manuscript, please pay close attention to our [href="https://www.nature.com/nature-portfolio/editorial-policies/image-integrity">Digital Image Integrity Guidelines. and to the following points below:](https://www.nature.com/nature-portfolio/editorial-policies/image-integrity)

Data availability: this journal strongly supports public availability of data. All data used in accepted papers should be available via a public data repository, or alternatively, as Supplementary Information. If data can only be shared on request, please explain why in your Data Availability Statement, and also in the correspondence with your editor. Please note that for some data types, deposition in a public repository is mandatory - more information on our data deposition policies and available repositories can be found below:

<https://www.nature.com/nature-research/editorial-policies/reporting-standards#availability-of-data>

Link Redacted

Sincerely,

Dimitris Typas
Senior Editor
Nature Structural & Molecular Biology
ORCID: 0000-0002-8737-1319

Reviewers' Comments:

Reviewer #1:

Remarks to the Author:

the authors have thoroughly responded to the prior reviews by additional analyses and some clarifications. I think this is a wonderful addition to the field and is ready for publication. I have just a few minor suggestions.

The observations that structures are depleted around splice sites, perhaps enriched around cryptic sites (esp. 3'SS), and are formed in a way compatible with spliceosome 3 dimensional structure are all highlights of the manuscript for those thinking about splicing mechanism. Indeed, to me it seems that substrate (pre-mRNA) structure may play just an important role in splicing fidelity as the proofreading ATPases!!! I really think this needs to be highlighted as a main text figure. If figures are limited, I would suggest moving Fig. 4 (in vitro to in vivo comparison) to supplemental and adding in a new figure based on the supplemental figure S13 (perhaps polished up a bit to make it easier to understand what is being plotted) that can be added to the main text. I think these are all important, fundamental observations.

Reviewer #2:

Remarks to the Author:

I appreciate the authors' efforts to enhance the manuscript and discussion with additional data, particularly concerning the structural changes and the induced unnatural cell states following PladB treatment. The acknowledgment of the simplifications in the 3D modeling approach ("We note that the 3D modeling approach here makes simplifying assumptions") reflects a commendable level of transparency. However, there are major concerns regarding the manuscript:

1. Chemical Probing Data and Zipper Stems: The chemical probing data presented does not convincingly support the existence of zipper stems in cells. Furthermore, the efforts to validate VAR-seq were partially successful; it was validated by RT-qPCR for one example (RPA9A, Fig. 6F) but not for another (RPL36B, Fig. S22A).
2. Functional Ambiguity: The role of zipper stems remains unclear, with the manuscript speculating that not all zipper stems enhance splicing by bringing splice sites closer ("Unexpectedly, zipper stem variants in the RPL28 intron (Fig. 6C) lowered RI, suggesting that not all zipper stems promote splicing by co-localizing splice sites"). This lack of clarity on their function adds to the speculative nature of the findings.
3. Quality and Reproducibility Concerns: The evidence for the existence of these zipper stems in cells is weak, and the data quality and reproducibility are questionable. The manuscript risks lowering the reproducibility standards within the field, which is a major concern. It is crucial for the authors to acknowledge the low reproducibility explicitly in the text and to justify the potential lowering of these standards if the manuscript is to be published.
4. Statistical Rigor: I strongly recommend that the authors report r^2 values to enhance the manuscript's statistical rigor, especially in the context of variance between replicates. Historically, different conventions for reporting correlation coefficients (such as using r instead of r^2) do not justify continuing to exclude these important values. Reporting r^2 is essential as it quantifies the proportion of variance in the dependent variable that is predictable from the independent variable, providing a clearer and more complete understanding of the data consistency across replicates.
5. Analysis of DMS Constraints: The manuscript could benefit from additional analyses, such as comparing the DMS signal from a denatured control to assess if similar stems are predicted. This could help clarify whether the high-confidence stems identified are genuinely influenced by the DMS constraints or merely reflect strong predictions by the RNA Structure Fold algorithm.

Given these concerns, it's crucial to consider whether this manuscript, as it stands, contributes significantly to the field or merely points out potential areas for future research without confirming the proposed structures.

Version 2:

Decision Letter:

Our ref: NSMB-A48001B

17th Jan 2025

Dear Professor Das,

Thank you for submitting your revised manuscript "RNA structure landscape of *S. cerevisiae* introns" (NSMB-A48001B). I apologise for the delay in returning a final decision; I am afraid that the usual delays during the winter holidays set the manuscript a bit back. Nevertheless, it has now been seen by the remaining original referee and their comments are below. That expert now finds that the paper has improved in revision. Thus, in conjunction with the support of the other reviewers in the previous rounds, we are happy to accept the manuscript in principle in Nature Structural & Molecular Biology, pending minor revisions to comply with our editorial and formatting guidelines.

To facilitate our work at this stage, it is important that we have a copy of the main text, without the figures, as a word file. If you could please send along a word version of this file as soon as possible, we would greatly appreciate it; please make sure to copy the NSMB account (cc'ed above).

Sincerely,

Dimitris Typas
Senior Editor
Nature Structural & Molecular Biology
ORCID: 0000-0002-8737-1319

Reviewer #2 (Remarks to the Author):

I appreciate the additional experiments and evidence for the zipper stems. The author did a great job in addressing my comments and demonstrated a commitment to scientific rigor. I find the work important and suitable for publication.

Version 3:

Decision Letter:

14th Apr 2025

Dear Professor Das,

We are now happy to accept your revised paper "Comprehensive analysis of *S. cerevisiae* intron structures *in vivo*" for publication as an Article in Nature Structural & Molecular Biology.

Your paper will be published online soon after we receive proof corrections and will appear in print in the next available issue. You can find out your date of online publication by contacting the production team shortly after sending your proof corrections.

Sincerely,

Dimitris Typas
Senior Editor
Nature Structural & Molecular Biology
ORCID: 0000-0002-8737-1319

Reviewer #1:

Remarks to the Author:

In their manuscript, Rangan et al. carry out a comprehensive analysis of pre-mRNA intron structure in *S. cerevisiae* using in vivo probing and test the functional impact of several of these with respect to splicing. This work is important for several reasons. First, the dogma in the field is that these structures are present because they facilitate splicing by decreasing 3 dimensional distances between splice sites. Rangan and coworkers show that this is not true—these structures can either enhance or inhibit splicing (or RNA decay, see below). Second, they have essentially provided a resource of well-characterized RNA structures that form in vivo that can be combined with the awesome power of yeast molecular genetics to assess structure/function relationships. I suspect many more papers to result from this in the future! Third, these structures may themselves be the source of new RNA structural motifs, which could be important for expanding the RNA structural repertoire and our ability to predict these. Fourth, they have developed new tools (VARs-Seq) and made new observations (the intron structures also form in vitro) that will likely spur future research in other areas related to RNA structure or in vitro mechanistic analysis of pre-mRNA splicing.

This work will be of high interest to anyone studying mechanisms of pre-mRNA splicing, regulation of gene expression (especially in yeast) and RNA structure. The experiments are expertly designed, conducted, and analyzed and the results presented clearly. This is really an outstanding piece of science.

The main criticism of the work is that the mechanistic conclusions, other than RNA structure in introns not being “neutral”, are a bit nebulous. As a result, the work is more of a “resource” for the community (but a terrific one!) than one that reveals a new mechanistic paradigm. In some sense this is function of the system—the structures appear to have idiosyncratic effects. It would be interesting to pin down, at least for one case, the molecular mechanism by which the structure is enhancing or inhibiting splicing (or RNA decay). Based on this, I have a few comments or suggestions that the authors could consider at their discretion prior to publication that may be able to increase the mechanistic impact.

We thank the reviewer for noting these contributions and suggesting strategies for further probing the mechanistic role for intron structures in *S. cerevisiae*. We have carried out new analyses demonstrating that structures are depleted near canonical splice sites and enriched around cryptic splice sites, suggesting additional functional roles for these structures.

1. The authors rightly conclude that changes in RI could be due to changes in splicing OR RNA decay. Can this be tested? Perhaps a UPF1 deletion strain or something similar could be used to answer this more definitively.

We agree that it would be interesting to understand whether splicing or RNA decay pathways are responsible for various structure mediated RI changes. However, we expect that deconvolving the role of the many relevant pathways in generating steady state mRNA levels would require substantial further experiments that are beyond the scope of this manuscript. Multiple different RNA decay pathways play a role in degrading unspliced pre-mRNA in *S. cerevisiae* (Parker, R.

Genetics 2012). For instance, unspliced pre-mRNA selectively retained in the nucleus by Mlp1 and Mlp2 can be targeted by the nuclear exosome (Galy, V, et al. Cell 2004), and two degradation mechanisms involving the nuclear exosome (a major 3' to 5' pathway and a minor 5' to 3' pathway) compete with splicing (Bousquet-Antonelli, C., Presutti, C., and Tollervey, D. Cell 2000). As noted by the reviewer, nonsense-mediated decay (NMD) mediated by UPF1 and other factors can recognize premature stop codons in introns and decay intron-containing pre-mRNA that have been exported to the cytosol (Losson, R., Lacroute, F. PNAS 1979). Additionally, some *S. cerevisiae* pre-mRNA appear protected from NMD and are degraded in the cytoplasm by 5' to 3' exonucleases independent of NMD (Hilleren, P.J., Parker, R. Molecular Cell 2003). For each variant, it would be possible in future work to obtain splicing measurements for wildtype, mutant, and rescue sequences in the context of a UPF1 deletion strain. However, this experiment would not disambiguate between effects on splicing rates and RNA decay, as we would expect that structures may continue to regulate other competing decay processes. We are excited about developing experiments that can dissect all the pathways that affect and respond to pre-mRNA structure but believe that significant methods development will be required, beyond what we are able to carry out in this study. We have added to the Discussion the following sentence: "Intron structures may additionally influence numerous pre-mRNA decay pathways, including nuclear retention followed by decay by the nuclear exosome, nonsense-mediated decay (NMD), and NMD-independent decay by cytoplasmic exonucleases."

2. There is potentially an interesting connection between the structure of the intron and the 3 dimensional structure of the spliceosome. Can the authors predict effects on splicing based on whether or not the intron stem is compatible with *published* structures of the spliceosome?

We thank the reviewer for this compelling suggestion. We have added new analyses on the connection between structures in pre-mRNA and sequence intervals compatible with spliceosome binding (Fig. S13; described in the results section "New structures found by probing *S. cerevisiae* pre-mRNA" and in the Methods section). We identified sequence intervals where secondary structures would be expected to clash with the spliceosome using published spliceosome structures across the splicing cycle. Our high confidence stem predictions were depleted in these regions, aligning with our prediction that stable structures in these regions would be incompatible with spliceosome binding and efficient splicing.

3. It occurred to me that one function of these stems could be to occlude cryptic splice sites. For example, there is a nonconsensus BS at the base of the stem in RPS23B (UAUUAAU). I'm sure there are others. Perhaps the recent paper from the Chanfreau lab on Biorxiv on activation of cryptic 3'SS could prove useful.

We have now added an analysis of cryptic splice site occlusion from our candidate structures to the manuscript (Fig. S13; described in the results section "New structures found by probing *S. cerevisiae* pre-mRNA" and in the Methods section). We have identified cryptic 5' splice site, branchpoint, and 3' splice site sequences by scanning relevant intervals in pre-mRNA for matches to consensus sequences from known standard introns or protointrons (Talkish, J., et al. PLoS Genetics 2019). Cryptic 3' splice sites are indeed more protected by high confidence stems than surrounding nucleotides, suggesting a role for intron structures in preventing use of cryptic 3' splice sites. We additionally identified cryptic sites across pre-mRNA in relevant sequence

intervals that matched non-standard cryptic 3' splice sites identified by the Chanfreau lab as activated upon Prp18p inactivation (Roy, K. R., et al. bioRxiv 2023). Interestingly, these non-standard cryptic 3' splice sites were also enriched for high confidence stems, though to a lesser extent than standard cryptic 3' splice sites.

4. It is interesting to note that introns apparently have more structure than mRNAs. This made me wonder about the translational consequences of these structures. Perhaps this is a NMD “trigger” and/or produces ribosome collisions?

It is indeed possible that intron structures can trigger decay pathways. As the reviewer notes, ribosomes stalled at stable stems might trigger no-go decay (NGD) pathways leading to endonucleolytic cleavage (Simms, C. L., et al. *Molecular Cell* 2017; Parker, R. *Genetics* 2012). While NGD is not a predominant decay pathway in wildtype *S. cerevisiae*, it is possible that upon treatment with pladB, accumulating introns with highly stable stems are subjected to NGD. Alternatively, it is possible that stable stems in pre-mRNA that potentially bind proteins slow cleavage by exonucleases and endonucleases. However, dissecting the impact of these diverse pathways and processes at a genome-wide scale will require further experimental methods development that we are not able to carry out for this study. In future work, it would be interesting to disentangle the complex roles that pre-mRNA secondary structures may play in enhancing or suppressing decay pathways like NMD and NGD. We include a new discussion point on the potential impact of stable stems on NGD in the manuscript.

Minor Points:

1. Do tRNA introns (the longer ones) have any interesting structures? Or are these specific to spliceosomal-processed introns?

Secondary structures have been proposed in the set of longer tRNA introns (approximately 30-60 nucleotides long) (Ogden, R. C., et al. *Nucleic Acids Research* 1984). We confirm the presence of these proposed stems in the 10 tRNA introns for which we have available data, and we include a new analysis of these structures in Fig. S11 (described in the section “Support for proposed functional structures in *S. cerevisiae* introns”).

2. Can the authors comment on why some introns (11?) are apparently more resistant to pladB inhibition than others? I didn't see a list of these, it would be nice to know what these introns are.

It was a good suggestion to look more deeply at these introns. Upon further inspection, we found that all 11 of these introns were in genes with multiple introns or multiple reported nearby alternative splice sites (introns in SRC1, RPL7B, VMA9, DYN2, GCR1, SUS1, and RPS22B). In the case of RPS22B, coverage for the first intron was low, as expected due to a special decay pathway acting on an RNA structure in this intron (Danin-Kreiselman, M., et al. *Molecular Cell* 2003). In the remaining 10 cases, low read alignment to one isoform was artefactual due to alignment of reads to another isoform containing the same intron. We have now corrected for these cases in the following way. For some genes with two distinct non-overlapping introns (RPL7B, VMA9, DYN2, and SUS1), our reference transcripts had included both introns separately along with a single construct including both introns together corresponding to the

skipped isoform. We have now consolidated data from these alternate isoforms. We have excluded GCR1 from analysis, as this gene includes multiple overlapping introns with distant alternative splice sites that cannot be distinguished in our analysis. We have consolidated reads mapping to the two overlapping SRC1 introns into a single intron, as these two introns differ by only 4 nucleotides between alternative 5' splice sites. We have updated the Methods section to reflect these changes, and we have updated our results section in "Transcriptome-wide structure probing with splicing inhibition in *S. cerevisiae*" to describe the new retained intron fraction statistics. Additionally, we have now reported RI fractions for all introns with and without pladB treatment in Table S1 to enable further analysis of the effects of pladB on intron retention.

3. Some intron structures have also been confirmed by cryo-EM structures of spliceosomes assembled on those introns. I don't think those are included in the references on lines 196-197.

Thank you for identifying references missing in this section; we have added them now.

4. Do any of the VAR-Seq variants introduce new splice sites that are used by the spliceosome?

We have further investigated alternative 5' and 3' splice sites visible from our VARS-seq experiments. We found 6 alternative splicing events across 4 introns that were present above our cutoffs for distinguishing events (present in at least 10 UMIs and at least 3 barcodes). A summary of these events is depicted in Fig. S20 with a description in the Supplemental Text and Methods. However, these events did not appear to be regulated by structure, appearing at low frequencies across variants of multiple stem sets in our experiments including wild type constructs.

5. No scale for DMS reactivity in Figures S2 and S3

6. -Parenthesis where there should not be, p. 4 "...using reactivity data to make DMS-guided structure predictions) with a helix confidence estimate..."

7. -p. 6, 2nd paragraph under "New structures found by probing *S. cerevisiae* pre-mRNA" Fig. 2E is referenced when it should be Fig. 2B for RPS23B

8. -Fig S10A: Are the nucleotide lengths given the averages of all used?

9. -Fig S18D: one plot not labeled with intron interval it is for

We thank the reviewer for this detailed feedback. We have corrected these errors and clarified these points through the text.

Reviewer #2:

Remarks to the Author:

This study examines the role of pre-mRNA secondary structures in RNA processing pathways, using the yeast species *S. cerevisiae*. The research used a technique called dimethyl sulfate probing to examine these structures, finding widespread formations known as "zipper stems" and "downstream stems", and identified new long stems that differentiate pre-mRNA from spliced mRNA. A new method called VARS-seq was developed to analyze variations in RNA structure. Interestingly, the study suggests that some structured elements could increase levels of spliced mRNA despite not being close to canonical splice sites, while others, like zipper stems, can

increase levels of retained introns. This study brings forth new perspectives on how pre-mRNA folding affects gene expression.

Overall, I find this manuscript interesting with broad implications for splicing regulation, but the quality of the current data is not high enough to support the conclusions. The authors also present contradictory data that go against their conclusions (Fig.S11) as discussed in more detailed below.

We thank the reviewer for noting these areas for improvement. We have included new experiments and analyses to address the reviewer's comments regarding data reproducibility and other conclusions from the paper.

Major concerns:

1. It is unclear what happens during 1h of pladB treatment, are some RNAs degrading? What is the half-life of the RNA? Could this readout be measuring the state of splicing or degrading some RNAs preferentially?

We thank the reviewer for asking important questions about the effects of pladB treatment.

We expect that RNAs are degrading through pladB treatment with varying half-lives. In general, *S. cerevisiae* mRNAs have half-lives ranging from a few minutes to around an hour (Chia, LL., McLaughlin, C., Mol Gen Genetics 1979). The degradation of spliced mRNA during pladB treatment enables the steady state accumulation of unspliced pre-mRNA relative to spliced mRNA. As unspliced pre-mRNA accumulate through pladB treatment, we would also expect these unspliced pre-mRNA molecules to decay with a distinct range of half-lives, influenced by many factors including the rate of export of unspliced pre-mRNA into the cytosol and the rate of cytosolic degradation. Furthermore, it is likely that some pre-mRNAs are degraded preferentially due to premature stop codons in unspliced RNAs, secondary structures in introns, protein binding partners interact with pre-mRNA, and other factors. However, nucleases are expected to process transcripts rapidly. For instance, in one study XRN1 exonuclease activity was reported to act on between 38-55 nucleotides per second (Athapattu, U.S., et al. Nucleic Acids Research 2021). Thus, with introns degraded by exonucleases within seconds, we would expect that our structure probing data reflects full-length pre-mRNA rather than partially degraded byproducts. Indeed, when analyzing intron coverage from our pladB treated sample, we find no significant accumulation of sequencing reads at the 3' ends of introns, which suggests that our sequencing data is not dominated by partially degraded byproducts that would result from NMD with XRN1-mediated 5' to 3' degradation (analysis added in Fig. S1, discussed in the section "Transcriptome-wide structure probing with splicing inhibition in *S. cerevisiae*").

As for splicing state, we expect most pre-mRNA in pladB-treated cells to remain unspliced, as the number of accumulating pre-mRNA molecules greatly outnumber the spliceosome components in pladB treated cells. In fact, even without pladB treatment, it has been proposed that *S. cerevisiae* have approximately 200-500 copies of the U2 snRNA and around 10,000 intron copies at steady state (Ares, M., Grate, L., and Pauling, M. H. RNA 1999). With this in mind, we do not generally expect that pladB treated pre-mRNA represent multiple later stages of splicing.

It is possible that excess pre-mRNA in *pladB* treated cells are interacting with other cellular components such as the nuclear exosome or nuclear pore complex (see new additions in Discussion).

2. My biggest concern is data reproducibility.

a. Starting off, when discussing reproducibility, the use of the r^2 value, which provides clear information about the proportion of the signal distribution that is consistent across experiments, is more informative and intuitive. I highly recommend that you report all values as r^2 . Although there is no universally accepted threshold for categorizing results as "same" versus "different", the current cutoff set by you at $R=0.5$, which equates to $r^2=0.25$, seems too low. This implies that the two patterns are 25% similar and 75% different, which is not a strong enough correlation to be considered "the same". I suggest increasing this cutoff to at least $r^2=0.8$. With the current cutoff, the two signals might yield completely different structures, which could confound the interpretation of your results.

We thank the reviewer for raising important points on data reproducibility. We note that prior studies have used Pearson correlation coefficients to evaluate similarity between biological replicate data as opposed to r^2 values (for instance in Fig. S2 in Sun, L., ..., Zhang Q. C. NSMB 2019; Fig. 2 in Zubradt, M., ..., Rouskin, S. Nat Methods 2017; Fig. S1 in Huston, N. C., ..., Pyle, A. M. Mol. Cell 2021; and Fig. S3 in Liu, Z., ..., Ding, Y. Genome Biology 2021). Keeping with this standard, we report replicate correlation statistics with Pearson correlation coefficients. We understand the concern for ensuring results are reproducible between biological replicates. We have added the following analyses and discussion to the text to address this point:

- Through the manuscript, we focus on analyzing high confidence stems, with 70% helix confidence estimates and stems of length at least 4 nucleotides. In general, we do not expect stem predictions with lower confidence estimates to be reliable, due to either lack of support of the structure from DMS data or the presence of multiple competing structures. However, we find that high confident stems can reproduce known control RNA structures (Fig. S3, Table S2). In the section "Assessing structure prediction from DMS-MaPseq after splicing inhibition", we have re-emphasized that we focus analyses in the manuscript on high confidence stems. Furthermore, we include a new analysis of the reproducibility of high confidence stems in in Fig. S4 and discuss this analysis in the main text. We find that these high confidence stems are reproducible between replicates, even for introns with relatively low global reactivity correlation. In fact, these high confidence stems can help focus analyses on regions of introns that have more replicable reactivity profiles. For the 161 introns analyzed in the manuscript with $r > 0.5$, 84.5% of high confidence stems are reproduced between replicates.
- We have isolated a set of 98 introns with $r > 0.75$ and we have ensured that all secondary structures featured in main text and supplemental figures are included in this set. Only aggregate analyses (e.g. the violin plots in Fig. 3E-H) and the gallery figures (Fig. S27-28) additionally include introns with $0.5 < r < 0.75$.

In Fig. S14, we include new comparisons of secondary structure features between introns and coding sequences analogous to Fig. 3E-H only using data from introns with $r > 0.75$

between replicate reactivity profiles. We additionally present these analyses when using data from both replicates separately. In all cases, we find similar patterns in structural properties when comparing intron and coding sequences.

- We report reactivity correlation values for all introns in Table S1, allowing readers to identify the data reproducibility for their introns of interest.
- We report all nucleotides that are included in high confidence stems in Table S1 to enable readers to focus on high confidence stems in subsequent analysis.

b. It's important to understand the cause of the poor reproducibility seen in many introns. Is it because of their low abundance, leading to a limited number of RNA molecules being evaluated? Or does it stem from inadequate sequencing coverage? Or could it possibly be a combination of both? Supplementary Figure S2 indicates a strong correlation with coverage. If increasing sequencing coverage is not technically feasible, the authors should consider excluding data that exhibits low coverage and reproducibility. By focusing on high-quality data, the conclusions drawn from the study will be more reliable and impactful.

With the correlation between coverage and replicate reactivity correlation (Fig. S2), the likely cause for lower r-values between introns is due to low abundance and subsequently low coverage from sequencing data. Although pladB treatment increases the retained intron fraction for most introns, these sequences remain a small minority of transcriptome-wide RNA sequencing samples compared to more abundant RNA like rRNA. We significantly increased the number of sequencing reads obtained from replicate 1 (557 million reads) to replicate 2 (1.30 billion reads) but did not substantially increase coverage of intron regions after deduplicating UMIs, suggesting that the number of reads covering introns is likely limited by the amount of initial sample or subsequent steps in the DMS-MaPseq experimental protocol. As described above, we focus on analyzing high confidence stems that are reproducible between replicates, enabling us to draw conclusions from the data more reliably.

c. Regarding Figure 1F, it's evident that the patterns between the pladB treated and untreated samples differ, leading to distinct structural models as demonstrated in Figure S4. The authors should explicitly acknowledge the observed changes associated with the addition of pladB. It's concerning to see such significant differences induced by pladB. I'm uncertain about how the authors are able to determine the proportion of helices that remain unchanged, especially when they only have sufficient coverage for a limited number of introns without pladB.

We have included further discussion of the differences between reactivity patterns with and without pladB treatment to the manuscript in the section "Assessing structure prediction from DMS-MaPseq after splicing inhibition." Here, we discuss a new figure (Fig. S5) comparing the difference between +/- pladB treatment to the difference between replicates of the +pladB experiment. We point out intervals in the three introns with available data where reactivity values between +pladB and -pladB treatment differ substantially. However, we note that the majority of sequence positions in these introns agree well between +/- pladB conditions, with differences in reactivity values between conditions not exceeding the difference between replicate profiles. When identifying high confidence stem predictions for these three introns with

and without pladB treatment, we find that high confidence stems are shared between these conditions, further confirming that replicable regions of reactivity profiles tend to have similar structural patterns with and without pladB treatment across these introns.

3. The data shown in Figure S11, depicting strikingly different structures *in vivo* and *in vitro*, contradicts the authors' assertion in the abstract stating that "intron structures can form *in vitro* without the presence of binding partners". There appears to be a discrepancy in the presented data and its explanation, which is not adequately addressed. The authors' hypothesis that "introns could be unfolded during *in vitro* refolded total RNA due to intermolecular interactions" seems contradictory. Intermolecular interactions would presumably reduce DMS accessibility and lead to the appearance of more structured RNAs. As the results of this experiment deviates considerably from the rest of the manuscript, further testing and explanation are needed, especially since it challenges a key point of the manuscript.

We agree that it is important to revisit results from DMS probing of *in vitro* refolded RNA. In response to earlier suggestions to focus only on introns with reproducible reactivity profiles, we found that for the *in vitro* refolded RNA condition, only 9 introns had $r > 0.5$ and only 3 introns had $r > 0.75$ between replicate reactivity profiles. Lower reproducibility in this experiment compared to the *in vivo* DMS probing experiment could be due to multiple factors, including fewer sequencing reads allocated for *in vitro* samples and loss of material during other library preparation steps. When comparing *in vivo* and *in vitro* structural patterns based on the 9 introns with $r > 0.5$, including Gini coefficients, MEE, helix confidence estimates, and longest stem lengths, we found that no comparisons showed significant differences at a threshold of p-value 0.01. With only a few introns meeting coverage thresholds, we have removed analyses based on these *in vitro* refolded RNA experiments from the manuscript.

4. Figure 6, focusing on the Var-seq functional assay, shows that the presence of different barcodes leads to the greatest variation in RIFraction. The impact of the barcode, which should ideally be minimal, appears bigger than the difference between the Wildtype and the Variant. This effect needs to be understood and minimized before making comparisons. The authors should also provide information about:

- The actual effect size in terms of fold change in mRNA expression or RI.
- The consistency of barcode distributions. A few outlier barcodes could potentially shift the overall distribution significantly.

We agree that the variation between barcodes in VARS-seq is high and should be better understood. In general, as most introns had low RI fractions in the VARS-seq assay, barcodes were necessary for effects to become visible, with some barcodes leading to higher RI fractions and thus allowing structural effects to become visible in the dynamic range of VARS-seq. As we do not expect randomized barcodes to systematically interact with variant sequences, significant differences between wildtype, variant, and rescue sequences can be discerned through comparisons of distributions of RI fractions across barcodes.

To better understand whether RI differences due to barcode sequences were due to sequencing artefacts, data processing issues, or true biological effects, we obtained RT-qPCR measurements for RI fractions from individual strains constructed from barcodes expected to generate low or

high RI fractions from VARS-seq (Fig. 6F). As RT-qPCR experiments showed significant differences in RI fractions based on the barcode sequence that aligned with VARS-seq, the effects of barcode sequences are likely due to biological effects and not simply artefacts of the RNA sequencing experiment or data analysis pipeline in VARS-seq.

We note that VARS-seq reports lower RI fractions than our RT-qPCR experiments. Each of these assays is expected to be impacted by distinct length-dependent biases. For instance, it is possible that size selection steps in the VARS-seq protocol enrich for shorter spliced mRNA products due to incomplete RNA fragmentation. Notably, prior data on RI fractions for RPS9A fall between our measurements from VARS-seq and RT-qPCR (30.8% RI from Hunter, O., et al. RNA 2024; ~60% RI from RT-qPCR; ~5% from VARS-seq). Due to these biases, we do not recommend directly obtaining absolute RI fractions from VARS-seq. Additionally, effect sizes will be dependent on barcode sequences and the dynamic range of the assay. With this in mind, we include the following caveat in the main text: “we suggest using VARS-seq to generate hypotheses on relative rather than absolute splicing differences, and we suggest verifying individual examples of interest with independent assays.”

5. For validation purposes, it would be more convincing to provide individual well-characterized examples with fold change and effect size of structure on function, even if it's just one for each, to support the discovery of new regulatory mechanisms. Specifically, the findings that certain structured elements can increase spliced mRNA levels despite their distance from canonical splice sites, and others, like zipper stems, can enhance levels of retained introns.

We agree that it is important to validate VARS-seq with individual examples characterized by well-established lower throughput techniques. For this, we carried out RT-qPCR on individual strains as an orthogonal assay to obtain RI fold changes and effect sizes. For each structural hypothesis, it was critical to generate at least 4 strains with wildtype, 5' mutant, 3' mutant, and rescue sequences. We have carried out these experiments for 3 stem sets and 2 barcodes, generating 16 *S. cerevisiae* strains and measuring RI fractions and normalized mRNA levels with RT-qPCR (Fig. 6F and Fig. S22).

The pattern of RI fractions on the stem set from RPS9A aligns well with observations from VARS-seq, with wildtype and rescue sequences having higher RI than variants. Barcode sequences that led to high RI fractions in VARS-seq also produced high RI fractions with RT-qPCR, with effect sizes between structure variants less visible due to high RI fractions. However, RT-qPCR results from the stem set in RPL36B do not show significant differences in mRNA levels visible from VARS-seq, perhaps because the individual variant and barcode sequence tested by RT-qPCR do not show effects found from VARS-seq when aggregating data across multiple variant sequences. We have changed text in the abstract to be more conservative, reflecting only conclusions that could be verified by independent RT-qPCR assays, focusing on the discovery of structured elements that can alter retained intron levels despite being distal from canonical splice sites.

Reviewer #3:

Remarks to the Author:

The authors attempt to elucidate the landscape of RNA secondary structure in the yeast transcriptome. This landscape is largely unknown for any species and is important for improved understanding of the regulatory role of RNA in living systems. This problem is critically important, both from the perspective of basic research and from the perspective of mRNA therapeutic and vaccine development. Specifically, the team focuses on splicing and pre-mRNA secondary structures. Using pladB, the team is able to inhibit splicing, effectively trapping for analysis pre-mRNA secondary structures across the transcriptome. This is a key step, since without doing this, a large ensemble of spliced and unspliced transcripts exists, making it difficult to disentangle. Using chemical probing strategies developed by the authors, the team finds highly structured upstream regions ('zipper stems' towards the 5') and downstream regions ('downstream stems' towards the 3'). They find interesting differences in the structure of the mRNA between coding and noncoding regions. They perform a careful comparison between *in vitro* and *in vivo* systems and detailed phylogenetic analysis. Importantly, they develop a high throughput functional assay, where they produce variants of secondary structure elements and test the effect of these variants on splicing activity. The paper is well written and well organized. The figures are excellent and the data is compelling. There is an excellent supplementary information section. The results have broad and general implications for life sciences as a whole, for the readership of NSMB and beyond. The results should be published without delay.

We thank the reviewer for the kind feedback and the helpful comments, which we have addressed with added discussion and explanations in the text.

Comments:

1. The hypothesis that 'introns could be unfolded *in vitro* and refolded total RNA due to intermolecular interactions in the complex RNA mixture' lacks support and is unclear. It seems more probable that non-equilibrium effects, kinetic effects, and environmental perturbations to the free energy landscape are more likely culprits in remodeling the folding landscape.

As discussed in response to Reviewer 2, we no longer include results from the *in vitro* refolded condition as we observed poor reproducibility between replicates for this condition. We instead focus our analysis on *in vitro* probing of individual *in vitro* transcribed introns.

2. For the 'zipper stem and downstream stem free energy', a brief discussion of the limitations of the free energy calculations should be included, especially in terms of how the entropic contribution is captured, the importance of sampling 3D configuration space, and the importance of magnesium.

These limitations are important considerations for estimating free energy parameters for RNA secondary structures. In this manuscript, we computed free energies for zipper and downstream stems using the nearest neighbor model in Vienna 2.0, which includes entropic and enthalpic contributions from individual base-stacks but does not include non-local contacts that might be found by sampling in 3D space. We include a discussion of these considerations in the Methods section "Zipper stem identification and stability calculation," and we point to the Methods section in the main results.

3. While R-scape has merits, the limitations of R-scape could also be discussed (e.g., Tavares, et al., 2019).

We have added a new sentence to the Discussion to point out some potential limitations of R-scape, remarking that it can miss functional structures in cases with low quality sequence alignments, high conservation of positions in alignments, or few sequences in alignments.

4. More details should be included in the description of the 3D models. For example, ‘we built models for variants’. How were the models ‘built’? Which software/code was used, what algorithms did the code use to do the building? How were ends of fragments handled? How were conformations determined? If multiple options model options (there a models that can fit the given constraints), how were the choices made? How were the nucleotides ‘replaced’? Which alignment algorithm was used? How was the alignment performed? How was the ‘docking’ performed? Which docking algorithm was used? Why this algorithm and why not a different algorithm? What are the limitations of using a rigid body approximation? It looks like electrostatics were neglected (or that a simple electrostatic relation was used). What are the limitations of this? What are the limitations of not included explicit solvent and explicit magnesium? How were nucleotides ‘replaced by corresponding sequences’? Since you don’t have an experimentally determined 3D structure for these sequences, what are the limitations of using a model?

We thank the reviewer for noting these points that required clarification. We have mentioned limitations of our 3D modeling approach in the Results section “New structures found by probing *S. cerevisiae* pre-mRNA”, and we have expanded on our modeling approach and its limitations in the Methods sections “Zipper stem identification and stability calculation” and “Modeling of intron stems in spliceosome structure”.

5. It would be interesting to discuss the implications of this study of mRNA secondary structure for riboswitches and long non-coding RNAs (e.g., Novikova, et al., JMB 2013), as well as how a similar approach could be attempted in 3D (e.g., Kim, et al., Nat. Comm. 2020) and if useful information could be gained from a 3D transcriptome wide approach.

We agree that it would be interesting to understand intron structures with approaches that can identify 3D contacts. One step towards this goal could be to use recent approaches like KARR-seq which can map higher order RNA contacts across the transcriptome (Wu, T., et al. Nature Biotechnology 2024). It would be interesting in the future to use these approaches in conjunction with pladB treatment to observe contacts in introns. We have suggested this new direction in our Discussion section.

Reviewer #1:

Remarks to the Author:

the authors have thoroughly responded to the prior reviews by additional analyses and some clarifications. I think this is a wonderful addition to the field and is ready for publication. I have just a few minor suggestions.

The observations that structures are depleted around splice sites, perhaps enriched around cryptic sites (esp. 3'SS), and are formed in a way compatible with spliceosome 3 dimensional structure are all highlights of the manuscript for those thinking about splicing mechanism. Indeed, to me it seems that substrate (pre-mRNA) structure may play just an important role in splicing fidelity as the proofreading ATPases!!! I really think this needs to be highlighted as a main text figure. If figures are limited, I would suggest moving Fig. 4 (in vitro to in vivo comparison) to supplemental and adding in a new figure based on the supplemental figure S13 (perhaps polished up a bit to make it easier to understand what is being plotted) that can be added to the main text. I think these are all important, fundamental observations.

We agree that the new analyses of the enrichment of structures at cryptic splice sites and the depletion of structures around canonical splice sites suggests an interesting potential functional role for intron structures. We have moved figures describing these results into panels of the main text Fig. 3 to highlight these findings.

Reviewer #2:

Remarks to the Author:

I appreciate the authors' efforts to enhance the manuscript and discussion with additional data, particularly concerning the structural changes and the induced unnatural cell states following PladB treatment. The acknowledgment of the simplifications in the 3D modeling approach ("We note that the 3D modeling approach here makes simplifying assumptions") reflects a commendable level of transparency. However, there are major concerns regarding the manuscript:

1. Chemical Probing Data and Zipper Stems: The chemical probing data presented does not convincingly support the existence of zipper stems in cells. Furthermore, the efforts to validate VAR-seq were partially successful; it was validated by RT-qPCR for one example (RPA9A, Fig.6F) but not for another (RPL36B, Fig. S22A).

From our transcriptome-wide DMS-MaPseq experiment, zipper stems are only annotated if they have helix confidence estimates of at least 70%. At this confidence threshold, 88% of stems across introns agree between replicates. We also note that 32 zipper-stem containing introns had between-replicate DMS reactivity $r^2 > 0.6$ from DMS-MaPseq.

As an additional, prospective evaluation of zipper stems, we have carried out new targeted DMS probing experiments for 36 introns containing zipper stems to increase coverage and reproducibility between replicate measurements, achieving $r^2 > 0.9$ for 24 of these introns. Zipper stems were present in 29 of the 36 introns probed with targeted DMS probing when generating structures guided by these DMS data, further confirming their presence. In our manuscript we have focused results for targeted DMS probing on the 30 introns which had $r^2 > 0.6$ from DMS-MaPseq, of which 25 introns had zipper stems. We have described these new experiments in the Results sections "Assessing structure prediction from DMS-MaPseq after splicing inhibition" and "New structures found by probing *S. cerevisiae* pre-mRNA", in Fig. S4, Fig. S5, and in the Methods section. Structures from targeted DMS probing are included in Table S1. We additionally note that in the case of RPL36B, mutations that break zipper stems alter steady state

spliced mRNA levels, and compensatory mutations that restore these stems also restore wildtype spliced mRNA levels, further supporting the existence of zipper stems in cells.

With respect to VARS-seq, we expect that there are some differences between RT-qPCR and VARS-seq due to differences in dynamic range between the assays, along with increased statistical power from VARS-seq from aggregating data across variant and barcode sequences. Since we only generated one variant and rescue strain for the RPL36B zipper stem, RT-qPCR measurements may not be able to recover the effects found from VARS-seq when looking across multiple barcode and variant sequences. Due to these potential differences between assays, we have noted in the text that “we suggest verifying individual examples of interest with independent assays.”

2. Functional Ambiguity: The role of zipper stems remains unclear, with the manuscript speculating that not all zipper stems enhance splicing by bringing splice sites closer (“Unexpectedly, zipper stem variants in the RPL28 intron (Fig. 6C) lowered RI, suggesting that not all zipper stems promote splicing by co-localizing splice sites”). This lack of clarity on their function adds to the speculative nature of the findings.

We expect that zipper stems play multiple roles in cells, for instance slowing intron decay in some cases and facilitating splicing in others. Because these have opposing effects on retained intron levels, the net effect is hard to predict. In our experiments, variants disrupting the RPL28 zipper stem lowered retained intron levels, which is consistent with zipper stems potentially slowing intron decay. However, in the case of RPL36B, RT-qPCR of variants disrupting zipper stems show increased retained intron levels, supporting previous findings that zipper stems in RPS17B enable efficient splicing (Rogic, et al. BMC Genomics 2008). We expect that zipper stems can shift gene expression patterns through multiple mechanisms, and we believe that detailed dissection of the roles of zipper stems in individual cases will be an important challenge but is outside the scope of this work.

3. Quality and Reproducibility Concerns: The evidence for the existence of these zipper stems in cells is weak, and the data quality and reproducibility are questionable. The manuscript risks lowering the reproducibility standards within the field, which is a major concern. It is crucial for the authors to acknowledge the low reproducibility explicitly in the text and to justify the potential lowering of these standards if the manuscript is to be published.

As discussed above, zipper stems have been annotated only if they have high helix confidence estimates (at least 70%). We have additionally carried out targeted DMS probing experiments for 36 introns containing zipper stems, with replicate DMS reactivities having $r^2 > 0.9$ for 24 introns (Fig. S4, Fig. S5). We confirmed the presence of zipper stems in 29 of the 36 tested introns (Fig. S5, Table S1).

We appreciate the concerns about data reproducibility. To increase the reproducibility of results reported in our manuscript, we have updated our manuscript to restrict all results to the 88 introns with between-replicate reactivity $r^2 > 0.6$. Intron structures are now reported in Fig. S28, Fig. S29, and Table S1 only for introns passing $r^2 > 0.6$, and all aggregate analyses (e.g. in Fig. 3, Fig. 5) have been updated to include only these 88 introns. These updated correlation cutoffs align with replicate reproducibility from previous studies (e.g. $r > 0.75$, or $r^2 > 0.56$, for the top 60% abundant transcripts in Sun, et al. NSMB 2019). By focusing on high confidence estimate stems, we note that we further improve the reproducibility of our conclusions, as 88% of these stems are shared between DMS-MaPseq replicates. Furthermore, we note that 90.5% of all high confidence stems from our original DMS-MaPseq experiment matched with structures from our new targeted DMS probing experiments, where 24 introns had $r^2 > 0.9$ (Fig. S4).

4. Statistical Rigor: I strongly recommend that the authors report r^2 values to enhance the manuscript's statistical rigor, especially in the context of variance between replicates. Historically, different conventions for reporting correlation coefficients (such as using r instead of r^2) do not justify continuing to exclude these important values. Reporting r^2 is essential as it quantifies the proportion of variance in the dependent variable that is predictable from the independent variable, providing a clearer and more complete understanding of the data consistency across replicates.

We understand this perspective that it is better to report the square of the correlation coefficient r^2 , and we have updated our manuscript to use r^2 throughout.

5. Analysis of DMS Constraints: The manuscript could benefit from additional analyses, such as comparing the DMS signal from a denatured control to assess if similar stems are predicted. This could help clarify whether the high-confidence stems identified are genuinely influenced by the DMS constraints or merely reflect strong predictions by the RNA Structure Fold algorithm.

Given these concerns, it's crucial to consider whether this manuscript, as it stands, contributes significantly to the field or merely points out potential areas for future research without confirming the proposed structures.

We agree that it would be good to check that DMS data are informing stems identified through DMS-MaPseq. We have carried out new targeted DMS probing experiments for 30 introns after heat-denaturing RNA extracted from *S. cerevisiae*. The resulting structures identified using RNAstructure show global unfolding of these introns (Fig. S4, Fig. S5). Across the 22 introns with $r^2 > 0.9$ between replicates of targeted DMS probing of denatured RNA and $r^2 > 0.6$ from DMS-MaPseq, 96 stems were present from DMS-MaPseq compared to only 18 stems from the denatured RNA samples. In some cases, remaining high confidence stems formed in the denatured RNA conditions involve sequence positions in primer-binding intervals used for targeted probing, where reactivity data cannot be obtained to inform stem predictions (gray stretches in Fig. S5F, Fig. S5I). Notably, these 30 introns were selected because they contained zipper stems as identified by DMS-MaPseq, but none of these introns contained zipper stems after denaturing RNA. These results confirm that DMS data are indeed informing structures generated from RNAstructure.